# Research on the Sustainability of Channel Strategy Selection on the Overall Efficiency of Listed Retail Enterprises: Evidence from China

**Nuo Chen** [1] , **Chong Ye** [1] **and Jiaonan Wang** [2,*]

1   School of Economics and Management, Fuzhou University, Fuzhou 350108, China
2   School of Economics and Management, Tongji University, Shanghai 200092, China
*   Correspondence: wjn@tongji.edu.cn

**Abstract:** Appropriate channel strategies can help to improve the operational efficiency of retail firms and stimulate the domestic demand market. To study the sustainability of retail enterprises in the COVID-19 era, this paper uses the data of 29 Chinese listed retail enterprises from 2012 to 2021 as a sample, and uses data envelopment analysis (DEA) to evaluate and analyze the efficiency of the listed retail enterprises that implement different channel strategies. The results of the study show that: (1) The implementation of channel strategy will re-integrate the information flow, capital flow, commercial flow and logistics of enterprises, and affect the efficiency of enterprises through four paths: market demand, channel resources, channel technology and channel system. (2) The implementation of dual-channel strategy enterprises, in the short term, has greater technical efficiency than single-channel strategy enterprises. (3) The channel integration strategy of online ordering and store delivery has a significant positive impact on the overall efficiency of listed retail enterprises. (4) The inventory turnover rate has a significant positive impact on the overall efficiency of the listed retail enterprises. The level of accounts receivable, the level of retail technology, the level of investment in fixed assets and the size of enterprises have a negative impact on the overall efficiency of the listed retail enterprises.

**Keywords:** retail efficiency; sustainability; retail enterprise; channel strategy; DEA model

## 1. Introduction

The negative impact of the COVID-19 on China's economy still exists, and people's living habits and consumption patterns have changed. With the major domestic cycle as the main body and the dual domestic and international cycles promoting each other, this is the new pattern to accelerate economic recovery. In the whole domestic cycle, consumption has become the first pulling force of economic growth for six consecutive years, and it is the main focus of accelerating the release of domestic demand potential, but also promoting the key engine of the dual cycle. As a key vehicle to guide production and expand consumption, the retail industry is directly related to the speed and efficiency of commodity circulation, and is a key support force for national economic growth and social development. Since the retail industry is characterized by high revenue and low profit, the management efficiency of the retail industry is particularly important. Whether the current retail channel strategy can improve the efficiency of retail enterprises and reverse the decline is an urgent and important issue in the development of the retail industry.

Nowadays, the retail industry has become a key force for national economic growth. Not only that, the retail industry has a large number of points and a wide range of characteristics, but also a large volume of labor demand, coupled with a low employment threshold, accommodating much of the laid-off and transferred labor; the contribution to national employment is also particularly outstanding. The retail industry has gradually

become a major industry for promoting economic growth and national employment; helping retail innovation and transformation, improving circulation efficiency and promoting consumer upgrading will become key initiatives for high-quality economic development. However, as economic development enters a new normal, the demographic dividend is nearly exhausted, economic growth and residential consumption growth is slowing down, the expansion of China's retail market is restricted after the pandemic, the development of internet technology has extended retail channels from offline to online, the real economy has been hit, and the new network mobile channels have further intensified the brutal competition pattern in the retail industry. Due to the rapidly changing consumption habits and consumption concepts of the consumer groups, rising retail operating costs and the rapid development of online channels, retail companies are not only facing a series of uncertainties in their investment in online channels, but also the layout and planning of their offline stores and other preliminary investments make it difficult to obtain equivalent returns in the short term. This series of reasons led to the revenue of many retail enterprises' not being equal to their operating costs. In the face of the above dilemma, can the current channel strategy of retail enterprises improve the overall efficiency of the enterprise? In the era of COVID-19, how should retailers choose their channel strategies to improve their overall efficiency, and maintain their business and sustainable development in the future?

In summary, changes in the macro environment and intensified competition in the industry have prompted continuous changes in retail channels, and retail transformation is urgent. It is of great theoretical and practical significance to study the relationship between channel strategies of retail enterprises and the overall efficiency of enterprises to transform the retail industry and stimulate market vitality. Considering that the sample needs to meet the requirements of comparability, stability and data availability, this paper first selects the retail enterprises listed in Shenzhen and Shanghai before 2011 according to the sample study period, and excludes the enterprises with major restructuring and missing main variables in the sample period. In addition, in order to ensure that the selected sample is representative of the retail industry, this paper takes the revenue share of retailing in the actual operation of enterprises as the standard, according to the regulations of China Tax Law and the Commerce Bureau, and eliminates enterprises whose main business share of retailing is below 60%, and finally, includes 29 retail enterprises as the final research sample of this paper. Internationalization has increasingly become an important issue and an inevitable development trend in the modern retail industry. This study focuses on Chinese enterprises and analyzes the impact of channel strategy on the efficiency of these 29 listed retail enterprises through data envelopment analysis (DEA) and tobit regression methods, with a view to improving the understanding of channel strategy for the retail industry at home and abroad. The aim is to raise awareness of channel strategy for the retail industry at home and abroad, and to provide useful and feasible suggestions for the channel selection of modern retail enterprises.

The study of this paper is divided into seven parts. The first part is the introduction. It introduces the research background of this paper in detail, presents the research topic, and explains the significance of this paper. The second part and the third part mainly comb through the literature related to retail channel strategy and retail firm efficiency, analyze the path and the inner mechanism of constructing retail channel strategy on firm efficiency, and provide methodological and theoretical references for the following empirical analysis and research. Part 4 and Part 5 will conduct the empirical analysis according to the common process of DEA efficiency research. In the fourth part, the efficiency of 29 Chinese listed retail companies is measured by applying the DEA method, and then classified and analyzed according to the different channel strategies implemented. In the fifth section, the impact of channel strategy on the overall efficiency of retailing is empirically demonstrated by using the global covariate efficiency value measured by DEA Malmquist as the explanatory variable, introducing a dummy variable regarding whether the channel strategy is implemented or not, and controlling for other relevant internal and external possible influencing factors. The sixth section presents the practical and theoretical

implications and recommendations of this paper. Finally, the conclusion section presents conclusions to promote the overall efficiency improvement of listed retail firms in their future sustainable development.

## 2. Literature Review

A retail channel refers to the way in which retail enterprises provide goods and services to final consumers, and is an important component of the marketing system of retail enterprises. It is widely believed, at home and abroad, that there are four shifts in retail channel strategy: single-channel strategy, multi-channel strategy, cross-channel strategy and omni-channel strategy. Among the above four channel strategies, multi-channel strategy mainly focuses on the increase in the different natures of retail channels, while omnichannel strategy places more emphasis on the seamless connection and synergistic cooperation between channels [1]. The channels involved in the single-channel strategy to cross-channel strategy are independent of each other and lack efficient integration. In contrast, the omnichannel strategy is a decision based on a "consumer" strategy [2], in which retailers effectively integrate and optimize multiple channels to deliver effective customer value at every stage of the consumer experience [3]. It is generally accepted that channel integration is the core way to implement omnichannel management [4–6], and the following two different channel integration strategies are mainly tried by retail companies in the implementation of the omnichannel. The first channel integration strategy emphasizes that companies can simultaneously transact and communicate with online or offline channels, which will then reduce transaction concerns, such as customer distrust of products, and will increase the convenience of transactions, thus playing a role in customer retention [7]. The second strategy emphasizes information integration or process integration to increase the degree of synergistic interaction between different channels. Information integration emphasizes the sharing of channel information, for example, providing online URLs in offline entities and sharing offline store information in online channels, so that online and online traffic can be mutually pulled [8,9]. Process integration, on the other hand, refers to the integration of order fulfillment functions or realization methods of channels of different natures by retail enterprises, which is manifested in real life as an online store purchase, store pickup or store delivery [10–12]. In our study, we focus on the channel integration strategy of the second model, i.e., the impact of the online purchase store distribution model on the overall efficiency of retail enterprises.

At this stage of research on the efficiency of retail enterprises, foreign countries still mostly use profitability as the main efficiency indicator for relationship research, but some scholars have suggested that the profitability of enterprises is more related to the external environment, and there is a deficiency in using this indicator to represent efficiency [13]. The previous studies on the influencing factors found that the investment in employee training, enterprise scale, technology investment and foreign investment introduction will have a positive impact on enterprise efficiency [14–17], but the relationship between enterprise equity structure and enterprise efficiency has been controversial [18]. Previous domestic studies have been conducted both at the micro level of enterprises and from the macro industry perspective to study the measurement and evaluation of efficiency and the main influencing factors. In terms of research methods, domestic and foreign are basically consistent, usually using data envelopment analysis, stochastic frontier analysis, tobit regression and other methods.

Some studies refer to the combination of operational management efficiency and enterprise scale efficiency as technical efficiency, while others categorize it as total factor productivity. Since total factor productivity of retail enterprises can be understood as a comprehensive evaluation of the rationality of organizational planning, scientific staffing and innovative technology introduction, it can also be used as a comprehensive efficiency of retail enterprises to measure the sustainable development capability of retail enterprises. In this article, we will also use total factor productivity to analyze the impact of channel strategy selection on the overall efficiency of listed retail enterprises based on the design of

an input–output system suitable for the retail industry, which is referred to as the overall efficiency below.

Most of them only analyze the efficiency of e-tailing enterprises and offline physical enterprises by comparing the DEA efficiency results alone, and the research results are not uniform. Wang et al. sorted out the channel integration motives and measured the efficiency of 41 retail enterprises using the DEA method and found that the enterprises implementing channel integration strategy were more efficient [19]. Huang et al. measured the level of channel development of retail enterprises through rooting theory and introduced a model to empirically prove its positive and significant impact on enterprise efficiency [20,21]. Yan et al. wanted to illustrate through a case study that the implementation of channel integration strategy should positively affect the efficiency of traditional retail enterprises by expanding the market, increasing consumption frequency and reducing transaction costs [22]. By comparing the DEA efficiency of traditional brick-and-mortar enterprises and pure e-commerce retailing, Lei et al. found that the latter had higher overall efficiency and scale efficiency than the former [23]. Lei compared and analyzed the efficiency of three types of enterprises with pure offline channel operation, pure e-commerce operation and multi-channel operation by the DEA model, and the results showed that the efficiency of enterprises with a pure e-commerce operation and multi-channel operation was better and much higher than that of enterprises that kept offline single-channel operation [24].

## 3. The Path of Channel Strategy's Impact on Retailer Efficiency

Previous studies have mentioned market demand, channel resources, channel technology and channel system as key elements influencing the formation and evolution of the channel ecosystem in the study of enterprise channel ecosystems [25]. The implementation of channel strategy will reintegrate the information flow, capital flow, commercial flow and logistics of enterprises to influence market demand, channel resources, channel technology and channel system, thus promoting better development of enterprises.

### 3.1. Path of Market Demand

Both the new channel strategy and the channel integration strategy enable companies to reap the benefits of increased consumer demand. The new channel strategy allows traditional retailers to enter new markets and achieve economies of scope in a short period of time. Tang, in his 2014 and 2015 studies, pointed out that retail online markets have more advantages in terms of economies of scope and technological innovation, with retailers expanding much faster in online markets than in traditional offline markets and requiring less investment [26,27]. Channel integration strategies can enhance loyalty by providing value-added services to consumers and designing inter-channel communication interactions. Wallace et al. showed that consumers perceive higher levels of service from multi-channel firms compared to single-channel firms, and consumers are correspondingly more loyal to multi-channel firms [28,29]. Cao et al. theoretically argued that channel integration strategy can change the sales scale of a company through five paths: increasing the consumer experience, improving trust of customers', increasing loyalty of customers', increasing the demand conversion rate and promoting mutual channel attraction [30]. The convenience that comes with implementing a channel strategy will also enhance business revenue. Some studies have shown that firms implementing cross-channel strategies will attract more consumers to increase the frequency of consumption using promotional offers, and the implementation of cross-channel strategies will attract a higher number of consumers and gain higher revenue [31]. Avery et al. stated that consumers will buy more goods using a combination of channels in cross-channel centralized purchasing [32]. Rangaswamy et al. then argued that Neslin used a VAR model to empirically demonstrate the impact of adding offline channels for pure e-merchants by analyzing the impact of the new channels on the firm's revenue in terms of the number of customers, order frequency and return frequency [33]. The results found that the new offline channel had a small

diversion effect on the original online channel, and the online order frequency instead increased and the overall revenue improved [34].

Although the implementation of multi-channel strategy has complementary effects between channels and can improve customer satisfaction by improving service quality, considering utility maximization, consumers will prefer the channel with the highest utility. That is, the higher the service quality of a channel of a retailer, the lower the shopping willingness of customers in other channels, and there is a competition and substitution effect between different channels of the company [35].

### 3.2. Path of Channel Resource

According to the resource-based view, corporate resources are defined as all tangible and intangible assets controlled by the firm that enable the firm to improve the efficiency of resource allocation through the implementation of strategies [36,37]. Adding new channel strategies or carrying out channel integration strategies require huge costs for retail firms, and although the positive impact of channel strategies has a lag, the huge costs invested will cause a capital cost disadvantage for the firm in the present, and even have the disadvantage of increasing costs in the future implementation process. In particular, the new online channels are relatively new for the entity companies; on the one hand, they do not need to invest in store rental fees, and on the other hand, they strengthen the circulation efficiency, thus achieving the role of lower transaction costs [38]. However, it is not clear whether the corresponding increase in overhead and sales costs can be offset by the savings in transaction costs. In addition to the difficulties in implementing channel integration strategies, most retail enterprises have failed to achieve effective integration of resources, in which case, further investment costs will only hinder the improvement of retail enterprise efficiency.

### 3.3. Path of Channel Technology

Retailers that add online channel strategies through independent R&D not only need to invest a lot of channel R&D costs in channel development and construction, but achieving new channels with the help of other platforms is also an indirect introduction of technology, which is more suitable for small enterprises; on the one hand, they will not be burdened by excessive R&D expenditures, and on the other hand, they can keep pace with the technology of the times. Lai proposed that online big data resources to help offline retail store operations is the key to achieving enterprise channel integration [39]. Wang believes that digital technology and artificial intelligence are the key "boosters" for retail transformation and upgrading in the digital economy, and retail enterprises can implement corresponding channel strategies and introduce key technologies to realize intelligent operations and create a new ecosystem [40].

### 3.4. Path of Channel System

The channel system of the enterprise is the management system of the enterprise. Andreini believes that a multi-channel strategy will improve the operation and management ability of enterprises because the different nature of channels have their own advantages and can also lead to each other, and enterprises will offset the overall management ability of different natures by improving their own marketing level and service level The cannibalization effect brought by different channels will be offset by improving their marketing and service levels [41].

As a result of combing through the previous literature, the path framework diagram of channel strategy affecting the efficiency of retail enterprises constructed in our study is shown in Figure 1.

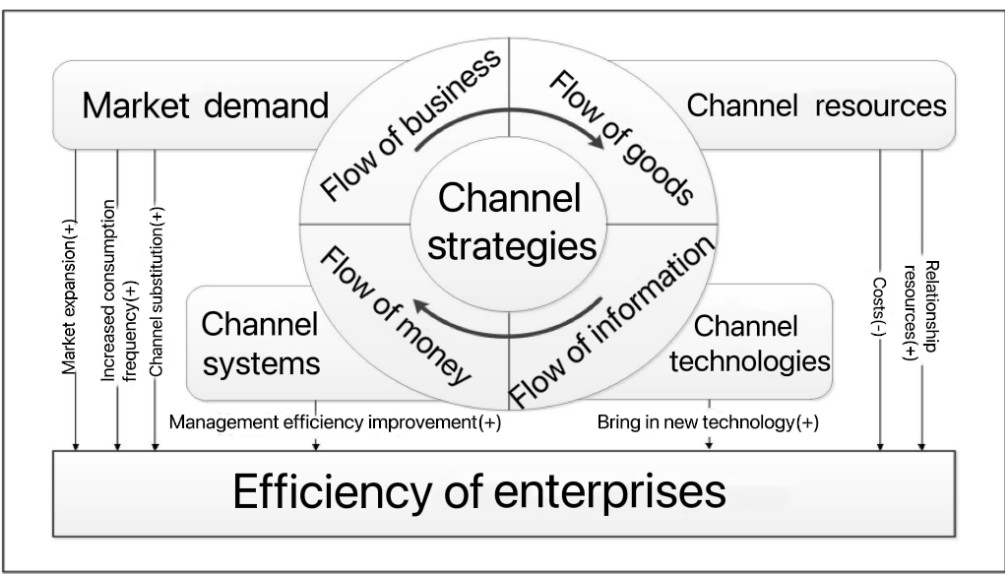

**Figure 1.** A pathway framework for channel strategy to influence retailer efficiency.

## 4. Comprehensive Efficiency and Analysis of Listed Retail Enterprises in China

### 4.1. DEA Malmquist Index

The DEA Malmquist index measures total factor productivity (TFP). Total factor productivity of retail enterprises can be understood as a comprehensive evaluation of the rationality of organizational planning, scientific staffing and innovative technology introduction, as well as a comprehensive efficiency of retail enterprises, to measure the sustainability of retail enterprises. DEA Malmquist index can be divided into two categories according to the method of its reference frontier: first, the Malmquist Index (MI) is calculated by referring to the frontier of different periods; second, the MI is calculated by referring to the same frontier for all periods. The MI can be decomposed into the combined technical efficiency change (EC) and the technical change (TC), where the technical progress change is the change in the technical progress of industry production in a given period. The combined technical efficiency change can be decomposed into pure technical efficiency change (PEC) and scale efficiency change (SEC). The relationship between these indicators can be expressed as follows: MI = EC × TC and EC = PEC × SEC. When the MI is greater than 1, it means that the efficiency of the enterprise has improved, and vice versa, it means that the efficiency has decreased. When one of the constituent MIs is greater than 1, it indicates that the index is contributing to the efficiency of the enterprise and vice versa. The MI calculation refers to the same frontier for all periods. There are three types of DEA Malmquist indices that use this rolling approach to covariance, namely, adjacent covariance, serial covariance and window covariance. There are three types of MI for the same frontier covariate: fixed covariate, global covariate and window covariate. Since the calculation of global covariates is equivalent to increasing the number of DMUs, it can improve the frontier accuracy and thus make the results more stable and reliable. Therefore, the global covariance Malmquist model was chosen in our study. As mentioned above, the global covariance Malmquist model uses all units under all sample periods as the reference set, i.e., the reference set $S^g$ for each period is $S^g = S^1 \cup S^2 \cup \cdots \cup S^P = \left\{ \left( x_i^1, y_j^1 \right) \right\} \cup \left\{ \left( x_i^2, y_i^2 \right) \right\} \cup \cdots \cup \left\{ \left( x_i^P, y_j^P \right) \right\}$.

The results of the global covariate Malmquist index measure are the efficiency ratios, and $E^{\text{reference sets}}$ (the evaluated DMU) is used in all subsequent cases to represent the efficiency derived through the DEA model.

$$\text{MPI} = \frac{E\left( x^{t+1}, y^{t+1} \right)}{E\left( x^t, y^t \right)} \tag{1}$$

Although the reference frontier surface is the same for the DMU calculations for two consecutive periods, the calculation of the combined technical efficiency changes uses the respective frontiers.

$$EC = \frac{E^{t+1}\left(x^{t+1}, y^{t+1}\right)}{E^t(x^t, y^t)} \tag{2}$$

The extent to which the frontier t+1 is close to the global frontier can be expressed as $\frac{E\left(x^{t+1}, y^{t+1}\right)}{E^{t+1}\left(x^{t+1}, y^{t+1}\right)}$, the extent to which frontier t is close to the global frontier can be expressed as $\frac{E^g\left(x^t, y^t\right)}{E^t(x^t, y^t)}$, the larger the ratio the closer; frontier t+1 compared to frontier t is the technical change.

$$TC = \frac{E\left(x^{t+1}, y^{t+1}\right)/E^{t+1}\left(x^{t+1}, y^{t+1}\right)}{E(x^t, y^t)/E^t(x^t, y^t)} = \frac{E\left(x^{t+1}, y^{t+1}\right)E^t(x^t, y^t)}{E^{t+1}(x^{t+1}, y^{t+1})E(x^t, y^t)} \tag{3}$$

In summary, considering that the global covariate DEA Malmquist index measures the efficiency changes in decision units in different periods, it is more applicable to panel data. Our study constructs a panel regression model using the efficiency value calculated by the global covariate as a proxy variable for the overall efficiency of listed retail enterprises to further analyze the impact of channel strategy on the efficiency of retail enterprises.

### 4.2. Selection of Evaluation Targets

Our study selects retail listed enterprises as the research sample; mainly considering the authenticity and availability of the data, listed enterprises have a mandatory information disclosure compared with non-listed enterprises, the information is more open and transparent, and the data obtained is more real and reliable. The specific selection of sample enterprises is mainly combined with the characteristics and requirements of the DEA method from the following three aspects: First, the selected sample is representative. Therefore, the retail listed enterprises in the sample selected in our study have been listed before 2011 and still occupy a place in the fierce market, which has certain benchmarking. And, in order to ensure that the selected sample can represent the retail industry, our study excludes the retail enterprises with the proportion of retail main business below 60%, in order to make the sample more representative and more relevant to the purpose of the study. Second, is the homogeneity of decision units. Based on this requirement of DEA model, our study mainly considers information from the external environment. In order to ensure that the selected samples are comparable in the operating and regulatory environment and meet the requirement of homogeneity, the samples selected in our study are retail enterprises listed in Shenzhen and Shanghai, to avoid the influence of different accounting standards or years. Third, the data are objective, comprehensive and available. In this regard, listed enterprises also have an unparalleled advantage.

Based on the above principles of enterprise sample selection, our study screens the sample pool of A-share listed retail enterprises in Shanghai and Shenzhen, discarding those listed after 2011 because the study period was 2011–2021, and eliminating those enterprises with major asset restructuring and missing main variables during the reporting period, and then filtering through the main business share of retail, deleting those listed retail enterprises with a main business share below 60%. Then, the final research sample of our study was obtained. The year of channel strategy implementation and related information of the sample retail listed companies are shown in Table 1. The time of implementing channel strategy and the notes of integration strategy of the sample are extracted from the annual reports of the enterprises.

**Table 1.** Basic Retail Listed Companies.

| | Company Code | Company Name | Adding Online Channel Time | Implementation of Integration Strategy Time | Remarks on Integration Strategies |
|---|---|---|---|---|---|
| 1 | 002024 | Suning.com | Before 2011 | 2013 | 2013 |
| 2 | 600712 | Nanning Department Store | 2011 | 2019 | Completion of the opening of online operations on takeaway platforms for all stores in 2019. |
| 3 | 601116 | Three Rivers Shopping Mall | 2013 | 2015 | 2015 Partnership with Jingdong to Home |
| 4 | 002264 | Xinhuadu Shopping Mall | 20216 | 2017 | End of 2017 |
| 5 | 600814 | Hangzhou Xiebai | Not listed | Not listed | Not listed |
| 6 | 002561 | Xujiahui | 2014 | 2016 | 2016 Omnichannel, including online ordering and store pickup |
| 7 | 600828 | Maoye Commercial | 2019 | Not listed | Not listed |
| 8 | 002419 | Tianhong Stock | Before 2011 | 2014 | 2014 |
| 9 | 600774 | Hanshang Group | 2020 | Not listed | Not listed |
| 10 | 600693 | Dongbai Group | 2018 | 2018 | Not listed |
| 11 | 600729 | Chongqing Department Store | 2014 | 2018 | 2018 Pilot Mission, Jingdong to Home |
| 12 | 600723 | Shoushang shares | 2015 | Not listed | Not listed |
| 13 | 600361 | Hualian Supermarket | 2017 | 2017 | Partnerships with third-party takeaway platforms in 2017 |
| 14 | 601010 | Wenfeng shares | 2014 | 2018 | 2018 Amoy Fresh, continues to strengthen its partnership with Meituan Hungry |
| 15 | 601933 | Wing Fai Supermarket | 2014 | 2014 | September 2014 online order and payment, store pickup; October same-city delivery |
| 16 | 600697 | Eurasian Group | 2012 | 2016 | Eurasia to Home goes live in 2016 |
| 17 | 000715 | Zhongxing Commercial | 2013 | 2018 | 2018 ZTE Cloud Shopping |
| 18 | 000419 | Tongcheng Holdings | 2012 | 2016 | 2016 Department Store Division to implement the "Shopping without Hassle, Home Delivery" campaign |
| 19 | 600785 | Xinhua Department Store | 2015 | 2019 | 2019 CCpark Store |
| 20 | 002187 | Guangbai shares | 2013 | 2016 | 2016 Jingdong to Home |
| 21 | 002277 | AUA shares | 2013 | 2017 | In 2017, we started the 020 model of "offline experience + community + online micro store APP"; in 2018, we developed Youa Shopping" app synchronized with physical stores |
| 22 | 600857 | Ningbo Ciba | 2016 | Not listed | Not listed |
| 23 | 000417 | Hefei Department Store | 2012 | 2016 | 2016 Same City Delivery, Mobile App |
| 24 | 000759 | Zhongbai Group | 2013 | 2017 | 2017 |
| 25 | 600858 | Ginza shares | 2014 | 2015 | 2015 Supermarket Flash Sale |
| 26 | 002251 | BBK | 2014 | 2018 | April 2018 |
| 27 | 600859 | Wangfujing | 2013 | 2020 | 2020 |
| 28 | 600861 | Beijing urban and rural areas | 2011 | 2017 | 2017 Community Supermarket Department 118 Supermarket Logistics Centralized Distribution |
| 29 | 600628 | New world | 2012 | 2013 | 2013 VW, order online and pick up offline. |

Note: The data are from the end of 2012 to the end of 2021. Data source: Guotaian database and author's compilation based on annual statements of each company.

The integration strategy of the enterprises studied focused on process integration, which is manifested in real life as online store purchase, store pick-up or store delivery, and the timing of integration is judged to be the point when the key words first appear in each enterprise's annual report. From the above table, it can be seen that, before 2019, except Ningbo Zhongbai, Nanning Department Store, Xinhua Department Store and Wangfujing, the rest of the dual-channel enterprises implemented the corresponding channel integration strategy.

### 4.3. Selection of Variables and Data Description

#### 4.3.1. Input–Output Indicators

Since the inputs and outputs of retail enterprises are complex, scholars tend to select input–output indicators according to the general principles of quantity, importance and accessibility according to the purpose of the study. In order to ensure the scientific integrity of the indicators, scholars consider the three aspects of human, financial and material in the selection of input variables. The output indicators are usually considered from a combination of enterprise revenue, operational efficiency, capital utilization and other aspects. In the past, the most common output indicators in retail efficiency research were financial indicators such as revenue and profit, while some scholars set the output indicators as indicators of intangible assets of enterprises, including product and service satisfaction. The specific input–output indicators in our study are shown in the following table (Table 2).

**Table 2.** Indicator setting for input–output indicators.

|  | Name of Indicator | Description of Data | Indicators Selection References |
|---|---|---|---|
| Input indicators | Staff | Number of employees, representing the labor input of the enterprise | Lei [24], Wang et al. [42] |
|  | Cost of main operations | Cost of main business expenses, reflecting the structure of the main costs of the enterprise | Huang et al. [20] |
|  | Inputs to management costs | Amount of management costs, inputs related to corporate management operations | Lei [24], Wang et al. [42] |
|  | Cost of sales | Amount of selling expenses, inputs to business activities of enterprises selling | Huang et al. [20] |
|  | Number of stores | Total number of stores, retail significant fixed asset investment | Hean et al. [43], Zhao et al. [44] |
| Output indicators | Income from main business | Revenue from main operations, reflecting the profitability of the business | Huang et al. [20] |
|  | Gross margin of main business | According to the formula "(revenue from main business−cost from main business)/revenue from main business", the important profitability indicators of retail enterprises | Zhao et al. [44] |

#### 4.3.2. Descriptive Statistics for Input–Output Indicators

Since most companies implemented dual-channel strategies in 2013 and 2014, and explored channel convergence one after another from 2015 to 2017, eight years of data from 2012 to 2021 were chosen for the measurement indicators in our study. The raw data for our study was obtained by searching the annual reports and notes to the statements of each listed retail company through the Giant Wave Consulting website (www.cnifo.com.cn (accessed on 2 June 2020)). The descriptive statistics for each specific indicator are shown in the following table (Table 3).

**Table 3.** Descriptive statistics for input–output indicators.

| | | 2012 | 2013 | 2014 | 2015 | 2016 | 2017 | 2018 | 2019 | 2020 | 2021 |
|---|---|---|---|---|---|---|---|---|---|---|---|
| | | Total number of employees (in persons) | | | | | | | | | |
| | Average value | 9859 | 10,127 | 10,661 | 11,141 | 11,125 | 11,175 | 11,586 | 12,090 | 12,358 | 12,679 |
| | Minimum value | 254 | 256 | 239 | 229 | 219 | 205 | 207 | 200 | 204 | 206 |
| | Maximum value | 51,365 | 57,561 | 73,085 | 75,179 | 70,440 | 84,931 | 92,047 | 110,778 | 109,657 | 111,362 |
| | Standard deviation | 12,106 | 12,895 | 14,878 | 15,386 | 14,643 | 16,506 | 18,103 | 21,274 | 21,974 | 22,031 |
| | | Cost of main operations (in millions of dollars) | | | | | | | | | |
| | Average value | 9093.58 | 9933.15 | 10,234.20 | 11,047.76 | 11,593.11 | 13,259.07 | 15,217.32 | 16,303.56 | 16,923.47 | 17,058.19 |
| | Minimum value | 655.14 | 725.27 | 671.60 | 691.32 | 641.53 | 671.12 | 709.91 | 652.84 | 678.39 | 694.86 |
| Input Indicators | Maximum value | 79,955.58 | 88,141.17 | 91,011.73 | 112,463.70 | 125,467.80 | 156,637.90 | 201,719.10 | 222,081.20 | 231,677.65 | 239,482.76 |
| | Standard deviation | 14,761.22 | 16,331.96 | 16,962.96 | 20,833.27 | 23,268.41 | 29,026.72 | 37,341.80 | 41,440.05 | 42,893.09 | 43,017.41 |
| | | Administrative expenses (in millions of dollars) | | | | | | | | | |
| | Average value | 389.24 | 446.37 | 472.80 | 535.17 | 539.02 | 613.94 | 763.80 | 764.76 | 755.94 | 739.62 |
| | Minimum value | 41.35 | 49.72 | 45.86 | 47.37 | 43.68 | 43.33 | 48.46 | 50.25 | 46.22 | 43.71 |
| | Maximum value | 2350.11 | 2805.67 | 3356.57 | 4291.48 | 3946.27 | 4864.05 | 7462.56 | 8212.64 | 8165.39 | 8124.08 |
| | Standard deviation | 447.07 | 526.38 | 621.60 | 782.14 | 736.23 | 913.67 | 1416.26 | 1505.01 | 1489.66 | 1463.80 |
| | | Cost of sales (in millions of dollars) | | | | | | | | | |
| | Average value | 1206.88 | 1367.65 | 1523.18 | 1720.05 | 1811.81 | 1991.26 | 2327.89 | 2690.79 | 2725.41 | 2913.05 |
| | Minimum value | 38.90 | 46.34 | 42.31 | 37.34 | 34.92 | 37.03 | 35.40 | 33.83 | 35.22 | 36.37 |
| | Maximum value | 11,810.94 | 12,739.71 | 14,105.02 | 16,644.68 | 17,451.42 | 20,635.78 | 26,066.68 | 33,532.02 | 34,177.91 | 34,806.47 |
| | Standard deviation | 2226.00 | 2433.21 | 2727.11 | 3218.02 | 3387.21 | 3988.80 | 5086.90 | 6488.69 | 6921.39 | 7134.95 |
| | | Total number of stores (in units) | | | | | | | | | |
| | Average value | 168 | 169 | 184 | 190 | 186 | 295 | 478 | 308 | 312 | 343 |
| | Minimum value | 1 | 1 | 1 | 1 | 1 | 1 | 1 | 1 | 1 | 1 |
| | Maximum value | 1705 | 1585 | 1696 | 1638 | 1510 | 3867 | 8881 | 3630 | 4519 | 4822 |
| | Standard deviation | 351 | 338 | 357 | 353 | 327 | 733 | 1638 | 705 | 882 | 831 |
| | | Revenue from main business (in millions of yuan) | | | | | | | | | |
| | Average | 9093.58 | 9933.15 | 10,234.20 | 11,047.76 | 11,593.11 | 13,259.07 | 15,217.32 | 16,303.56 | 16,953.29 | 16,837.71 |
| | Minimum | 655.14 | 725.27 | 671.60 | 691.32 | 641.53 | 671.12 | 709.91 | 652.84 | 679.48 | 692.36 |
| | Maximum | 79,955.58 | 88,141.17 | 91,011.73 | 112,463.70 | 125,467.80 | 156,637.90 | 201,719.10 | 222,081.20 | 229,056.38 | 234,617.59 |
| Output Indicators | Standard deviation | 14,761.22 | 16,331.96 | 16,962.96 | 20,833.27 | 23,268.41 | 29,026.72 | 37,341.80 | 41,440.05 | 42,371.87 | 44,298.53 |
| | | Gross margin | | | | | | | | | |
| | Average | 0.16 | 0.16 | 0.16 | 0.16 | 0.17 | 0.16 | 0.17 | 0.16 | 0.17 | 0.18 |
| | Minimum | 0.09 | 0.09 | 0.10 | 0.10 | 0.10 | 0.11 | 0.11 | 0.10 | 0.11 | 0.11 |
| | Maximum | 0.23 | 0.32 | 0.30 | 0.25 | 0.25 | 0.25 | 0.27 | 0.27 | 0.27 | 0.28 |
| | Standard deviation | 0.03 | 0.04 | 0.04 | 0.04 | 0.04 | 0.04 | 0.04 | 0.04 | 0.4 | 0.4 |

## 4.4. Efficiency Measurement Results and Analysis

Dynamic Analysis of the Integrated Efficiency of Listed Retail Enterprises in China

By 2021, only Hangzhou Xiebai, Maoye Commercial and Hanshang Group were among the 29 retail enterprises still maintaining their single-channel operation strategy, while the remaining 22 of the 26 dual-channel enterprises (not Ningbo Zhongbai, Nanning Department Store, Xinhua Department Store, and Wangfujing) had accordingly implemented the channel integration strategy of online–offline store delivery. Through Maxdea8's global covariate and Malmquist's efficiency measure on the panel data of 29 listed retail enterprises from 2012 to 2021, the enterprises' annual total factor productivity change index, MI, and its preliminary decomposition index technical efficiency change index (EC) and technical progress growth index (TC) can be obtained. On this basis, further decomposition by the decomposition method proposed by Zofio in 2007, four indicators of pure technical efficiency change (PEC) and scale efficiency change (SEC), pure technical change (PTC) and scale technical change (STC) can be obtained.

As shown in Table 4, the average change index value of MI in 2012–2021 for retail enterprises that maintained single-channel strategy operations was 1.0325, indicating that

the average total factor productivity of these three enterprises had slightly increased in 2021 compared to 2012. However, the average results of the companies implementing a dual-channel strategy and a channel convergence strategy are not satisfactory, which can indicate that the implementation of a new channel strategy puts higher requirements on the level of technology and scale management of retail companies, and most of the companies have not been able to achieve total factor productivity growth through the new channel strategy. From the data in the table, it can be seen that the average change index of MI of enterprises implementing a dual-channel strategy in 2012–2021 was 1.1721, which indicates that the total factor productivity of enterprises that implemented dual-channel strategy in 2021 had decreased by 1.2% compared to 2012, among which the average decrease in total factor productivity of enterprises implementing dual-channel strategy was smaller than that of retail enterprises implementing only a dual-channel strategy without implementing a convergence strategy. The average rate of decline for firms implementing a dual-channel strategy was smaller than the average rate of decline for retail firms implementing only a dual-channel strategy without a convergence strategy, indicating that the implementation of a channel convergence strategy helped mitigate the decline in total factor productivity. From the decomposition index of the MI index of listed retail enterprises, the average technical efficiency change value and its decomposition among retail enterprises that implemented a dual-channel strategy are higher than those that did not implement the fusion strategy, indicating that in the trend of general decline in the performance of dual-channel retail enterprises, the implementation of a channel fusion strategy can enhance the technical efficiency of enterprises by improving the pure technical efficiency and scale efficiency of enterprises. Therefore, retail enterprises implementing new channel strategies can also improve enterprise efficiency from these two aspects. In addition, it can be found that both single-channel enterprises and enterprises that have implemented a dual-channel strategy and channel convergence strategy have a lower technological progress index, which indicates that retail enterprises still need to continuously strengthen the innovation and application of new technologies in the process of operation, channel change, and transformation and upgrading.

**Table 4.** Malmquist productivity index and its decomposition for listed firms in the retail sector (2012–2021).

| | Company Name | PEC | SEC | PTC | STC | MI |
|---|---|---|---|---|---|---|
| | Companies implementing a dual-channel strategy | | | | | |
| | Suning.com, PRC e-commerce company | 1.0000 | 1.0000 | 1.0000 | 0.9895 | 0.9895 |
| | Three Rivers Shopping | 1.0000 | 1.0000 | 0.9989 | 0.9977 | 0.9964 |
| | Xinhua all over the world | 0.9949 | 0.9991 | 0.9955 | 0.9962 | 0.9862 |
| | Xujiahui neighborhood of Shanghai | 1.0000 | 1.0000 | 0.9970 | 0.9986 | 0.9956 |
| | Tianhong Stock | 1.0000 | 1.0000 | 1.0002 | 1.0000 | 1.0002 |
| | Dongbai Group | 1.0003 | 0.9986 | 0.9926 | 1.0005 | 0.9920 |
| | Chongqing Department Store | 0.9976 | 0.9916 | 0.9975 | 1.0016 | 0.9879 |
| | Shoushang shares | 0.9952 | 1.0048 | 0.9981 | 0.9995 | 0.9955 |
| | Hualian Supermarket | 0.9976 | 0.9999 | 1.0001 | 1.0001 | 0.9977 |
| Those imple- | Wenfeng shares | 1.0000 | 0.9977 | 0.9934 | 0.9996 | 0.9907 |
| mentation | Wing Fai Supermarket | 1.0044 | 0.9924 | 1.0006 | 0.9982 | 0.9953 |
| integration | Eurasian Group | 1.0000 | 0.9916 | 1.0000 | 0.9810 | 0.9724 |
| strategies | Zhongxing Commercial | 1.0000 | 1.0000 | 1.0000 | 1.0000 | 1.0000 |
| | Tongcheng Holdings | 1.0100 | 1.0013 | 0.9967 | 1.0040 | 1.0094 |
| | Guangbai shares | 1.0000 | 1.0000 | 1.0011 | 1.0000 | 1.0011 |
| | AUA share | 0.9958 | 0.9982 | 0.9972 | 0.9950 | 0.9861 |
| | Hefei Department Store | 1.0000 | 0.9982 | 0.9984 | 0.9872 | 0.9836 |
| | China Hundred Group | 0.9996 | 1.0024 | 0.9997 | 1.0023 | 1.0037 |
| | Ginza shares | 0.9932 | 1.0006 | 0.9981 | 1.0006 | 0.9924 |
| | BBK | 1.0046 | 0.9920 | 0.9917 | 1.0066 | 0.9935 |
| | Beijing urban and rural areas | 0.9959 | 0.9970 | 0.9871 | 1.0012 | 0.9809 |
| | new world | 1.0000 | 1.0000 | 1.0000 | 1.0000 | 1.0000 |

**Table 4.** *Cont.*

| | Company Name | PEC | SEC | PTC | STC | MI |
|---|---|---|---|---|---|---|
| | Average of enterprises implementing integration strategies. | 0.9995 | 0.9984 | 0.9975 | 0.9981 | 0.9932 |
| No integration strategy implemented | Nanning Department Store | 0.9974 | 1.0020 | 0.9909 | 1.0011 | 0.9890 |
| | Xinhua Department Store | 0.9949 | 0.9991 | 0.9955 | 0.9962 | 0.9862 |
| | Ningbo Ciba | 1.0000 | 1.0000 | 1.0000 | 1.0000 | 1.0000 |
| | Wangfujing neighborhood of central Beijing, famous for its many shops and restaurants | 1.0000 | 0.9922 | 1.0000 | 1.0003 | 0.9921 |
| | Average of enterprises not implementing integration strategies. | 0.9981 | 0.9983 | 0.9966 | 0.9994 | 0.9918 |
| | Average of dual-channel strategy firms | 0.9983 | 0.9994 | 1.0073 | 1.0983 | 1.1721 |
| | Single-channel strategy | | | | | |
| | Hangzhou Xiebai | 1.0038 | 1.0004 | 1.0060 | 0.9998 | 1.0106 |
| | Maoye Commercial | 1.0000 | 1.0000 | 1.0008 | 1.0000 | 1.0013 |
| | Hanshang Group | 1.0000 | 0.9849 | 0.9786 | 1.0353 | 0.9981 |
| | Average of single-channel strategy firms | 1.0013 | 0.9951 | 0.9951 | 1.0117 | 1.0325 |

## 5. Analysis of Channel Strategy Choice Affecting the Overall Efficiency of Retail Enterprises

### 5.1. Description of Variables and Descriptive Statistics

5.1.1. Description of Variable Selection and Data Sources

Since the efficiency values of the global covariates of the DEA Malmquist measures represent the variation of DMU over multiple consecutive periods and can be used in the analysis of panel data, our study uses them as the observed variables of the explanatory variable retailer's overall efficiency, with the choice of a dual-channel strategy and the implementation of an online store delivery or online store pickup online integration strategy as one of the independent variables of the two panel regressions, respectively. Due to the difficulty of quantifying some of the impact paths of channels on retail firms' efficiency sorted out above, there are problems of cross-influence and the availability of data. In our study, based on the previous literature, only five indicators in the four paths are selected as explanatory variables, and the five indicators are as follows: "Accounts receivable level", an indicator representing the degree of relationship between the firm and its customers, and "Channel marketing level", an indicator representing the degree of marketing, selected from the market perspective. From the perspective of channel resources investment, the indicator "Fixed assets investment intensity" is selected to represent the level of offline channel construction investment. From the perspective of channel technology investment, the indicator "Channel technology investment intensity" is selected. From the perspective of channel technology investment, the indicator "Channel technology investment intensity" is selected to represent the level of offline channel construction investment. From the perspective of channel operation and management, the indicator "Inventory turnover rate" is selected to represent the level of channel operation and management. The control variables in our study were selected on the basis of prior research, and three indicators were chosen: socio-economic development level, enterprise scale and asset–liability ratio. The regression equation between the overall efficiency of listed retail enterprises and channel strategy and related influencing factors is thus established. The specific variables were selected as shown in the following table (Table 5).

**Table 5.** Variable Settings.

| | | Variable NAME | Variable Definition | Variable Symbol |
|---|---|---|---|---|
| Explained variables | | Integrated retail efficiency indicators | | Efficiency |
| Explanatory variables | | Whether to choose a dual-channel strategy | | channel |
| | | Whether to choose a fusion strategy of online shop delivery or online shop pickup | | Integrated |
| | Channel Strategy-Efficiency Related Influencing Factors | Degree of business-customer relationship | Level of accounts receivable | receivables |
| | | Degree of channel marketing | Selling expense ratio | Marketing |
| | | Level of investment in offline channel building | Intensity of fixed asset investment | Fixed |
| | | Channel technology input intensity | Level of technical inputs | Technology |
| | | Channel Operations Management Level | Inventory turnover rate | Invent |
| Control variables | external environment | Level of socio-economic development | Total retail sales of social consumer goods | Retail |
| | internal factor | | gearing | Asset |
| | | | Business Size | scale |

### 5.1.2. Descriptive Statistics of Variables

In our study, the total retail sales of consumer goods were obtained from the 2012–2021 statistical yearbook, and the rest of the indicators were calculated from the annual reports of each sample enterprise and the notes to the statements. The descriptive statistics of the variables are shown in the following table (Table 6).

**Table 6.** Descriptive statistics of variables.

| Impact Variables | 2013 | 2014 | 2015 | 2016 | 2017 | 2018 | 2019 | 2020 | 2021 |
|---|---|---|---|---|---|---|---|---|---|
| *Level of receivables (receivables)* | | | | | | | | | |
| Average | 0.0065 | 0.0061 | 0.0059 | 0.0075 | 0.0102 | 0.0122 | 0.0146 | 0.0138 | 0.0142 |
| Minimum | 0.0000 | 0.0000 | 0.0001 | 0.0003 | 0.0006 | 0.0004 | 0.0004 | 0.0004 | 0.0004 |
| Maximum | 0.0329 | 0.0344 | 0.0381 | 0.0316 | 0.0501 | 0.0732 | 0.1584 | 0.1682 | 0.1893 |
| Standard deviation | 0.0068 | 0.0069 | 0.0072 | 0.0079 | 0.0115 | 0.0152 | 0.0288 | 0.0317 | 0.0396 |
| *Selling expense ratio (Marketing)* | | | | | | | | | |
| Average | 0.0885 | 0.0966 | 0.0992 | 0.1014 | 0.0993 | 0.1039 | 0.1081 | 0.1143 | 0.1375 |
| Minimum | 0.0205 | 0.0220 | 0.0213 | 0.0194 | 0.0212 | 0.0210 | 0.0193 | 0.0204 | 0.0254 |
| Maximum | 0.1697 | 0.1793 | 0.1874 | 0.1992 | 0.1843 | 0.1895 | 0.2074 | 0.2676 | 0.2865 |
| Standard deviation | 0.0476 | 0.0503 | 0.0525 | 0.0525 | 0.0513 | 0.0516 | 0.0546 | 0.0569 | 0.0583 |
| *Fixed asset input intensity (Fixed)* | | | | | | | | | |
| Average | 0.2708 | 0.2758 | 0.2734 | 0.2818 | 0.2797 | 0.2752 | 0.2671 | 0.2890 | 0.2946 |
| Minimum | 0.0677 | 0.0536 | 0.0538 | 0.0603 | 0.0749 | 0.0662 | 0.0752 | 0.0815 | 0.0947 |
| Maximum | 0.7292 | 0.7269 | 0.7000 | 0.6633 | 0.6608 | 0.6490 | 0.6400 | 0.6695 | 0.6946 |
| Standard deviation | 0.1593 | 0.1604 | 0.1656 | 0.1635 | 0.1528 | 0.1562 | 0.1610 | 0.1677 | 0.1745 |
| *Level of technological input (Technology)* | | | | | | | | | |
| Average | 0.00126 | 0.00216 | 0.00221 | 0.00183 | 0.00170 | 0.00140 | 0.00148 | 0.00168 | 0.00186 |
| Minimum | 0.00000 | 0.00000 | 0.00000 | 0.00000 | 0.00000 | 0.00001 | 0.00002 | 0.00002 | 0.00002 |
| Maximum | 0.00515 | 0.01507 | 0.01489 | 0.01105 | 0.01014 | 0.00637 | 0.00680 | 0.00695 | 0.00734 |
| Standard deviation | 0.00155 | 0.00318 | 0.00333 | 0.00231 | 0.00218 | 0.00158 | 0.00178 | 0.00186 | 0.00189 |
| *Inventory turnover rate (Invent)* | | | | | | | | | |
| Average | 18.64 | 16.38 | 16.64 | 15.95 | 17.18 | 18.91 | 19.79 | 19.92 | 20.17 |
| Minimum | 1.82 | 0.80 | 0.46 | 0.72 | 0.89 | 0.84 | 1.19 | 1.25 | 1.38 |
| Maximum | 106.86 | 68.32 | 64.10 | 64.38 | 78.08 | 95.06 | 116.85 | 134.98 | 150.73 |
| Standard deviation | 21.83 | 17.54 | 16.72 | 16.94 | 19.98 | 23.93 | 28.70 | 32.86 | 37.83 |

**Table 6.** *Cont.*

| Impact Variables | 2013 | 2014 | 2015 | 2016 | 2017 | 2018 | 2019 | 2020 | 2021 |
|---|---|---|---|---|---|---|---|---|---|
| Gearing ratio (Asset) | | | | | | | | | |
| Average | 0.5652 | 0.5514 | 0.5414 | 0.5279 | 0.5299 | 0.5210 | 0.5197 | 0.5158 | 0.5093 |
| Minimum | 0.1998 | 0.1871 | 0.1752 | 0.1865 | 0.1809 | 0.1666 | 0.1534 | 0.1521 | 0.1437 |
| Maximum | 0.9408 | 0.9401 | 0.9356 | 0.9315 | 0.9316 | 0.9298 | 0.9205 | 0.9201 | 0.9188 |
| Impact variables | 2013 | 2014 | 2015 | 2016 | 2017 | 2018 | 2019 | 2020 | 2021 |
| Standard deviation | 0.1489 | 0.1545 | 0.1664 | 0.1689 | 0.1677 | 0.1781 | 0.1760 | 0.1794 | 0.1821 |
| Enterprise size (scale) | | | | | | | | | |
| Average | 8.617 | 8.653 | 8.750 | 8.860 | 8.891 | 8.966 | 9.017 | 9.344 | 9.575 |
| Minimum | 6.607 | 6.631 | 6.656 | 6.712 | 6.714 | 6.715 | 7.229 | 7.486 | 7.492 |
| Maximum | 11.327 | 11.317 | 11.386 | 11.829 | 11.966 | 12.203 | 12.375 | 12.464 | 12.581 |
| (statistics) Standard deviation | 0.886 | 0.903 | 0.923 | 1.018 | 1.051 | 1.092 | 1.103 | 1.179 | 1.221 |
| Total retail sales of consumer goods (Retail, in billions of yuan) | | | | | | | | | |
| Average | 8157.45 | 9106.07 | 10,069.10 | 11,084.04 | 12,161.93 | 13,134.52 | 14,043.80 | 14,468.65 | 14,679.18 |
| Minimum | 668.51 | 737.18 | 789.60 | 850.10 | 930.40 | 1330.10 | 1399.40 | 1473.45 | 1498.22 |
| Maximum | 25,453.93 | 28,471.15 | 31,517.60 | 34,739.10 | 38,200.10 | 39,767.10 | 42,951.80 | 42,977.35 | 43,176.64 |
| Standard deviation | 4271.67 | 4767.73 | 5312.20 | 5861.13 | 6441.16 | 6739.47 | 7298.13 | 7375.57 | 7396.13 |

*5.2. Modeling Method Selection and Model Construction*

5.2.1. Panel Data Regression Model

The following content focusses on the impact of selected factors on the overall efficiency of retail firms using econometric analysis. Given that the data in our study are panel data, a panel data model is used for the econometric analysis.

Let the dependent variable yit and the k × 1-dimensional independent variable $X_{it} = (X_{1,it}, X_{2,it}, \cdots, X_{k,it})'$ that satisfies the following linear relationship:

$$y_{it} = a_{it} + x'_{it}\beta_{it} + u_{it}, i = 1, 2, \cdots, N, t = 1, 2, \cdots, T \tag{4}$$

where N denotes the number of samples, T is the sample observation period quantity, $\beta_{it}$ is the coefficient matrix of the independent variables, and k denotes the number of independent variables. $a_{it}, u_{it}$ are the constant term and random error term of the model, respectively.

Considering a panel data model containing N cross-sectional samples from cross-sectional samples, Equation (5) can be simplified as:

$$y_i = a_i e + x_i \beta_i + u_i, e = (1, 1, \cdots 1)', i = 1, 2, \cdots, N \tag{5}$$

where yi, xi denote the T × 1 and T × k-dimensional matrices of variables, respectively. Equation (6) is expanded into matrix form as follows:

$$\begin{pmatrix} y_1 \\ y_2 \\ \vdots \\ y_N \end{pmatrix} = \begin{pmatrix} a_1 e \\ a_2 e \\ \vdots \\ a_N e \end{pmatrix} + \begin{pmatrix} x_1 & 0 & \cdots & 0 \\ 0 & x_2 & \cdots & \vdots \\ \vdots & \vdots & \ddots & 0 \\ 0 & \cdots & 0 & x_N \end{pmatrix} \begin{pmatrix} \beta_1 \\ \beta_2 \\ \vdots \\ \beta_N \end{pmatrix} + \begin{pmatrix} u_1 \\ u_1 \\ \vdots \\ u_N \end{pmatrix} \tag{6}$$

Depending on the data needed to choose a different panel data model for analysis, the mixed estimation model, fixed effects model or random effects model [45,46] is usually chosen. Equation (4) is a mixed estimation model, which is characterized by the slope term and constant term being the same for any cross-section or individual. The remaining two models consider the estimation results to differ in the slope term and constant term, and the difference between the two models is in the judgment of the relationship between the error term and the explanatory variables, which is considered relevant by the fixed effects model and uncorrelated by the random effects model. Since our study hopes that the results of the

sample selected will be applicable to the whole retail industry, the random effects model is initially considered more applicable. In the specific selection, our study discriminates by F-test, LM test and Hausman test. Where the F-test and LM test are used to judge the choice of the fixed effects model and mixed estimation model, and the Hausman test is used to judge the choice of the fixed effects model and random effects model.

### 5.2.2. Modeling Method Selection

A discrete restricted dependent variable model should be used when the dependent variable is a discrete restricted variable. A discrete restricted variable is one in which the observations of the natural number or integer measure are incomplete, i.e., do not reflect the full reality of the situation [47]. In this case, if OLS is used for modeling analysis, bias will occur, and usually the overall sample characteristics need to be inferred from the restricted local sample, often choosing truncated regression models such as tobit models. Since the dependent variable in our study takes the global reference efficiency value, its variation is restricted, so we chose to establish a subsumption tobit model for empirical research. The tobit model was first proposed by the American economist Tobin in the study of demand for consumer durables. Assuming that the dependent variable is $y_i$, the minimum level is $y_0$, and the independent variable is xi, the tobit model is constructed as follows:

$$
\begin{aligned}
y_i &= \begin{cases} y_t^*, & y_t^* > 0 \\ 0, & y_t^* \leq 0 \end{cases} \\
y_t^* &= \beta^T x_i + e_i, e_i \sim N\left(0, \sigma^2\right)
\end{aligned}
\tag{7}
$$

where xi is a $(k + 1)$-dimensional vector, and $\beta^T$ is the $(k + 1)$-dimensional parameter vector, and $y_t^*$ is the vector of efficiency values, which is estimated by the method of great likelihood estimation, with the likelihood function

$$
\begin{aligned}
L &= \prod\nolimits_{y_i=0}(1 - F_i) \prod\nolimits_{y_i>0} \frac{1}{\sqrt{2\pi}} \exp\left[-\frac{1}{2\sigma^2}\left(y_t - \beta^T x_i\right)^2\right] \\
F_i &= \int_{-\infty}^{\beta^T x_i/\sigma} \frac{1}{\sqrt{2\pi}} \exp\left[-\frac{t^2}{2}\right] dt
\end{aligned}
\tag{8}
$$

If $y_t^* > \sigma$, then take yi $\geq$ 0, at which point $y_i$ takes the actual observation; if $y_t^* \leq 0$, then $y_i$ = 0. When the dependent variable is constrained both left and right, the model is shown below:

$$
\begin{aligned}
y_i &= \begin{cases} C_t^-, & y_t^* \leq C_t^- \\ y_t^*, & C_t^- < y_t^* \leq C_t^+ \\ C_t^+, & y_t^* > C_t^+ \end{cases} \\
y_t^* &= \beta^T x_i + e_i, e_i \sim N\left(0, \sigma^2\right)
\end{aligned}
\tag{9}
$$

### 5.2.3. Construction of the Econometric Model

The tobit panel regression model, constructed by incorporating the above explanatory variables, explanatory variables and control variables, will be represented in the equation by the above characters of the variable symbols corresponding to the variables in order to make the equation expression more concise. The specific equations are as follows:

1.　Whether to choose a dual-channel strategy (channel) affects the efficiency model of retail firms

$$
Ef1_{it} = \alpha_0 + \alpha_1 CH_{it} + \alpha_2 Rec_{it} + \alpha_3 Ma_{it} + \alpha_4 Fi_{it} + \alpha_5 Te_{it} + \alpha_6 Inv_{it} + \alpha_7 Ret_{it} + \alpha_8 As_{it} + \alpha_9 Sc_{it} + u_{it}
\tag{10}
$$

where Ef1 is the composite efficiency of 29 listed retail enterprises from 2012 to 2021, taking values from the DEA Malmquist index calculated by Maxdea8 for the global reference; Ch is a dummy variable for whether to choose a dual-channel strategy, 1 for dual channel and 0 for single channel; Rec represents the level of accounts receivable; Mar represents the selling

expense ratio; Fi represents the fixed asset input level; Inv represents inventory turnover rate; Ret represents total retail sales of consumer goods; As represents gearing ratio; Sc represents firm size; and $u_{it}$ is a random error term that follows a normal distribution. i and t refer to the firm and its year, respectively.

2.　Channel convergence strategy affects the retailer efficiency model

$$Ef2_{it} = \beta_0 + \beta_1 Int_{it} + \beta_2 Rec_{it} + \beta_3 Ma_{it} + \beta_4 Fi_{it} + \beta_5 Te_{it} + \beta_6 In_{it} + \beta_7 Ret_{it} + \beta_8 As_{it} + \beta_9 Sc_{it} + u_{it} \qquad (11)$$

where Ef2 is the DEA Malmquist index for the global reference of 26 listed retail enterprises from 2012 to 2021, after excluding three enterprises, namely Hangzhou Xiebai, Maoye Commercial, and Hanshang Group, for enterprises that have implemented a dual-channel strategy in 2018 and before; In is a dummy variable for whether to choose the channel integration strategy of store delivery to home, with 1 for implementation and 0 otherwise; the meaning of the remaining variables is the same as in the previous model. i and t refer to the firm and its year, respectively.

*5.3. Model Regression Results and Analysis*

5.3.1. Model Regression Results

Our study performs a regression analysis with the help of Stata13 software. The relationship between the choice of a dual-channel strategy and the related influencing factors and the overall efficiency of retail firms is empirically demonstrated through model (10). Model (10) LR likelihood ratio test result is chibar2(01) = 380.47; Prob>chibar2 = 0.0000 significantly rejects the mixed regression approach. The Hausman test result is chi2(9) = (b − B)′[(V_b − V_B)^(−1)](b − B) = 8.29, Prob>chi2 = 0.5362 (greater than 0.05); the original hypothesis is rejected and a random effects model should be used.

The relationship between the implementation of channel integration strategies and related influencing factors and the overall efficiency of retail firms is empirically demonstrated through model (11). Model (11) LR likelihood ratio test results are:

chibar2(01) = 301.63, Prob>chibar2 = 0.0000, significantly rejecting the mixed regression approach, followed by the Hausman test, which was

chi2(9) = (b − B)′[(V_b − V_B)^(−1)](b − B) = 12.56, Prob>chi2 = 0.1834 (greater than 0.05), the original hypothesis is rejected and the random effect model should be used. Therefore, our study establishes a random effects model for regression analysis of the combined retail efficiency and each influencing factor, and Table 7 gives the estimated values of the coefficients of the explanatory variables obtained based on Models (10) and (11).

The results in Table 7 can be obtained from the Model 10 panel regression, and our study also conducted stepwise regression and found that the direction and significance of the impact variables did not change significantly, indicating that the regression results are robust. From the regression results, it has been determined whether implementing a dual-channel strategy (channel) has a positive but insignificant. Inventory turnover (invent) has a significant positive effect on the overall efficiency effect on the overall efficiency of retail enterprises of the firm at the 5% level; fixed asset investment intensity (fixed), technology investment level (technology) and firm size (scale) have a significant negative effect on the overall efficiency of the firm at the 10% level. In contrast, total retail sales of the consumer goods (retail), accounts receivable level (receivables), sales expense ratio (marketing) and gearing ratio (asset) have no significant effect on the overall efficiency of retail firms.

The results in Table 7 were obtained from the Model 11 panel regression, and our study conducted another stepwise regression to find that the direction and significance of the influencing variables did not change significantly, indicating that the regression results are robust. From the regression results, it has been determined whether implementing a channel integration strategy (integrated) has a significant positive effect on the overall efficiency of retail enterprises at the 10% level. Total retail sales of consumer goods (retail) and inventory turnover (invent) significantly and positively affect the overall efficiency of the firm at the 10% level; the receivables level (receivables), fixed asset investment intensity

(fixed) and firm size (scale) significantly and negatively affect the overall efficiency of the firm at the 5%, 1% and 1% levels, respectively. In contrast, the cost of goods sold (marketing), technology input level (technology) and gearing (asset) have no significant effect on the overall efficiency of retail firms.

**Table 7.** Tobit random effects panel regression results for the effect of channel strategy on the combined retail efficiency.

| Models 10 | | Models 11 | |
|---|---|---|---|
| Independent variables | Regression results | Independent variables | Regression results |
| Ch | 0.0043 (0.67) | In | 0.0120 * (1.84) |
| Rec | −0.2447 (−1.04) | Rec | −0.5847 ** (−2.10) |
| Mar | −0.1084 (−0.93) | Mar | −0.1009 (−0.69) |
| Fi | −0.0674 * (−1.90) | Fi | −0.2138 *** (−4.30) |
| Te | −1.8435 * (−1.71) | Te | −0.8612 (−0.64) |
| Models 10 | | Models 11 | |
| Independent variables | Regression results | Independent variables | Regression results |
| Inv | 0.0012 ** (2.26) | Inv | 0.0008 * (1.92) |
| Ret | $1.10 \times 10^{-6}$ (1.06) | Ret | $2.74 \times 10^{-6}$ * (1.91) |
| As | −0.0006 (−0.02) | As | −0.0603 (−1.44) |
| Sc | −0.0129 * (−1.75) | Sc | −0.0363 *** (−3.00) |
| P0 | 1.0992 *** (16.95) | P0 | 1.3648 *** (11.87) |

Note: ***, **, * denote estimated coefficients significant at the 1%, 5%, and 10% levels, respectively, and the z-statistic is in parentheses. All coefficients are calculated and tested with the help of Stata13.

### 5.3.2. Analysis of Model Regression Results

From the empirical results, it can be seen that the implementation of a dual-channel strategy (channel) has no significant effect on the overall efficiency of retail enterprises, while the implementation of a channel integration strategy (integrated) has a significant positive effect on the overall efficiency of retail enterprises. In the process of implementing a channel strategy, the total retail sales of consumer goods (retail) and inventory turnover (invent) have a positive impact on the overall efficiency of the firm; the level of accounts receivable (receivables), fixed assets investment intensity (fixed), technology investment level (technology) and firm size (scale) negatively affect the overall efficiency of the firm. There is no significant effect of sales expense ratio (marketing) and gearing ratio (asset) on the overall efficiency of retail firms.

The coefficient of whether to implement a dual-channel strategy (channel) is positive but insignificant, the significance is different from the previous literature, probably because of the long time span of the research sample selected for our study. It also indicates that the implementation of a dual-channel strategy may have a significant positive impact on retail efficiency in the short term, as confirmed by the previous literature; however, in the long term, the implementation of a dual-channel strategy did not keep the efficiency of all retail firms growing. From the specific analysis of retail enterprises' overall efficiency in the section above, it can be seen that the efficiency of enterprises that implemented a dual-channel strategy generally increased between 2012 and 2015, but after 2015, more and more retail enterprises ventured into dual-channel operation with unsatisfactory results. The reason for this is that most retail enterprises implement a dual-channel strategy mostly via a kind of passive choice and blindly follow the trend under the market push, and they do not really know how to operate online channels. Moreover, most retailers believe that the service cost of additional online channels is lower than that of traditional brick-and-mortar stores, and that the digital business will be profitable when it grows to a certain scale. However, there is no positive correlation between dual-channel and multi-channel operations and return on revenue. When laying out online channel operations, companies

should fully consider the return on investment and develop appropriate channel strategies so that sales growth can truly lead to increased corporate profitability.

Whether or not to implement the channel integration strategy of online order placement and store delivery (integrated) is a new indicator introduced in our study, and the empirical results find that this channel strategy positively affects the efficiency of retail firms. This indicates that the implementation of a channel integration strategy helps to improve the overall efficiency of retail enterprises, mainly because this strategy eliminates customers' concerns about the quality, style and model of goods when shopping online, and if they are not satisfied with the goods, they can choose a physical store nearby to enjoy after-sales service. And, customers picking up or returning goods to the store is also a disguised online to offline diversion that can prevent channel cannibalization.

A negative coefficient on the level of accounts receivable (receivables) indicates that an increase in the level of accounts receivable of retail firms has a negative effect on the overall efficiency of retail firms. In the early stage, retail enterprises can promote sales by selling products on credit, expanding their market share, improving their relationship with customers and increasing their revenue, thus improving their overall efficiency. However, if the enterprise has a large amount of accounts receivable, it not only occupies the enterprise capital, but also has the risk of being uncollectible. And, with the continuous upgrading of consumer demand, consumers are more concerned about the quality of products, and only using credit sales means businesses can no longer achieve the role of attracting consumers to achieve the improvement of business efficiency.

The negative coefficients of fixed asset investment intensity (fixed) and enterprise size (scale) indicate that increasing the fixed asset investment and enterprise size of retail enterprises will have a negative impact on the overall efficiency of enterprises, which is consistent with the results of previous studies. At the present stage, retail enterprises are in full competition and most of them are expanding their business scale, but the scale expansion of retail enterprises does not effectively bring into play the economies of scale, but leads to chaotic corporate structure and inefficient management, resulting in the lower overall efficiency of enterprises. The excessive investment in fixed assets will further increase the cost of depreciation and maintenance and reduce the realizable capital of the enterprise, which will adversely affect the profit and solvency of the enterprise. Therefore, retail enterprises should start to streamline the size of the enterprise to reduce the negative impact of diseconomies of scale on the efficiency of the enterprise.

The positive coefficient of inventory turnover (invent) is also consistent with the results of previous studies, indicating that retail enterprises can promote the overall efficiency of the enterprise by increasing the inventory turnover. This is because the higher the inventory turnover rate, the more liquid the inventory is, and at the same time, the capital occupied by the inventory and the related management storage costs will be reduced, which also reflects the high capital utilization rate and the relatively strong selling ability of the enterprise, thus positively affecting the efficiency of the enterprise.

The negative direction of the influence of technology input level (technology) on enterprise efficiency is mainly due to the restriction of index selection; most of the retail enterprises do not have a separate account of R&D expenditure, so it is impossible to study the influence of enterprise R&D investment on enterprise efficiency. While the value of technology input index selected in our study is the technology input in intangible assets, and this is a part of technology input, it also includes the technology input of enterprise online construction. The technology input is also included in the online construction of the enterprise, but it is related more to the input of the enterprise information software. For enterprises implementing new channel strategy, technology investment is only the cost of third-party information software, which is far from promoting the efficiency of enterprises.

The positive coefficient of total retail sales of consumer goods (retail) indicates that the higher the level of consumption, the higher the overall efficiency of retail enterprises in the geographical area. The two variables of sales expense ratio and gearing ratio are not significant in both models, where the impact of sales expense ratio on enterprise efficiency

is not significant, indicating that the current investment of enterprises in advertising and marketing, promotion expenses, etc., does not play a significant role in promoting enterprise efficiency.

## 6. Discussion

Retail is the basis of commodity circulation, and the operation condition of the retail industry is directly related to the speed and efficiency of commodity circulation; retail is the key carrier to guide production and expand consumption, and the development condition of the retail industry is related to market prosperity and employment security. Based on the current development situation of retail enterprises in China, this paper summarizes the path of influence of retail channel strategy on enterprise efficiency from four perspectives: market demand, channel resources, channel technology and channel system. Furthermore, it empirically analyzes the influence of a retail channel strategy on enterprise comprehensive efficiency, which helps guide retail enterprises to adjust the channel development strategy, promoting the transformation and upgrading of retail industry, releasing development vitality and enhancing development momentum.

Compared with other industries, the implementation of a retail channel strategy is closely related to the successful transformation and upgrading of enterprises. With the development of technology, online channel strategies of different natures such as internet platforms and mobile terminals have provided retail enterprises with new development solutions over time. However, as the demographic dividend disappears and the drawbacks of online channels begin to emerge, the retail industry has started to experiment with various new channel practices. This has also led to academic discussions and studies on the relationship between the implementation of retail channel strategies and the efficiency of retail enterprises. However, there is a lack of literature on the relationship between the two, and most of the existing literature is only a description and summary of the symptoms, and further research is needed. Therefore, the theoretical significance of this paper is to further reveal the inner mechanism of the influence of retail channel strategy on enterprise efficiency according to the four elements of channel formation and evolution. The study of retail enterprise efficiency from the perspective of channel strategy selection not only enriches the study of the relationship between channel development and retail enterprise efficiency, but also provides retail enterprises with channel strategy guidance ideas, which has certain theoretical significance.

This paper finds that the implementation of a dual-channel strategy may have a positive impact on the overall efficiency of the firm in the early stage, but does not have a significant impact in the long term. Implementing a channel convergence strategy on this basis will have a positive impact on the overall efficiency of the firm. Given that most of the retailers have already opened dual-channel strategies, online channels do allow retailers with a large number of local stores to break through technological bottlenecks and improve overall efficiency. Therefore, this paper encourages domestic and international retailers implementing dual-channel strategies to implement corresponding channel integration measures after adopting online channel operations.

Based on the above findings, this paper makes the following suggestions:

1. Focus on demand to promote efficiency. At this stage, retail enterprises need to recognize a problem. New channels or the implementation of various channel strategies should be based around the customer and insight into the changes in customer demand, to achieve accurate service. This requires enterprises in all aspects of channel operations to be customer-centric with online and offline resource allocation, with a channel to capture customers and seize customers. From the perspective of actual development, a channel integration strategy refers to the integration of online stores, physical stores and modern logistics, so that the traditional retailing by people of goods and fields breaks through the limitations of time and space, bringing higher value returns for enterprises. However, in fact, many retail companies that have implemented convergence strategies have not seen significant growth in performance,

and even a decline. The reason for this situation is that this transformation has not produced the expected effect on the improvement of the viscosity of existing users and the attraction of new ones. So, to break the diversion dilemma, companies need independent innovation in the general implementation of the "retail + internet" logic to add a content section to result in "retail + internet + content". In the future, the mass marketing strategy with low marketing efficiency will gradually disappear, and mass marketing will be gradually transformed into niche marketing. Enterprises should first clarify the target customer groups and their needs, and then carry out precise marketing to them, so that customers can get a more personalized shopping experience. In the channel integration mode, all channels of the enterprise can be used to help customers solve their problems. Each channel has advantages and disadvantages, and retail enterprises must give full play to the advantages of each channel by using and integrating existing resources to achieve mutual diversion of online and offline traffic, so that online traffic and offline traffic can be shared. In this way, enterprises can fully understand customer information and carry out accurate marketing to lay a good foundation.

2. Good placement of resources to increase efficiency. Facing the impact of e-commerce and consumers' diversified channel demand, many retail enterprises are trying to operate multiple channels and actively implement relevant channel integration strategies. However, the rapid development of online sales does not drive all enterprises to improve efficiency. The reason for this is that many traditional retailers are developing online channels as a passive choice and blindly following the trend forced by the market, and they do not really understand how to operate online channels. At the same time, retailers often have two misconceptions about online sales: one is that the service cost of online channels is lower than that of physical stores; the other is that digital business growth to a certain scale will bring profitability. However, "more" is not the same as profitability, and there is no positive correlation between multi-channel operations and return on revenue. When companies lay out multi-channel operations, they should fully consider financial costs such as return on investment and develop appropriate channel strategies so that sales growth can really bring about an increase in profitability. With the maturity of the overall business ecology of the internet, the expansion of the online shopping scale has become an irreversible trend; but, in the online retail boom, enterprises need to think cold. Although the addition of online channels allows companies to gain objective sales, the growth of digital sales does not necessarily lead to increased profitability, and may even threaten the overall effectiveness of the company. For example, for companies implementing dual-channel and channel convergence strategies, their supply chain and IT costs are much higher than the costs of previous offline channels. Therefore, retailers should realize that the key to building a multi-channel retail model is to look at the whole picture, find suitable partners from the perspective of a higher-level retail ecosystem, focus on improving their own core competitiveness and attract more consumers through quality experiences.

3. Enhance the strong effect of technology. Leveraging the "internet+" to improve information technology capabilities and achieve integrated development in the COVID-19 era. In the study of the path of retail enterprises to influence the efficiency of enterprises through the introduction of technology, it is found that most retail enterprises' existing technology investment is limited to the investment in information software. With the in-depth development of the "internet+", retail enterprises should take advantage of the "internet+", play the role of internet, coordinate the mutually exclusive influence of different channels, amplify the complementary advantages of channels and realize multi-channel integration. To achieve integrated development by leveraging the "internet+", the key lies in the following two points: First, actively introduce online channels and implement a dual-channel strategy, which should pay attention to the scale efficiency of enterprises and improve the operation process from the

global system level to bring into play the synergy effect of multiple channels. Second, the introduction of the internet and other related advanced technologies should be based on the strategic guidance of the enterprise, and it should be introduced in a target-oriented and targeted manner under the condition that the enterprise fully understands its own strengths and weaknesses, rather than blindly following the trend of development and imitating other enterprises, resulting in a decline in enterprise efficiency. With the help of big data, data analysis capabilities should be improved and retail transformation realized. There are only a few retail enterprises using big data to operate in China, and the application of big data is still in the exploration stage. However, the advantages of big data are obvious to all, and enterprises can use big data to analyze consumer behavior information and improve category management and marketing strategies. In the era of big data, it is necessary to analyze the data. And, to leverage big data to achieve transformation, retail enterprises should first broaden the depth and breadth of the application of big data, and access to data and analysis of all parties, on the basis of data analysis to speed up the operation of enterprises in logistics, supply chain and other links, in the enterprise to make decisions to provide accurate more for information. Secondly, enterprises should focus on the development of big data products. In the process of upgrading, a platform-based enterprise can also be used to build a platform ecosystem with the application of big data, to provide help for the partners in the platform and achieve common development.

4.  Strategic management of superior efficiency. The ultimate goal of enterprises is to obtain profits, so when retail enterprises carry out the strategic layout of channels, they should start from the perspective of improving the return on investment, looking at the whole picture, and letting multi-channel retailing serve the overall development of enterprises. From the current point of view, most domestic retail enterprises, in the implementation of a multi-channel operation strategy, do not establish this global concept, and it is more likely to just be a blind behavior. To truly tap into the value of multi-channel retailing, Chinese retailers must find the best time and the most suitable investment area for multi-channel layout from the strategic perspective of long-term development, so that multi-channel operation can become a great help for enterprises to break through the offline sales dilemma and achieve benign and rapid development. The fundamental support for multi-channel retailing and even omni-channel retailing is still the offline capability of retail enterprises. Therefore, the key to the sustainable development and long-term profitability of multi-channel retailing model is for retail enterprises to return to their original mindset and continuously enhance their offline capabilities. For example, in the empirical evidence, we found that the inventory turnover rate of retail enterprises has a positive impact on the overall efficiency of enterprises, so enterprises can consider optimizing their management techniques from aspects such as merchandise planning capability, purchasing and transportation capability, ordering and replenishment capability, etc., to improve the turnaround speed and further improve the overall efficiency of the enterprise. With the integration of different channels such as offline, PC and mobile, the decentralized nature of channels will also play a key role in retail inventory optimization, and the multi-channel shared inventory strategy will also effectively improve the speed of product delivery and reduce logistics costs.

This paper researches the impact of channel strategy selection on the overall efficiency of listed retail enterprises and achieves some research results, but there are still some shortcomings to be improved upon in the specific research process. Due to the availability of data, all of the listed retail enterprises in this paper were selected within the sample period, and considering the requirement of the homogeneity of decision units in the DEA model, the retail enterprises listed outside Shenzhen and Shanghai and those whose main business of retailing was less than 60% were removed from this paper, making the final sample size smaller. Besides, future studies may optimize or use other efficiency measurement models to expand the sample scope for research. Further research on refined channel strategies

can be conducted in the future. This study focuses more on the impact of a dual-channel strategy and convergence strategy on the efficiency of enterprises, but no in-depth empirical analysis has been conducted on the specific ways of implementing a dual-channel strategy and convergence strategy (self-built or cooperative). Therefore, future research can explore the specific ways of implementing channel strategies for enterprises and propose clearer development directions for management decisions.

## 7. Conclusions

In our study, based on combing the prior literature, we constructed the path of channel strategy selection on enterprise efficiency to provide a theoretical basis for the model of channel strategy selection affecting retail enterprise efficiency. In this paper, the design of the input–output indicators of retail enterprise efficiency is often lacking in the design of the industry-specific characteristics. Therefore, this paper sets input–output indicators of retail firm efficiency based on the characteristics of retail firms, with reference to the prior research. In terms of introducing new dummy variables, according to the literature review and the analysis of the current situation of retail channel development, it is clear that retail enterprises generally implement a dual-channel strategy and channel integration strategy at present. Therefore, in this paper, in order to empirically prove the impact of channel strategy on retail enterprise efficiency, two models were constructed to introduce two dummy variables concerning whether to implement a dual-channel strategy and whether to implement a channel convergence strategy, respectively. The impact of the above two channel strategies on the comprehensive retail efficiency was empirically proven by combining the quantifiable related indicators in the impact path proposed in this paper, so as to provide suggestions for how domestic and foreign retail enterprises can maintain their own development and future sustainability in the era of the pandemic. This paper provides suggestions on how domestic and international retail enterprises can maintain their own development and future sustainability in the COVID-19 era.

At present, a dual-channel strategy and channel convergence strategy are commonly implemented in the retail industry, and our study sought to investigate whether the current channel strategy plays a positive role in retail integrated efficiency. To this end, based on the theoretical and research review of domestic and foreign scholars, our study first proposed the impact path of channel strategy on retail enterprise efficiency, sorted out the development of channel practice, then measured and analyzed the comprehensive efficiency of listed retail enterprises from 2012 to 2021. Finally, it empirically demonstrated the impact of the implementation of a dual-channel strategy and channel convergence strategy on the comprehensive efficiency of retail enterprises. In summary, the main findings of our study are the following three points:

The implementation of channel strategy will reintegrate the information flow, capital flow, commercial flow and logistics of the enterprise, which will change the efficiency of the enterprise by affecting the market demand, channel resources, channel technology and channel system. Accordingly, our study proposed the path framework of channel strategy on the comprehensive efficiency of retail enterprises in Section 3. On the path that channel strategy affects enterprise efficiency by changing the market demand, our study analyzed that the new channel strategy will increase the enterprise market demand, positively affecting enterprise efficiency. This is followed by the implementation of channel integration strategy, which can provide more accurate service for customers, so that consumer satisfaction, loyalty and consumption frequency increase, positively affecting enterprise efficiency, while the substitution and negative spillover between channels will bring the channel disadvantages, negatively affecting the efficiency of the firm. On the path where channel strategy affects firm efficiency by expanding channel resources, our study analyzed that the addition of new channels or the huge costs invested in channel integration will negatively affect firm efficiency over time, but the implementation of relationship channel integration strategy with suppliers will help transfer channel rights to retailers. On the path where channel strategy affects firm efficiency through the introduction of

channel technology, our study analyzed that the implementation of channel strategy helps to introduce new technology and thus has a positive impact on firm efficiency. On the path of the channel strategy optimizing the channel system and affecting enterprise efficiency, our study analyzed that the implementation of a channel strategy helps to improve the enterprise management ability of enterprises and thus improves enterprise efficiency.

Our study evaluated and analyzed the efficiency of the sample retail listed enterprises by selecting the total number of employees, retail main business cost, administrative expense, selling expense and total number of stores as the input indicators, and selecting retail main business revenue and gross margin as the output indicators to construct input–output indicators for the sample retail listed enterprises with respect to the characteristics of the retail industry. From the dynamic analysis, it was found that the average total factor productivity of enterprises implementing a channel integration strategy among dual-channel operation enterprises was higher than that of enterprises not implementing an integration strategy.

The results of the empirical study found that the dual-channel strategy positively but insignificantly affects the overall efficiency of retail firms and the channel convergence strategy significantly and positively affects the overall retail efficiency. The variables selected on the influence path of market demand have a negative but non-significant effect on the degree of channel marketing and a significant negative effect on the level of accounts receivable, which indicates that today's simple credit sales activities no longer attract consumers. The variables selected on the influence path of channel resources have a significant negative effect on the level of investment in offline channel construction, mainly because retail firms are generally in a situation of poor economies of scale; the variable channel technology investment intensity selected in the path of influence has a negative and insignificant effect. The variable channel operations management level selected in the path of influence channel system has a significant positive effect; this is because the higher the inventory turnover rate, the more liquid the inventory is and the higher the capital utilization of the enterprise.

**Author Contributions:** Conceptualization and methodology, C.Y.; software, N.C.; validation and formal analysis, N.C.; investigation and resources, N.C.; data curation, N.C.; writing—original draft preparation, N.C.; writing—review and editing, J.W.; visualization, N.C.; supervision, C.Y.; project administration, C.Y.; funding acquisition, C.Y. All authors have read and agreed to the published version of the manuscript.

**Funding:** This research was funded by "the National Social Science Foundation of China", grant number (19FJYB043).

**Institutional Review Board Statement:** Not applicable.

**Informed Consent Statement:** Not applicable.

**Data Availability Statement:** The processed data required to reproduce these findings cannot be shared at this time as the data also forms part of an ongoing study.

**Conflicts of Interest:** The authors declare no conflict of interest.

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
