# Peer review of "Research on the Sustainability of Channel Strategy Selection on the Overall Efficiency of Listed Retail Enterprises: Evidence from China"

_sustainability, doi:10.3390/su14169992_

Round 1
Reviewer 1 Report
The paper entitled Research on the impact of channel strategy selection on the overall efficiency of listed retail enterprises: Empirical evidence from China deals with the actual topic. Suggestions for paper improvement are below:
· *The first problem that arises in abstract is some kind of contradiction:
“In the post-COVID-19 time, China's retail industry has accelerated the formation of a new pattern of development with dual domestic and international cycles promoting each other. “
“This article uses the data of 29 Chinese listed retail enterprises with retail main business accounting for more than 60% from 2012 to 2019 as samples. “
· *The most significant years were not considered. How can the post-covid period be analyzed with only pre-covid information? The observed period is questionable, and also all results.
· * In the introduction, the last paraph is a brief description of each section (some kind of short methodology).
· * The separate section Practical and theoretical implications (or Discussion) is missing.
· * Conclusion section is not on a satisfactory level. It is not written in a scientific manner. Clearly state your unique research contributions in the conclusion section. The authors need to clearly provide several solid future research directions (this confirms a bad relationship with the gaps in the literature).
· *Limitations of this research should be provided in the last section.
· *The paper must be better connected to previous research
· *The Scientific and practical contributions must be explicitly stated.
· *Technical problems:
o Avoid bullets numbering in the abstract;
o References are not in defined style (Capital letters, Abbreviated journal names, and everything else, see template)
o Align the manuscript with instructions for authors
o References are double numbered
o Tables are confusing and not arranged (for example Table 3 (complete table), Table 4 (second column)).
Suggested references:
Andrejić M., Kilibarda, M. Pajić, V., (2021). Measuring efficiency change in time applying malmquist productivity index: a case of distribution centres in Serbia. FACTA UNIVERSITATIS, Series Mechanical Engineering, 19 (3), 499-514.
Qin, W.; Qi, X. Evaluation of Green Logistics Efficiency in Northwest China. Sustainability 2022, 14, 6848. https://doi.org/10.3390/su14116848
Zhao, W.; Qiu, Y.; Lu, W.; Yuan, P. Input–Output Efficiency of Chinese Power Generation Enterprises and Its Improvement Direction-Based on Three-Stage DEA Model. Sustainability 2022, 14, 7421. https://doi.org/10.3390/su14127421
Author Response
Response to Reviewer 1 Comments
Dear reviewer:
Hello, thank you very much for your careful review of our paper in your busy schedule and for your constructive suggestions. We have carefully studied your valuable suggestions and comments, and have revised and improved the paper according to your suggestions and comments (the modified part of the paper is marked out with the "Track Changes" function in Word). The specific responses are as follows:
Point 1: The paper entitled Research on the impact of channel strategy selection on the overall efficiency of listed retail enterprises:Empirical evidence from China deals with the actual topic. Suggestions for paper improvement are below:
1.The first problem that arises in abstract is some kind of contradiction:”In the post-COVID-19 time, China's retail industry has accelerated the formation of a new pattern of development with dual domestic and international cycles promoting each other. “
“This article uses the data of 29 Chinese listed retail enterprises with retail main business accounting for more than 60% from 2012 to 2019 as samples.”
The most significant years were not considered. How can the post-covid period be analyzed with only pre-covid information? The observed period is questionable, and also all results.
Response 1: Thank you very much for this valuable suggestion. We completely agree with your suggestion. We apologize for the lack of consideration in our writing.As recommended, we extended the observation period from 2012-2019 to 2012-2021, to better examine the channel selection options for retailers in the era of the COVID-19.However, because of the completeness of the study time and the availability of data, we were unable to update the data to 2022.On the bright side, the data updated to 2021 still contributes much to the reliability of this paper.We have rewritten the discussion section and some of the conclusions based on the updated data, and in the discussion section we offer suggestions for retail businesses to grow in today's epidemic. Here is the updated data and discussion section. The discussion section is in lines 29-53 of page 21,lines 1-54 of page 22,lines 1-54 of page 23 and lines 1-18 of page 24 in the Word's " Track Changes " model.
Specific modifications:
Table 3. Descriptive statistics for input-output indicators
|
|
2012 |
2013 |
2014 |
2015 |
2016 |
2017 |
2018 |
2019 |
2020 |
2021 |
|||||||
|
Input Indicators
|
Total number of employees (in persons) |
|
|
||||||||||||||
|
average value |
9859 |
10127 |
10661 |
11141 |
11125 |
11175 |
11586 |
12090 |
12358 |
12679 |
|||||||
|
minimum value |
254 |
256 |
239 |
229 |
219 |
205 |
207 |
200 |
204 |
206 |
|||||||
|
maximum value |
51365 |
57561 |
73085 |
75179 |
70440 |
84931 |
92047 |
110778 |
109657 |
111362 |
|||||||
|
standard deviation |
12106 |
12895 |
14878 |
15386 |
14643 |
16506 |
18103 |
21274 |
21974 |
22031 |
|||||||
|
Cost of main operations (in millions of dollars) |
|
|
|||||||||||||||
|
average value |
9093.58 |
9933.15 |
10234.20 |
11047.76 |
11593.11 |
13259.07 |
15217.32 |
16303.56 |
16923.47 |
17058.19 |
|||||||
|
minimum value |
655.14 |
725.27 |
671.60 |
691.32 |
641.53 |
671.12 |
709.91 |
652.84 |
678.39 |
694.86 |
|||||||
|
maximum value |
79955.58 |
88141.17 |
91011.73 |
112463.70 |
125467.80 |
156637.90 |
201719.10 |
222081.20 |
231677.65 |
239482.76 |
|||||||
|
standard deviation |
14761.22 |
16331.96 |
16962.96 |
20833.27 |
23268.41 |
29026.72 |
37341.80 |
41440.05 |
42893.09 |
43017.41 |
|||||||
|
Administrative expenses (in millions of dollars) |
|
|
|||||||||||||||
|
average value |
389.24 |
446.37 |
472.80 |
535.17 |
539.02 |
613.94 |
763.80 |
764.76 |
755.94 |
739.62 |
|||||||
|
minimum value |
41.35 |
49.72 |
45.86 |
47.37 |
43.68 |
43.33 |
48.46 |
50.25 |
46.22 |
43.71 |
|||||||
|
maximum value |
2350.11 |
2805.67 |
3356.57 |
4291.48 |
3946.27 |
4864.05 |
7462.56 |
8212.64 |
8165.39 |
8124.08 |
|||||||
|
standard deviation |
447.07 |
526.38 |
621.60 |
782.14 |
736.23 |
913.67 |
1416.26 |
1505.01 |
1489.66 |
1463.80 |
|||||||
|
Cost of sales (in millions of dollars) |
|
|
|||||||||||||||
|
average value |
1206.88 |
1367.65 |
1523.18 |
1720.05 |
1811.81 |
1991.26 |
2327.89 |
2690.79 |
2725.41 |
2913.05 |
|||||||
|
minimum value |
38.90 |
46.34 |
42.31 |
37.34 |
34.92 |
37.03 |
35.40 |
33.83 |
35.22 |
36.37 |
|||||||
|
maximum value |
11810.94 |
12739.71 |
14105.02 |
16644.68 |
17451.42 |
20635.78 |
26066.68 |
33532.02 |
34177.91 |
34806.47 |
|||||||
|
standard deviation |
2226.00 |
2433.21 |
2727.11 |
3218.02 |
3387.21 |
3988.80 |
5086.90 |
6488.69 |
6921.39 |
7134.95 |
|||||||
|
Total number of stores (in units) |
|
|
|||||||||||||||
|
average value |
168 |
169 |
184 |
190 |
186 |
295 |
478 |
308 |
312 |
343 |
|||||||
|
minimum value |
1 |
1 |
1 |
1 |
1 |
1 |
1 |
1 |
1 |
1 |
|||||||
|
maximum value |
1705 |
1585 |
1696 |
1638 |
1510 |
3867 |
8881 |
3630 |
4519 |
4822 |
|||||||
|
standard deviation |
351 |
338 |
357 |
353 |
327 |
733 |
1638 |
705 |
882 |
831 |
|||||||
|
Output Indicators |
Revenue from main business (in millions of yuan) |
|
|
||||||||||||||
|
average |
9093.58 |
9933.15 |
10234.20 |
11047.76 |
11593.11 |
13259.07 |
15217.32 |
16303.56 |
16953.29 |
16837.71 |
|||||||
|
minimum |
655.14 |
725.27 |
671.60 |
691.32 |
641.53 |
671.12 |
709.91 |
652.84 |
679.48 |
692.36 |
|||||||
|
maximum |
79955.58 |
88141.17 |
91011.73 |
112463.70 |
125467.80 |
156637.90 |
201719.10 |
222081.20 |
229056.38 |
234617.59 |
|||||||
|
standard deviation |
14761.22 |
16331.96 |
16962.96 |
20833.27 |
23268.41 |
29026.72 |
37341.80 |
41440.05 |
42371.87 |
44298.53 |
|||||||
|
Gross margin |
|
|
|||||||||||||||
|
average |
0.16 |
0.16 |
0.16 |
0.16 |
0.17 |
0.16 |
0.17 |
0.16 |
0.17 |
0.18 |
|||||||
|
minimum |
0.09 |
0.09 |
0.10 |
0.10 |
0.10 |
0.11 |
0.11 |
0.10 |
0.11 |
0.11 |
|||||||
|
maximum |
0.23 |
0.32 |
0.30 |
0.25 |
0.25 |
0.25 |
0.27 |
0.27 |
0.27 |
0.28 |
|||||||
|
standard deviation |
0.03 |
0.04 |
0.04 |
0.04 |
0.04 |
0.04 |
0.04 |
0.04 |
0.4 |
0.4 |
|||||||
Table 4. Malmquist Productivity Index and its decomposition for listed firms in the retail sector (2012-2021)
|
|
Company Name |
PEC |
SEC |
PTC |
STC |
MI |
|
Implementing a dual-channel strategy companies |
||||||
|
Of which: implementation of integration strategies |
Suning.com, PRC e-commerce company |
1.0000 |
1.0000 |
1.0000 |
0.9895 |
0.9895 |
|
Three Rivers Shopping |
1.0000 |
1.0000 |
0.9989 |
0.9977 |
0.9964 |
|
|
Xinhua all over the world |
0.9949 |
0.9991 |
0.9955 |
0.9962 |
0.9862 |
|
|
Xujiahui neighborhood of Shanghai |
1.0000 |
1.0000 |
0.9970 |
0.9986 |
0.9956 |
|
|
Tianhong Stock |
1.0000 |
1.0000 |
1.0002 |
1.0000 |
1.0002 |
|
|
Dongbai Group |
1.0003 |
0.9986 |
0.9926 |
1.0005 |
0.9920 |
|
|
Chongqing Department Store |
0.9976 |
0.9916 |
0.9975 |
1.0016 |
0.9879 |
|
|
Shoushang shares |
0.9952 |
1.0048 |
0.9981 |
0.9995 |
0.9955 |
|
|
Hualian Supermarket |
0.9976 |
0.9999 |
1.0001 |
1.0001 |
0.9977 |
|
|
Wenfeng shares |
1.0000 |
0.9977 |
0.9934 |
0.9996 |
0.9907 |
|
|
Wing Fai Supermarket |
1.0044 |
0.9924 |
1.0006 |
0.9982 |
0.9953 |
|
|
Eurasian Group |
1.0000 |
0.9916 |
1.0000 |
0.9810 |
0.9724 |
|
|
Zhongxing Commercial |
1.0000 |
1.0000 |
1.0000 |
1.0000 |
1.0000 |
|
|
Tongcheng Holdings |
1.0100 |
1.0013 |
0.9967 |
1.0040 |
1.0094 |
|
|
Guangbai shares |
1.0000 |
1.0000 |
1.0011 |
1.0000 |
1.0011 |
|
|
AUA share |
0.9958 |
0.9982 |
0.9972 |
0.9950 |
0.9861 |
|
|
Hefei Department Store |
1.0000 |
0.9982 |
0.9984 |
0.9872 |
0.9836 |
|
|
China Hundred Group |
0.9996 |
1.0024 |
0.9997 |
1.0023 |
1.0037 |
|
|
Ginza shares |
0.9932 |
1.0006 |
0.9981 |
1.0006 |
0.9924 |
|
|
BBK |
1.0046 |
0.9920 |
0.9917 |
1.0066 |
0.9935 |
|
|
Beijing urban and rural areas |
0.9959 |
0.9970 |
0.9871 |
1.0012 |
0.9809 |
|
|
new world |
1.0000 |
1.0000 |
1.0000 |
1.0000 |
1.0000 |
|
|
Average of enterprises implementing integration strategies. |
0.9995 |
0.9984 |
0.9975 |
0.9981 |
0.9932 |
|
|
No integration strategy implemented |
Nanning Department Store |
0.9974 |
1.0020 |
0.9909 |
1.0011 |
0.9890 |
|
Xinhua Department Store |
0.9949 |
0.9991 |
0.9955 |
0.9962 |
0.9862 |
|
|
Ningbo Ciba |
1.0000 |
1.0000 |
1.0000 |
1.0000 |
1.0000 |
|
|
Wangfujing neighborhood of central Beijing, famous for its many shops and restaurants |
1.0000 |
0.9922 |
1.0000 |
1.0003 |
0.9921 |
|
|
Average of enterprises not implementing integration strategies. |
0.9981 |
0.9983 |
0.9966 |
0.9994 |
0.9918 |
|
|
Average of dual-channel strategy firms |
0.9983 |
0.9994 |
1.0073 |
1.0983 |
1.1721 |
|
|
Single-channel strategy |
||||||
|
|
Hangzhou Xiebai |
1.0038 |
1.0004 |
1.0060 |
0.9998 |
1.0106 |
|
Maoye Commercial |
1.0000 |
1.0000 |
1.0008 |
1.0000 |
1.0013 |
|
|
Hanshang Group |
1.0000 |
0.9849 |
0.9786 |
1.0353 |
0.9981 |
|
|
Average of single-channel strategy firms |
1.0013 |
0.9951 |
0.9951 |
1.0117 |
1.0325 |
|
Table 6. Descriptive statistics of variables
|
Impact variables |
2013 |
2014 |
2015 |
2016 |
2017 |
2018 |
2019 |
2020 |
2021 |
|
Level of receivables (receivables) |
|
|
|||||||
|
average |
0.0065 |
0.0061 |
0.0059 |
0.0075 |
0.0102 |
0.0122 |
0.0146 |
0.0138 |
0.0142 |
|
minimum |
0.0000 |
0.0000 |
0.0001 |
0.0003 |
0.0006 |
0.0004 |
0.0004 |
0.0004 |
0.0004 |
|
maximum |
0.0329 |
0.0344 |
0.0381 |
0.0316 |
0.0501 |
0.0732 |
0.1584 |
0.1682 |
0.1893 |
|
standard deviation |
0.0068 |
0.0069 |
0.0072 |
0.0079 |
0.0115 |
0.0152 |
0.0288 |
0.0317 |
0.0396 |
|
Selling expense ratio (marketing) |
|
|
|||||||
|
average |
0.0885 |
0.0966 |
0.0992 |
0.1014 |
0.0993 |
0.1039 |
0.1081 |
0.1143 |
0.1375 |
|
minimum |
0.0205 |
0.0220 |
0.0213 |
0.0194 |
0.0212 |
0.0210 |
0.0193 |
0.0204 |
0.0254 |
|
maximum |
0.1697 |
0.1793 |
0.1874 |
0.1992 |
0.1843 |
0.1895 |
0.2074 |
0.2676 |
0.2865 |
|
standard deviation |
0.0476 |
0.0503 |
0.0525 |
0.0525 |
0.0513 |
0.0516 |
0.0546 |
0.0569 |
0.0583 |
|
Fixed asset input intensity (fixed) |
|
|
|||||||
|
average |
0.2708 |
0.2758 |
0.2734 |
0.2818 |
0.2797 |
0.2752 |
0.2671 |
0.2890 |
0.2946 |
|
minimum |
0.0677 |
0.0536 |
0.0538 |
0.0603 |
0.0749 |
0.0662 |
0.0752 |
0.0815 |
0.0947 |
|
maximum |
0.7292 |
0.7269 |
0.7000 |
0.6633 |
0.6608 |
0.6490 |
0.6400 |
0.6695 |
0.6946 |
|
standard deviation |
0.1593 |
0.1604 |
0.1656 |
0.1635 |
0.1528 |
0.1562 |
0.1610 |
0.1677 |
0.1745 |
|
Level of technological input (technology) |
|
|
|||||||
|
average |
0.00126 |
0.00216 |
0.00221 |
0.00183 |
0.00170 |
0.00140 |
0.00148 |
0.00168 |
0.00186 |
|
minimum |
0.00000 |
0.00000 |
0.00000 |
0.00000 |
0.00000 |
0.00001 |
0.00002 |
0.00002 |
0.00002 |
|
maximum |
0.00515 |
0.01507 |
0.01489 |
0.01105 |
0.01014 |
0.00637 |
0.00680 |
0.00695 |
0.00734 |
|
standard deviation |
0.00155 |
0.00318 |
0.00333 |
0.00231 |
0.00218 |
0.00158 |
0.00178 |
0.00186 |
0.00189 |
|
Inventory turnover rate (Invent) |
|
|
|||||||
|
average |
18.64 |
16.38 |
16.64 |
15.95 |
17.18 |
18.91 |
19.79 |
19.92 |
20.17 |
|
minimum |
1.82 |
0.80 |
0.46 |
0.72 |
0.89 |
0.84 |
1.19 |
1.25 |
1.38 |
|
maximum |
106.86 |
68.32 |
64.10 |
64.38 |
78.08 |
95.06 |
116.85 |
134.98 |
150.73 |
|
standard deviation |
21.83 |
17.54 |
16.72 |
16.94 |
19.98 |
23.93 |
28.70 |
32.86 |
37.83 |
|
Gearing ratio (asset) |
|
|
|||||||
|
average |
0.5652 |
0.5514 |
0.5414 |
0.5279 |
0.5299 |
0.5210 |
0.5197 |
0.5158 |
0.5093 |
|
minimum |
0.1998 |
0.1871 |
0.1752 |
0.1865 |
0.1809 |
0.1666 |
0.1534 |
0.1521 |
0.1437 |
|
maximum |
0.9408 |
0.9401 |
0.9356 |
0.9315 |
0.9316 |
0.9298 |
0.9205 |
0.9201 |
0.9188 |
|
standard deviation |
0.1489 |
0.1545 |
0.1664 |
0.1689 |
0.1677 |
0.1781 |
0.1760 |
0.1794 |
0.1821 |
|
Enterprise size (scale) |
|
|
|||||||
|
average |
8.617 |
8.653 |
8.750 |
8.860 |
8.891 |
8.966 |
9.017 |
9.344 |
9.575 |
|
minimum |
6.607 |
6.631 |
6.656 |
6.712 |
6.714 |
6.715 |
7.229 |
7.486 |
7.492 |
|
Continued Table 6 |
|||||||||
|
Impact variables |
2013 |
2014 |
2015 |
2016 |
2017 |
2018 |
2019 |
2020 |
2021 |
|
maximum |
11.327 |
11.317 |
11.386 |
11.829 |
11.966 |
12.203 |
12.375 |
12.464 |
12.581 |
|
(statistics) standard deviation |
0.886 |
0.903 |
0.923 |
1.018 |
1.051 |
1.092 |
1.103 |
1.179 |
1.221 |
|
Total retail sales of consumer goods (retail, in billions of yuan) |
|
|
|||||||
|
average |
8157.45 |
9106.07 |
10069.10 |
11084.04 |
12161.93 |
13134.52 |
14043.80 |
14468.65 |
14679.18 |
|
minimum |
668.51 |
737.18 |
789.60 |
850.10 |
930.40 |
1330.10 |
1399.40 |
1473.45 |
1498.22 |
|
maximum |
25453.93 |
28471.15 |
31517.60 |
34739.10 |
38200.10 |
39767.10 |
42951.80 |
42977.35 |
43176.64 |
|
standard deviation |
4271.67 |
4767.73 |
5312.20 |
5861.13 |
6441.16 |
6739.47 |
7298.13 |
7375.57 |
7396.13 |
Discussion part:
Retail is the basis of commodity circulation, and the operation condition of retail industry is directly related to the speed and efficiency of commodity circulation; retail is the key carrier to guide production and expand consumption, and the development condition of retail industry is related to market prosperity and employment security. Based on the current development situation of retail enterprises in China, this paper summarizes the influence path of retail channel strategy on enterprise efficiency from four perspectives: market demand, channel resources, channel technology and channel system, and empirically analyzes the influence of retail channel strategy on enterprise comprehensive efficiency, which helps guide retail enterprises to adjust channel development strategy, promote the transformation and upgrading of retail industry, release development vitality and enhance development momentum.
Compared with other industries, the implementation of retail channel strategy is closely related to the successful transformation and upgrading of enterprises. With the development of technology, online channel strategies of different nature such as Internet platforms and mobile terminals have provided retail enterprises with new development solutions over time, but as the demographic dividend disappears and the drawbacks of online channels begin to emerge, the retail industry has started to experiment with various new channel practices. This has also led to academic discussions and studies on the relationship between the implementation of retail channel strategies and the efficiency of retail enterprises. However, there is a lack of literature on the relationship between the two, and most of the existing literature is only a description and summary of the symptoms, and further research is needed. Therefore, the theoretical significance of this paper is to further reveal the inner mechanism of the influence of retail channel strategy on enterprise efficiency according to the four elements of channel formation and evolution. The study of retail enterprise efficiency from the perspective of channel strategy selection not only enriches the study of the relationship between channel development and retail enterprise efficiency, but also provides retail enterprises with channel strategy guidance ideas, which has certain theoretical significance.
This paper finds that the implementation of a dual-channel strategy may have a positive impact on the overall efficiency of the firm in the early stage, but does not have a significant impact in the long term. Implementing a channel convergence strategy on this basis will have a positive impact on the overall efficiency of the firm. Given that most of the retailers have already opened dual-channel strategies, online channels do allow retailers with a large number of local stores to break through technological bottlenecks and improve overall efficiency. Therefore, this paper encourages domestic and international retailers implementing dual-channel strategies to implement corresponding channel integration measures after adopting online channel operations.
Based on the above findings, this paper makes the following suggestions:
(1)Focus on demand to promote efficiency. At this stage, retail enterprises need to recognize a problem, new channels or the implementation of various channel strategies should be around the customer, insight into the changes in customer demand, to achieve accurate service. This requires enterprises in all aspects of channel operations should be customer-centric online and offline resource allocation, with the channel to capture customers and seize customers. From the perspective of actual development, channel integration strategy refers to the integration of online stores, physical stores and modern logistics, so that the traditional retailing of people, goods and fields break through the limitations of time and space, bringing higher value returns for enterprises. However, in fact, many retail companies that have implemented convergence strategies have not seen significant growth in performance, and even a decline. The reason for this situation is that this transformation has not produced the expected effect on the improvement of the viscosity of existing users and the attraction of new ones. So to break the diversion dilemma, companies need to do is independent innovation, in the general implementation of the "retail + Internet" logic to add content section into the "retail + Internet + content". In the future, the mass marketing strategy with low marketing efficiency will gradually disappear, and mass marketing will be gradually transformed into niche marketing. Enterprises should first clarify the target customer groups and their needs, and then carry out precise marketing to them, so that customers can get a more ultimate shopping experience. In the channel integration mode, all channels of the enterprise can be used to help customers solve their problems. Each channel has advantages and disadvantages, and retail enterprises must give full play to the advantages of each channel, use and integrate existing resources to achieve mutual diversion of online and offline traffic, so that online traffic and offline traffic can be shared. In this way, enterprises can fully understand customer information and carry out accurate marketing to lay a good foundation.
(2)Good placement of resources to increase efficiency. Facing the impact of e-commerce and consumers' diversified channel demand, many retail enterprises are trying to operate multi-channel and actively implement relevant channel integration strategies, but the rapid development of online sales does not drive all enterprises to improve efficiency. The reason for this is that many traditional retailers are developing online channels as a passive choice and blindly following the trend forced by the market, and do not really understand how to operate online channels. At the same time, retailers often have two misconceptions about online sales: one is that the service cost of online channels is lower than that of physical stores; the other is that digital business growth to a certain scale will bring profitability. However, "more" is not the same as profitability, and there is no positive correlation between multi-channel operations and return on revenue. When companies lay out multi-channel operations, they should fully consider financial costs such as return on investment and develop appropriate channel strategies so that sales growth can really bring about an increase in profitability. With the maturity of the overall business ecology of the Internet, the expansion of online shopping scale has become an irreversible trend, but in the online retail boom, enterprises need to think cold. Although the addition of online channels allows companies to gain objective sales, the growth of digital sales does not necessarily lead to increased profitability, and may even threaten the overall effectiveness of the company. For example, for companies implementing dual-channel and channel convergence strategies, their supply chain and IT costs are much higher than the costs of previous offline channels. Therefore, retailers should realize that the key to building a multi-channel retail model is to look at the whole picture, find suitable partners from the perspective of a higher-level retail ecosystem, focus on improving their own core competitiveness, and attract more consumers through quality experiences.
(3)Enhance the strong effect of technology. Leveraging the "Internet+" to improve information technology capabilities and achieve integrated development in the COVID-19 era. In the study of the path of retail enterprises to influence the efficiency of enterprises through the introduction of technology, it is found that most retail enterprises' existing technology investment is limited to the investment in information software. With the in-depth development of "Internet+", retail enterprises should take advantage of "Internet+", play the role of Internet, coordinate the mutually exclusive influence of different channels, amplify the complementary advantages of channels, and realize multi-channel integration. To achieve integrated development by leveraging "Internet+", the key lies in the following two points. First, actively introduce online channels and implement dual-channel strategy, which should pay attention to the scale efficiency of enterprises and improve the operation process from the global system level to bring into play the synergy effect of multiple channels. Second, the introduction of the Internet and other related advanced technologies should be based on the strategic guidance of the enterprise, and should be introduced in a target-oriented and targeted manner under the condition that the enterprise fully understands its own strengths and weaknesses, rather than blindly following the trend of development and imitating other enterprises, resulting in a decline in enterprise efficiency. With the help of big data, improve data analysis capabilities and realize retail transformation. There are only a few retail enterprises using big data to operate in China, and the application of big data is still in the exploration and exploration stage. However, the advantages of big data are obvious to all, and enterprises can use big data to analyze consumer behavior information and improve category management and marketing strategies. In the era of big data, it is necessary to analyze the data. And to leverage big data to achieve transformation, retail enterprises should first broaden the depth and breadth of the application of big data, access to data and analysis of all parties, on the basis of data analysis to speed up the operation of enterprises in logistics, supply chain and other links, in the enterprise to make decisions to provide accurate more for information. Secondly, enterprises should focus on the development of big data products. In the process of upgrading to a platform-based enterprise can also be used to build a platform ecosystem with the application of big data, to provide help for the partners in the platform and achieve common development.
(4)Strategic management of superior efficiency. The ultimate goal of enterprises is to obtain profits, so when retail enterprises carry out the strategic layout of channels, they should start from the perspective of improving the return on investment, look at the whole picture, and let multi-channel retailing serve the overall development of enterprises. From the current point of view, most domestic retail enterprises in the implementation of multi-channel operation strategy does not establish this global concept, more just a blind behavior. To truly tap the value of multi-channel retailing, Chinese retailers must find the best time and the most suitable investment area for multi-channel layout from the strategic perspective of long-term development, so that multi-channel operation can become a great help for enterprises to break through the offline sales dilemma and achieve benign and rapid development. The fundamental support for multi-channel retailing and even omni-channel retailing is still the offline capability of retail enterprises. Therefore, the key to the sustainable development and long-term profitability of multi-channel retailing model is for retail enterprises to return to their original mindset and continuously enhance their offline capabilities. For example, in the empirical evidence, we found that the inventory turnover rate of retail enterprises has a positive impact on the overall efficiency of enterprises, so enterprises can consider optimizing their management techniques from aspects such as merchandise planning capability, purchasing and transportation capability, ordering and replenishment capability, etc. to improve the turnaround speed to further improve the overall efficiency of the enterprise. With the integration of different channels such as offline, PC and mobile, the decentralized nature of channels will also play a key role in retail inventory optimization, and the multi-channel shared inventory strategy will also effectively improve the speed of product delivery and reduce logistics costs.
Point 2: In the introduction, the last paragraph is a brief description of each section (some kind of short methodology).
Response 2: We found this comment is important and useful. As suggested, we have added a brief description of each section in the last paragraph of Introduction.They are on lines 46-55 of page 2 and 1-8 of page 3 in Word's " Track Changes " mode.
Specific modifications:
The study of this paper is divided into seven parts. The first part is the introduction. It introduces the research background of this paper in detail, presents the research topic, and explains the significance of this paper. The second part and the third part mainly comb through the literature related to retail channel strategy and retail firm efficiency, analyze the path and the inner mechanism of constructing retail channel strategy on firm efficiency, and provide methodological and theoretical references for the following empirical analysis and research. Part 4 and Part 5 will conduct the empirical analysis according to the common process of DEA efficiency research. In the fourth part, the efficiency of 29 Chinese listed retail companies is measured by applying the DEA method, and then classified and analyzed according to the different channel strategies implemented. In the fifth section, the impact of channel strategy on the overall efficiency of retailing is empirically demonstrated by using the global covariate efficiency value measured by DEA-Malmquist as the explanatory variable, introducing a dummy variable of whether channel strategy is implemented or not, and controlling for other relevant internal and external possible influencing factors. The sixth section presents the practical and theoretical implications and recommendations of this paper. Finally, the conclusion section presents conclusions to promote the overall efficiency improvement of listed retail firms in their future sustainable development.
Point 3: The separate section Practical and theoretical implications (or Discussion) is missing.
Response 3: We found this suggestion is crucial. For an article,practical and theoretical implications are very important. Therefore, we added a discussion section which including practical and theoretical implications. They are in lines 30-53 of page 21 and lines 1-4 of page 22 in Word's " Track Changes " mode.
Specific modifications:
Practical implications: Retail is the basis of commodity circulation, and the operation condition of retail industry is directly related to the speed and efficiency of commodity circulation; retail is the key carrier to guide production and expand consumption, and the development condition of retail industry is related to market prosperity and employment security. Based on the current development situation of retail enterprises in China, this paper summarizes the influence path of retail channel strategy on enterprise efficiency from four perspectives: market demand, channel resources, channel technology and channel system, and empirically analyzes the influence of retail channel strategy on enterprise comprehensive efficiency, which helps guide retail enterprises to adjust channel development strategy, promote the transformation and upgrading of retail industry, release development vitality and enhance development momentum.
Theoretical implications: Compared with other industries, the implementation of retail channel strategy is closely related to the successful transformation and upgrading of enterprises. With the development of technology, online channel strategies of different nature such as Internet platforms and mobile terminals have provided retail enterprises with new development solutions over time, but as the demographic dividend disappears and the drawbacks of online channels begin to emerge, the retail industry has started to experiment with various new channel practices. This has also led to academic discussions and studies on the relationship between the implementation of retail channel strategies and the efficiency of retail enterprises. However, there is a lack of literature on the relationship between the two, and most of the existing literature is only a description and summary of the symptoms, and further research is needed. Therefore, the theoretical significance of this paper is to further reveal the inner mechanism of the influence of retail channel strategy on enterprise efficiency according to the four elements of channel formation and evolution. The study of retail enterprise efficiency from the perspective of channel strategy selection not only enriches the study of the relationship between channel development and retail enterprise efficiency, but also provides retail enterprises with channel strategy guidance ideas, which has certain theoretical significance.
Point 4: Conclusion section is not on a satisfactory level. It is not written in a scientific manner. Clearly state your unique research contributions in the conclusion section. The authors need to clearly provide several solid future research directions (this confirms a bad relationship with the gaps in the literature).
Response 4: Thank you very much for this valuable suggestion. It is important to written article in a scientific manner. As suggested, we have rewritten conclusion section and stated our unique research contributions in the first paragraph of conclusion section.Also, we clearly provided several solid future research directions in the last paragraph of discussion section.They are in lines 19-53 of page 24, lines 1-54 of page 25 and lines 1-5 of page 26.
Specific modifications:
Discussion part(limitations and future research directions):
This paper researches the impact of channel strategy selection on the overall efficiency of listed retail enterprises and achieves some research results, but there are still some shortcomings to be improved in the specific research process. Due to the availability of data, all the listed retail enterprises in this paper were selected within the sample period, and considering the requirement of homogeneity of decision units in the DEA model, the retail enterprises listed outside Shenzhen and Shanghai and those whose main business of retailing is less than 60% were removed from this paper, making the final sample size smaller. Besides, future studies may optimize or use other efficiency measurement models to expand the sample scope for research. Further research on refined channel strategies can be conducted in the future. This study focuses more on the impact of dual-channel strategy and convergence strategy on the efficiency of enterprises, but no in-depth empirical analysis has been conducted on the specific ways of implementing dual-channel strategy and convergence strategy (self-built or cooperative). Therefore, future research can explore the specific ways of implementing channel strategies for enterprises and propose clearer development directions for management decisions.
Conclusions part:
In our study, based on combing the prior literature, we construct the path of channel strategy selection on enterprise efficiency to provide a theoretical basis for the model of channel strategy selection affecting retail enterprise efficiency. In this paper, the design of the input-output indicators of retail enterprise efficiency is often lacking in the design of the industry-specific characteristics. Therefore, this paper sets input-output indicators of retail firm efficiency based on the characteristics of retail firms with reference to the prior research. In terms of introducing new dummy variables, according to the literature review and the analysis of the current situation of retail channel development, it is clear that retail enterprises generally implement dual-channel strategy and channel integration strategy at present. Therefore, in this paper, in order to empirically prove the impact of channel strategy on retail enterprise efficiency, two models are constructed to introduce two dummy variables of whether to implement dual-channel strategy and whether to implement channel convergence strategy respectively, and the impact of the above two channel strategies on the comprehensive retail efficiency is empirically proven by combining the quantifiable related indicators in the impact path proposed in this paper, so as to provide suggestions for how domestic and foreign retail enterprises can maintain their own development and future sustainability in the era of epidemic. This paper provides suggestions on how domestic and international retailers can maintain their own development and future sustainability in the COVID-19 era.
At present, dual-channel strategy and channel convergence strategy are commonly implemented in the retail industry, and our study seeks to investigate whether the current channel strategy plays a positive role in retail integrated efficiency. To this end, based on the theoretical and research review of domestic and foreign scholars, our study first proposes the impact path of channel strategy on retail enterprise efficiency, sorts out the development of channel practice, then measures and analyzes the comprehensive efficiency of listed retail enterprises from 2012-2021, and finally empirically demonstrates the impact of the implementation of dual channel strategy and channel convergence strategy on the comprehensive efficiency of retail enterprises.In summary, the main findings of our study are the following three points.
The implementation of channel strategy will reintegrate the information flow, capital flow, commercial flow and logistics of the enterprise, which will change the efficiency of the enterprise by affecting the market demand, channel resources, channel technology and channel system, accordingly, our study proposes the path framework of channel strategy on the comprehensive efficiency of retail enterprises in Chapter 2. In the path that channel strategy affects enterprise efficiency by changing market demand, our study analyzes that the new channel strategy will make the enterprise market expand demand increase positively affect enterprise efficiency, followed by the implementation of channel integration strategy can provide more accurate service for customers, so that consumer satisfaction, loyalty increase, consumption frequency increase, positively affect enterprise efficiency, while the substitution and negative spillover between channels will bring the channel disadvantages negatively affect the efficiency of the firm. In the path where channel strategy affects firm efficiency by expanding channel resources, our study analyzes that the addition of new channels or the huge costs invested in channel integration will negatively affect firm efficiency over time, but the implementation of relationship channel integration strategy with suppliers will help transfer channel rights to retailers. In the path where channel strategy affects firm efficiency through the introduction of channel technology, our study analyzes that the implementation of channel strategy helps to introduce new technology and thus has a positive impact on firm efficiency. In the path of channel strategy optimizing channel system affecting enterprise efficiency, our study analyzes that the implementation of channel strategy helps to improve the enterprise management ability of enterprises and thus improves enterprise efficiency.
Our study evaluates and analyzes the efficiency of the sample retail listed enterprises by selecting the total number of employees, retail main business cost, administrative expense, selling expense, and total number of stores as input indicators, and selecting retail main business revenue and gross margin as output indicators to construct input-output indicators for the sample retail listed enterprises with respect to the characteristics of the retail industry. From the dynamic analysis, it is found that the average total factor productivity of enterprises implementing channel integration strategy among dual-channel operation enterprises is higher than that of enterprises not implementing integration strategy.
The results of the empirical study found that the dual-channel strategy positively but insignificantly affects the overall efficiency of retail firms and the channel convergence strategy significantly and positively affects the overall retail efficiency. The variables selected in the influence path of market demand have a negative but non-significant effect on the degree of channel marketing and a significant negative effect on the level of accounts receivable, which indicates that today's simple credit sales activities no longer attract consumers; the variables selected in the influence path of channel resources have a significant negative effect on the level of investment in offline channel construction, mainly because retail firms are generally in a situation of poor economies of scale; the channel technology influence The variable channel technology investment intensity selected in the path of influence has a negative and insignificant effect; the variable channel operations management level selected in the path of influence channel system has a significant positive effect, this is because the higher the inventory turnover rate, the more liquid the inventory is and the higher the capital utilization of the enterprise.
Point 5: Limitations of this research should be provided in the last section.
Response 5: Thank you very much for this useful suggestion. As suggested, we have added limitations of this research in the discussion section.They are in lines 19-34 of page 24.
Specific modifications:
Discussion part(limitations and future research directions):
This paper researches the impact of channel strategy selection on the overall efficiency of listed retail enterprises and achieves some research results, but there are still some shortcomings to be improved in the specific research process. Due to the availability of data, all the listed retail enterprises in this paper were selected within the sample period, and considering the requirement of homogeneity of decision units in the DEA model, the retail enterprises listed outside Shenzhen and Shanghai and those whose main business of retailing is less than 60% were removed from this paper, making the final sample size smaller. Besides, future studies may optimize or use other efficiency measurement models to expand the sample scope for research. Further research on refined channel strategies can be conducted in the future. This study focuses more on the impact of dual-channel strategy and convergence strategy on the efficiency of enterprises, but no in-depth empirical analysis has been conducted on the specific ways of implementing dual-channel strategy and convergence strategy (self-built or cooperative). Therefore, future research can explore the specific ways of implementing channel strategies for enterprises and propose clearer development directions for management decisions.
Point 6: The paper must be better connected to previous research.
Response 6: We found this suggestion is important. Therefore,in the conclusion section, we have added a paragraph to connect our study to the previous research and state our contributions. They are in the lines 36-53 of page 24 and lines 1-2 of page 25 in Word's " Track Changes " mode.
Specific modifications:
In our study, based on combing the prior literature, we construct the path of channel strategy selection on enterprise efficiency to provide a theoretical basis for the model of channel strategy selection affecting retail enterprise efficiency. In this paper, the design of the input-output indicators of retail enterprise efficiency is often lacking in the design of the industry-specific characteristics. Therefore, this paper sets input-output indicators of retail firm efficiency based on the characteristics of retail firms with reference to the prior research. In terms of introducing new dummy variables, according to the literature review and the analysis of the current situation of retail channel development, it is clear that retail enterprises generally implement dual-channel strategy and channel integration strategy at present. Therefore, in this paper, in order to empirically prove the impact of channel strategy on retail enterprise efficiency, two models are constructed to introduce two dummy variables of whether to implement dual-channel strategy and whether to implement channel convergence strategy respectively, and the impact of the above two channel strategies on the comprehensive retail efficiency is empirically proven by combining the quantifiable related indicators in the impact path proposed in this paper, so as to provide suggestions for how domestic and foreign retail enterprises can maintain their own development and future sustainability in the era of epidemic. This paper provides suggestions on how domestic and international retailers can maintain their own development and future sustainability in the COVID-19 era.
Point 7: The Scientific and practical contributions must be explicitly stated.
Response 7: We found this suggestion is crucial. For an article,practical and scientific contributions are very important. Therefore, we added a discussion section which including practical and scientific contributions. They are in lines 30-53 of page 21 and lines 1-4 of page 22 in Word's " Track Changes " mode.
Specific modifications:
Practical contributions: Retail is the basis of commodity circulation, and the operation condition of retail industry is directly related to the speed and efficiency of commodity circulation; retail is the key carrier to guide production and expand consumption, and the development condition of retail industry is related to market prosperity and employment security. Based on the current development situation of retail enterprises in China, this paper summarizes the influence path of retail channel strategy on enterprise efficiency from four perspectives: market demand, channel resources, channel technology and channel system, and empirically analyzes the influence of retail channel strategy on enterprise comprehensive efficiency, which helps guide retail enterprises to adjust channel development strategy, promote the transformation and upgrading of retail industry, release development vitality and enhance development momentum.
Scientific contributions: Compared with other industries, the implementation of retail channel strategy is closely related to the successful transformation and upgrading of enterprises. With the development of technology, online channel strategies of different nature such as Internet platforms and mobile terminals have provided retail enterprises with new development solutions over time, but as the demographic dividend disappears and the drawbacks of online channels begin to emerge, the retail industry has started to experiment with various new channel practices. This has also led to academic discussions and studies on the relationship between the implementation of retail channel strategies and the efficiency of retail enterprises. However, there is a lack of literature on the relationship between the two, and most of the existing literature is only a description and summary of the symptoms, and further research is needed. Therefore, the theoretical significance of this paper is to further reveal the inner mechanism of the influence of retail channel strategy on enterprise efficiency according to the four elements of channel formation and evolution. The study of retail enterprise efficiency from the perspective of channel strategy selection not only enriches the study of the relationship between channel development and retail enterprise efficiency, but also provides retail enterprises with channel strategy guidance ideas, which has certain theoretical significance.
Point 8: Technical problems: Avoid bullets numbering in the abstract.
Response 8: Thank you for this useful comment. As recommended, we have avoided bullets numbering in the abstract.They are in lines 10-24 of page 1 Word's " Track Changes " mode.
Specific modifications:
Appropriate channel strategies can help improve the operational efficiency of retail firms and stimulate the domestic demand market. To study the sustainability of retail enterprises in the COVID-19 era, this paper uses the data of 29 Chinese listed retail enterprises from 2012-2021 as a sample, and uses Data Envelopment Analysis (DEA) to evaluate and analyze the efficiency of listed retail enterprises that implement different channel strategies. The results of the study show that (1) The implementation of channel strategy will re-integrate the information flow, capital flow, commercial flow and logistics of enterprises, and affect the efficiency of enterprises through four paths: market demand, channel resources, channel technology and channel system. (2) The implementation of dual-channel strategy enterprises in the short term technical efficiency than single-channel strategy enterprises. (3) The channel integration strategy of online ordering and store delivery has a significant positive impact on the overall efficiency of listed retail enterprises. (4) Inventory turnover rate has a significant positive impact on the overall efficiency of listed retail enterprises. The level of accounts receivable, the level of retail technology, the level of investment in fixed assets and the size of enterprises have a negative impact on the overall efficiency of listed retail enterprises.
Point 9: References are not in defined style (Capital letters, Abbreviated journal names, and everything else, see template)
Response 9: This comment is important. We apologize for not carefully checking the format of references.After checking the template of Sustainability, we have changed the references’ format into”Author 1, A.B.; Author 2, C.D. Title of the article. Abbreviated Journal Name Year, Volume, page rang”.This is in “References” section, lines 20-55 and 1-52 of page 26 and 27 in the Word's " Track Changes " model.
Specific modifications:
References
Verhoef, C; Kannan, K; Inman, J. From multi-channel retailing to omni-channel retailing: Introduction to the special issue on multi-channel retailing. J. Retail. 2015, 91(2):174—181.
Li, F; Li,D; Sun Y. The development process of omnichannel retailing theory research. J. Beijing. Univ. Technol. Bus. 2018, 33(5):33-40.
Amrouche, N; Yan, R. Implementing online store for national brand competing against private label. J. Bus. Res. 2012,65(3):325—332.
Lee, Z.W; Chan, T.H; Chong, A.Y; et al. Customer engagement through omni-channel retailing: The effects of channel integration quality. Industrial. Marketing. Management. 2018 ,77(2):90-101.
Li, Y; Lou, H; Yang, F; et al. Customer's reaction to cross-channel integration in omni-channel retailing: The mediating roles of retailer uncertainty, identity attractiveness and switching costs. Decision. Support. Systems. 2018, 109(5):50-60.
Shen, X.L; Li, Y.J; Sun, Y; et al. Channel integration quality, perceived fluency and omnichannel service usage: The inoderating roles of intemnal and external usage experience. Decision. Surort. Systems. 2018, 109(5):61-73.
Cao, L. Business model transformation in moving to a cross-channel retail strategy: A case study. Intenational. J. Electronic Commerce. 2014, 18(4):69-96.
Klcinlcrchcrk, Emrich, O; Hcrhauscn, D; et al. Websites as information hubs: How informational channel Integration and shopping benefit density interact in steering customers to the physical store.J. Association. Consumer. Res. 2018, 3(3):330-342.
Herhausen, D; Binder, J; Schoegel, M; et al. Integrating bricks with clicks: retailer-level and channel-level outcomes of online-offline channel integration. J. Retail. 2015, 91(2):309-325.
Akturk, M.S; Ketzenberg, M; Heim, G.R. Assessing impacts of introducing ship-to-store service on sales and returns in omni-channel retailing: A data analytics study. J. Operations. Management. 2018, 61(7):15-45.
Gallino, S; Moreno, A. Integration of online and offline channels in retail: The impact of sharing reliable inventory availability information. Management. Science. 2014, 60(6):1434-1451.
Gao, F; Su, X. Omni-channel retail operations with buy-online-and-pickup-in-store. Management. Science. 2016, 63(8):2478-2492.
Zhang, T.P; Ou, Y. Is RMB appreciation a nightmare for the Chinese firms? An analysis on firm profitability and exchange rate. International. Review. Economics. Finance. 2017, 54 (5):27-43.
Freeman, R.B; Nakamura, A.O; Nakamura, L; et al. Wal-Mart innovation and productivity: A viewpoint . Canadian J. Economics: Revue canadienne deconomique, 2011, 44(2):486-508.
Haskel, J; Sadun, R. Regulation and UK Retailing Productivity: Evidence from Micro Data. IZA Discussion Paper. 2012, 79(315):425-448.
Terpstra, D.E; Rozell, E. The Relationship of Staffing Practices to Organizational level Measures of Performance.Personnel. Psychology. 2010, 46(1):27-48.
Zhang, J; Paul, W; John, W; et al. Crafting Integrated Multi-channel Retailing Strategies. J. Interactive. Marketing. 2010,24(2):168-80.
Cao, J; Pan, X.F; Tian, G. Disproportional ownership structure and pay performance relationship: Evidence from China' s listed firms. J. Corporate. Finance. 2016, 6(17):541-554.
Wang, B. W. Reasons and countermeasures of online and offline integration in modern retail industry. J. Heilongjiang. Institute. Technol. 2017,17(6):76-80.
Huang, M; Li Y. Research on the impact path and effect of retail enterprises' omnichannel development level on efficiency. J. Beijing. Univ. Technol. Bus.2017, 32(6):35-44.
Sun Z. The impact of omnichannel development on the efficiency of retail enterprises. Bus. Econ. Res. 2019, (1):104-107.
Yan X. Research on the transformation and upgrading of China's traditional retail industry in the context of network economy. J. Heilongjiang. Institute. Technol. 2017, 17(12):75-79.
Lei, B; Zhao, M. Evaluation of input-output efficiency of online and offline retail enterprises. Statistics. Information. Forum. 2015, 30(5):80-86.
Lei L. Comparative study on the efficiency of pure physical retail, e-tailing and multi-channel retail enterprises. J. Beijing. Univ. Technol. Bus. 2018, 33(1):44-51.
Zhu H. Research on the structure, evolution and operation mechanism of channel ecosystem. Beijing:Economic Science Press. 2017:74-80.
Tang H. Virtual business circle growth:A theoretical analysis and empirical test. J. Hunan. Bus. School. 2015(2):25-32.
Tang, H.T; Zhang, J.Y. Virtual business district agglomeration:mechanism and effect analysis. China. Circulation. economy. 2014(2):83-87.
Stone, M; Matt, H; Mahnaz, K. Multichannel Customer Management: The Benefits and Challenges. J. Database. Marketing. 2002, 10(1):39-52.
Wallace, D.W; Giese, J.L; Johnson, J.L. Customer retailer loyalty in the context of multiple channel strategies. J. Retail. 2004, 80(4):249-263.
Cao, L; Li, L. The Impact of Cross-Channel Integration on Retailers' Sales Growth. J. Retail. 2015,91(2):198-216.
Kushwaha, T; Shankar, V. Optimal allocation of Marketing Efforts by customer Channel segment. MSI Reports 2007:7-207.
Avery, J; Steenburgh, T.J; Deighton, J. et al. Adding Bricks to Clicks: Predicting the Patterns of Cross-Channel Elasticities Over Time. Social. Science. Electronic. Publishing. 2013,76(3):96-111.
Rangaswamy, A; Bruggen, G. Opportunities and Challenges in Multichannel Marketing an Introduction to the Special Issue. J. Interactive Marketing. 2010,19(2):5-11.
Neslin, S.A; Venkatesh, S. Key Issues in Multichannel Customer Management: Current Knowledge and Future Directions. J. Interactive. Marketing. 2009, 23(1):70-81.
Balasubramanian, S; Raghunathan, R; Mahajan, V. Consumers in a Multichannel Environment: Product Utility, Process Utility, and Channe Choice. J. Interactive. Marketing. 2005,19(2):12-30.
Barney, J. Firm resources and sustained competitive advantage. J. Management. 1991,17(1):99-120.
Wernerfelt, B.A. Resource-based view of the firm, strategic management Journal. Strategic. Management. Journal.1984,5(2):171-180.
Peng, Y. Research on transaction cost of B2C e-commerce enterprises in China. North China University of Electric Power. 2015.
Lai, H. The innovation mechanism of digital technology empowerment and "new retailing" with the example of Ali Rhino and Jindoduo. China. circulation. economy. 2020, 34(12):11-19.
Wang, X. New retailing in the new era: the trend of e-commerce transformation and upgrading under the wave of digital economy. J. Beijing. Univ. Commerce. Industry. 2020, 35(5):38-45.
Andreini, D. Multi-Channel Integration Strategies and Environmental Aspects A Conceptual Framework In Retailing. 8th Global conference on business Economics. 2008:1-25.
Wang, G.S; Zhi, X.J; Hu, G.W. An empirical analysis of efficiency changes in O2O transformation of retail enterprises. Systems. Engineering. 2016, 34(11):98-104.
Hean, T.K; Singfat, C. Retail Productivity and Scale Economies at the Firm Level: a DEA Approach. The International. J. Management. Science. 2003, 31(1):75-82.
Zhao, Y; Zhang, F. Research on the evaluation mechanism of marketing efficiency and profitability efficiency of listed retail enterprises based on a two-stage DEA perspective. Shanghai. Management. Science. 2016, 38(1):31-36.
Qin, W.; Qi, X. Evaluation of Green Logistics Efficiency in Northwest China. Sustainability. 2022, 14, 6848.
Zhao, W.; Qiu, Y.; Lu, W.; Yuan, P. Input–Output Efficiency of Chinese Power Generation Enterprises and Its Improvement Direction-Based on Three-Stage DEA Model. Sustainability. 2022, 14, 7421.
Andrejić M., Kilibarda, M. Pajić, V. Measuring efficiency change in time applying malmquist productivity index: a case of distribution centres in Serbia. Ser. Mech. Eng. 2021, 19 (3), 499-514.
Point 10: Align the manuscript with instructions for authors.
Response 10: We found this comment is important and useful. As suggested, we have aligned the manuscript with instructions for authors.
Point 11: References are double numbered.
Response 11: This suggestion is crucial. We apologize again for not carefully checking the format of references, and we have removed double numbered of References.They are in “References” section, lines 20-55 and 1-52 of page 26 and 27 in the Word's " Track Changes " model.
Point 12: Tables are confusing and not arranged (for example Table 3 (complete table), Table 4 (second column)).
Response 12: We found this suggestion is important. Therefore,we rearranged Table 3 (complete table) and Table 4 (second column). They are in the page 11 and page 13 in Word's " Track Changes " mode.
Specific modifications:
Table 3. Descriptive statistics for input-output indicators
|
|
2012 |
2013 |
2014 |
2015 |
2016 |
2017 |
2018 |
2019 |
2020 |
2021 |
|||||||
|
Input Indicators
|
Total number of employees (in persons) |
|
|
||||||||||||||
|
average value |
9859 |
10127 |
10661 |
11141 |
11125 |
11175 |
11586 |
12090 |
12358 |
12679 |
|||||||
|
minimum value |
254 |
256 |
239 |
229 |
219 |
205 |
207 |
200 |
204 |
206 |
|||||||
|
maximum value |
51365 |
57561 |
73085 |
75179 |
70440 |
84931 |
92047 |
110778 |
109657 |
111362 |
|||||||
|
standard deviation |
12106 |
12895 |
14878 |
15386 |
14643 |
16506 |
18103 |
21274 |
21974 |
22031 |
|||||||
|
Cost of main operations (in millions of dollars) |
|
|
|||||||||||||||
|
average value |
9093.58 |
9933.15 |
10234.20 |
11047.76 |
11593.11 |
13259.07 |
15217.32 |
16303.56 |
16923.47 |
17058.19 |
|||||||
|
minimum value |
655.14 |
725.27 |
671.60 |
691.32 |
641.53 |
671.12 |
709.91 |
652.84 |
678.39 |
694.86 |
|||||||
|
maximum value |
79955.58 |
88141.17 |
91011.73 |
112463.70 |
125467.80 |
156637.90 |
201719.10 |
222081.20 |
231677.65 |
239482.76 |
|||||||
|
standard deviation |
14761.22 |
16331.96 |
16962.96 |
20833.27 |
23268.41 |
29026.72 |
37341.80 |
41440.05 |
42893.09 |
43017.41 |
|||||||
|
Administrative expenses (in millions of dollars) |
|
|
|||||||||||||||
|
average value |
389.24 |
446.37 |
472.80 |
535.17 |
539.02 |
613.94 |
763.80 |
764.76 |
755.94 |
739.62 |
|||||||
|
minimum value |
41.35 |
49.72 |
45.86 |
47.37 |
43.68 |
43.33 |
48.46 |
50.25 |
46.22 |
43.71 |
|||||||
|
maximum value |
2350.11 |
2805.67 |
3356.57 |
4291.48 |
3946.27 |
4864.05 |
7462.56 |
8212.64 |
8165.39 |
8124.08 |
|||||||
|
standard deviation |
447.07 |
526.38 |
621.60 |
782.14 |
736.23 |
913.67 |
1416.26 |
1505.01 |
1489.66 |
1463.80 |
|||||||
|
Cost of sales (in millions of dollars) |
|
|
|||||||||||||||
|
average value |
1206.88 |
1367.65 |
1523.18 |
1720.05 |
1811.81 |
1991.26 |
2327.89 |
2690.79 |
2725.41 |
2913.05 |
|||||||
|
minimum value |
38.90 |
46.34 |
42.31 |
37.34 |
34.92 |
37.03 |
35.40 |
33.83 |
35.22 |
36.37 |
|||||||
|
maximum value |
11810.94 |
12739.71 |
14105.02 |
16644.68 |
17451.42 |
20635.78 |
26066.68 |
33532.02 |
34177.91 |
34806.47 |
|||||||
|
standard deviation |
2226.00 |
2433.21 |
2727.11 |
3218.02 |
3387.21 |
3988.80 |
5086.90 |
6488.69 |
6921.39 |
7134.95 |
|||||||
|
Total number of stores (in units) |
|
|
|||||||||||||||
|
average value |
168 |
169 |
184 |
190 |
186 |
295 |
478 |
308 |
312 |
343 |
|||||||
|
minimum value |
1 |
1 |
1 |
1 |
1 |
1 |
1 |
1 |
1 |
1 |
|||||||
|
maximum value |
1705 |
1585 |
1696 |
1638 |
1510 |
3867 |
8881 |
3630 |
4519 |
4822 |
|||||||
|
standard deviation |
351 |
338 |
357 |
353 |
327 |
733 |
1638 |
705 |
882 |
831 |
|||||||
|
Output Indicators |
Revenue from main business (in millions of yuan) |
|
|
||||||||||||||
|
average |
9093.58 |
9933.15 |
10234.20 |
11047.76 |
11593.11 |
13259.07 |
15217.32 |
16303.56 |
16953.29 |
16837.71 |
|||||||
|
minimum |
655.14 |
725.27 |
671.60 |
691.32 |
641.53 |
671.12 |
709.91 |
652.84 |
679.48 |
692.36 |
|||||||
|
maximum |
79955.58 |
88141.17 |
91011.73 |
112463.70 |
125467.80 |
156637.90 |
201719.10 |
222081.20 |
229056.38 |
234617.59 |
|||||||
|
standard deviation |
14761.22 |
16331.96 |
16962.96 |
20833.27 |
23268.41 |
29026.72 |
37341.80 |
41440.05 |
42371.87 |
44298.53 |
|||||||
|
Gross margin |
|
|
|||||||||||||||
|
average |
0.16 |
0.16 |
0.16 |
0.16 |
0.17 |
0.16 |
0.17 |
0.16 |
0.17 |
0.18 |
|||||||
|
minimum |
0.09 |
0.09 |
0.10 |
0.10 |
0.10 |
0.11 |
0.11 |
0.10 |
0.11 |
0.11 |
|||||||
|
maximum |
0.23 |
0.32 |
0.30 |
0.25 |
0.25 |
0.25 |
0.27 |
0.27 |
0.27 |
0.28 |
|||||||
|
standard deviation |
0.03 |
0.04 |
0.04 |
0.04 |
0.04 |
0.04 |
0.04 |
0.04 |
0.4 |
0.4 |
|||||||
Table 4. Malmquist Productivity Index and its decomposition for listed firms in the retail sector (2012-2021)
|
|
Company Name |
PEC |
SEC |
PTC |
STC |
MI |
|
Implementing a dual-channel strategy companies |
||||||
|
Of which: implementation of integration strategies |
Suning.com, PRC e-commerce company |
1.0000 |
1.0000 |
1.0000 |
0.9895 |
0.9895 |
|
Three Rivers Shopping |
1.0000 |
1.0000 |
0.9989 |
0.9977 |
0.9964 |
|
|
Xinhua all over the world |
0.9949 |
0.9991 |
0.9955 |
0.9962 |
0.9862 |
|
|
Xujiahui neighborhood of Shanghai |
1.0000 |
1.0000 |
0.9970 |
0.9986 |
0.9956 |
|
|
Tianhong Stock |
1.0000 |
1.0000 |
1.0002 |
1.0000 |
1.0002 |
|
|
Dongbai Group |
1.0003 |
0.9986 |
0.9926 |
1.0005 |
0.9920 |
|
|
Chongqing Department Store |
0.9976 |
0.9916 |
0.9975 |
1.0016 |
0.9879 |
|
|
Shoushang shares |
0.9952 |
1.0048 |
0.9981 |
0.9995 |
0.9955 |
|
|
Hualian Supermarket |
0.9976 |
0.9999 |
1.0001 |
1.0001 |
0.9977 |
|
|
Wenfeng shares |
1.0000 |
0.9977 |
0.9934 |
0.9996 |
0.9907 |
|
|
Wing Fai Supermarket |
1.0044 |
0.9924 |
1.0006 |
0.9982 |
0.9953 |
|
|
Eurasian Group |
1.0000 |
0.9916 |
1.0000 |
0.9810 |
0.9724 |
|
|
Zhongxing Commercial |
1.0000 |
1.0000 |
1.0000 |
1.0000 |
1.0000 |
|
|
Tongcheng Holdings |
1.0100 |
1.0013 |
0.9967 |
1.0040 |
1.0094 |
|
|
Guangbai shares |
1.0000 |
1.0000 |
1.0011 |
1.0000 |
1.0011 |
|
|
AUA share |
0.9958 |
0.9982 |
0.9972 |
0.9950 |
0.9861 |
|
|
Hefei Department Store |
1.0000 |
0.9982 |
0.9984 |
0.9872 |
0.9836 |
|
|
China Hundred Group |
0.9996 |
1.0024 |
0.9997 |
1.0023 |
1.0037 |
|
|
Ginza shares |
0.9932 |
1.0006 |
0.9981 |
1.0006 |
0.9924 |
|
|
BBK |
1.0046 |
0.9920 |
0.9917 |
1.0066 |
0.9935 |
|
|
Beijing urban and rural areas |
0.9959 |
0.9970 |
0.9871 |
1.0012 |
0.9809 |
|
|
New world |
1.0000 |
1.0000 |
1.0000 |
1.0000 |
1.0000 |
|
|
Average of enterprises implementing integration strategies. |
0.9995 |
0.9984 |
0.9975 |
0.9981 |
0.9932 |
|
|
No integration strategy implemented |
Nanning Department Store |
0.9974 |
1.0020 |
0.9909 |
1.0011 |
0.9890 |
|
Xinhua Department Store |
0.9949 |
0.9991 |
0.9955 |
0.9962 |
0.9862 |
|
|
Ningbo Ciba |
1.0000 |
1.0000 |
1.0000 |
1.0000 |
1.0000 |
|
|
Wangfujing neighborhood of central Beijing, famous for its many shops and restaurants |
1.0000 |
0.9922 |
1.0000 |
1.0003 |
0.9921 |
|
|
Average of enterprises not implementing integration strategies. |
0.9981 |
0.9983 |
0.9966 |
0.9994 |
0.9918 |
|
|
Average of dual-channel strategy firms |
0.9983 |
0.9994 |
1.0073 |
1.0983 |
1.1721 |
|
|
Single-channel strategy |
||||||
|
|
Hangzhou Xiebai |
1.0038 |
1.0004 |
1.0060 |
0.9998 |
1.0106 |
|
Maoye Commercial |
1.0000 |
1.0000 |
1.0008 |
1.0000 |
1.0013 |
|
|
Hanshang Group |
1.0000 |
0.9849 |
0.9786 |
1.0353 |
0.9981 |
|
|
Average of single-channel strategy firms |
1.0013 |
0.9951 |
0.9951 |
1.0117 |
1.0325 |
|
Point 13: .Suggested references:
Andrejić M., Kilibarda, M. Pajić, V., (2021). Measuring efficiency change in time applying malmquist productivity index: a case of distribution centres in Serbia. FACTA UNIVERSITATIS, Series Mechanical Engineering, 19 (3), 499-514.
Qin, W.; Qi, X. Evaluation of Green Logistics Efficiency in Northwest China. Sustainability 2022, 14, 6848. https://doi.org/10.3390/su14116848
Zhao, W.; Qiu, Y.; Lu, W.; Yuan, P. Input–Output Efficiency of Chinese Power Generation Enterprises and Its Improvement Direction-Based on Three-Stage DEA Model. Sustainability 2022, 14, 7421. https://doi.org/10.3390/su14127421
Response 13: We found this suggestion is important and useful. Reading through the studies gave us a deeper understanding, which was helpful in writing our paper. Therefore, we have cited it. They are in the lines 20-21 of page 16 and lines 8-10 of page 17 in Word's " Track Changes " mode.
Specific modifications:
Depending on the data need to choose different panel data model for analysis, usually choose mixed estimation model, fixed effect model or random effect model [45,46].
A discrete restricted variable is one in which the observations of the natural number or integer measure are incomplete, i.e., do not reflect the full reality of the situation [47].
45.Qin, W.; Qi, X. Evaluation of Green Logistics Efficiency in Northwest China. Sustainability 2022, 14, 6848.
46.Zhao, W.; Qiu, Y.; Lu, W.; Yuan, P. Input–Output Efficiency of Chinese Power Generation Enterprises and Its Improvement Direction-Based on Three-Stage DEA Model. Sustainability 2022, 14, 7421.
47.Andrejić M., Kilibarda, M. Pajić, V. Measuring efficiency change in time applying malmquist productivity index: a case of distribution centres in Serbia. Ser. Mech. Eng. 2021, 19 (3), 499-514.
Point 14: English language and style are fine/minor spell check required.
Response 14: Thank you for this useful suggestion. As suggested, we have combed through the entire paper, and revised and corrected some expressions.
Thanks again to you for your careful review of our research paper during your busy schedules!

Reviewer 2 Report
Dear Authors
at the outset, I liked the study. However, I have a few concerns.
1. Abstract is too lengthy. Many sentences can be curtailed. e.g., there is no need to state a 1% or 5% level in the abstract. Cut it down.
2. Introduction lacks coherence. The introduction seems like a background section only. REVISE THE INTRODUCTION. The introduction should be in 2-4paragraphs: background, rationale, RQs, and section plan.
RQs must be laid out with a sound relevant literature review.
3. Methods, Results seem okay to me. However, why till 2019 only? it is 2022. Due to COVID19 or any specific reason?
4. P-18, L-44: From the specific analysis of retail enterprises' overall efficiency in Chapter 4----what do you mean by chapter 4?????
5. Discussion must be followed by implications. Implications (theoretical and practical) can be added as separate sections. (separate from the conclusion and expand)
6. Further, limitations and subsequent future directions too can be added.
7. Finally, where is sustainability???? How do you relate to the scope of the journal?
All the best.
Author Response
Response to Reviewer 2 Comments
Dear reviewer:
Hello, thank you very much for your careful review of our paper in your busy schedule and for your constructive suggestions. We have carefully studied your valuable suggestions and comments, and have revised and improved the paper according to your suggestions and comments (the modified part of the paper is marked out with the "Track Changes" function in Word). The specific responses are as follows:
Point 1: Dear Authors:
At the outset, I liked the study. However, I have a few concerns.Abstract is too lengthy. Many sentences can be curtailed. e.g., there is no need to state a 1% or 5% level in the abstract. Cut it down.
Response 1: Thank you very much for this valuable suggestion. We completely agree with your suggestion. A concise abstract is very important for an article.As suggested, we have rewritten the abstract and removed unnecessary sentences. They are on lines 10-24 of page 1 in Word's " Track Changes " mode.
Specific modifications:
Abstract: Appropriate channel strategies can help improve the operational efficiency of retail firms and stimulate the domestic demand market. To study the sustainability of retail enterprises in the COVID-19 era, this paper uses the data of 29 Chinese listed retail enterprises from 2012-2021 as a sample, and uses Data Envelopment Analysis (DEA) to evaluate and analyze the efficiency of listed retail enterprises that implement different channel strategies. The results of the study show that (1) The implementation of channel strategy will re-integrate the information flow, capital flow, commercial flow and logistics of enterprises, and affect the efficiency of enterprises through four paths: market demand, channel resources, channel technology and channel system. (2) The implementation of dual-channel strategy enterprises in the short term technical efficiency than single-channel strategy enterprises. (3) The channel integration strategy of online ordering and store delivery has a significant positive impact on the overall efficiency of listed retail enterprises. (4) Inventory turnover rate has a significant positive impact on the overall efficiency of listed retail enterprises. The level of accounts receivable, the level of retail technology, the level of investment in fixed assets and the size of enterprises have a negative impact on the overall efficiency of listed retail enterprises.
Point 2: Introduction lacks coherence. The introduction seems like a background section only. REVISE THE INTRODUCTION. The introduction should be in 2-4paragraphs: background, rationale, RQs(RQ = research question), and section plan.
RQs must be laid out with a sound relevant literature review.
Response 2: This comment is crucial. We apologize for the incoherence of the introduction. As the beginning of the article, the introduction is a very important part.Based on the comment, we divide the original Introduction into two new sections: Introduction and Literature review.We wrote the introduction into four paragraphs: background, rationale& RQs, Summarize and section plan. This is in the lines 27-43 of page 1, lines 1-55 of page 2 and lines 1-8 of page 3 in the Word's " Track Changes " model.
Specific modifications:
Background: The negative impact of the COVID-19 on China's economy still exists, and people's living habits and consumption patterns have changed. With the major domestic cycle as the main body and the dual domestic and international cycles promoting each other is the new pattern to accelerate economic recovery. In the whole domestic cycle, consumption has become the first pulling force of economic growth for six consecutive years, is the main focus of accelerating the release of domestic demand potential, but also to promote the key engine of the dual cycle. As a key vehicle to guide production and expand consumption, the retail industry is directly related to the speed and efficiency of commodity circulation, and is a key support force for national economic growth and social development. Since the retail industry is characterized by high revenue and low profit, the management efficiency of the retail industry is particularly important. Whether the current retail channel strategy can improve the efficiency of retail enterprises and reverse the decline is an urgent and important issue in the development of the retail industry.
Rationale&RQs: Nowadays, the retail industry has become a key force for national economic growth. Not only that, the retail industry has a large number of points and a wide range of characteristics, but also a large volume of labor demand, coupled with a low employment threshold, accommodating many laid-off and transferred labor, the contribution to national employment is also particularly outstanding. The retail industry has gradually become a major industry to promote economic growth and national employment, helping retail innovation and transformation, improving circulation efficiency and promoting consumer upgrading will become a key initiative for high-quality economic development. However, as economic development enters a new normal, the demographic dividend is nearly exhausted, economic growth and residential consumption growth is slowing down, the expansion of China's retail market is restricted after the epidemic, the development of Internet technology has extended retail channels from offline to online, the real economy has been hit, and the new network mobile channels have further intensified the brutal competition pattern in the retail industry. Due to the rapidly changing consumption habits and consumption concepts of the consumer groups, rising retail operating costs and the rapid development of online channels, retail companies are not only facing a series of uncertainties in their investment in online channels, but also the layout and planning of their offline stores and other preliminary investments are difficult to obtain equivalent returns in the short term. This series of reasons lead to many retail enterprises' revenue not being equal to their operating costs. In the face of the above dilemma, can the current channel strategy of retail enterprises improve the overall efficiency of the enterprise? In the era of COVID-19, how should retailers choose their channel strategies to improve their overall efficiency, maintain their business and sustainable development in the future?
Summarize: In summary, changes in the macro environment and intensified competition in the industry have prompted continuous changes in retail channels, and retail transformation is urgent. It is of great theoretical and practical significance to study the relationship between channel strategies of retail enterprises and the overall efficiency of enterprises to transform the retail industry and stimulate market vitality. Considering that the sample needs to meet the requirements of comparability, stability and data availability, this paper first selects the retail enterprises listed in Shenzhen and Shanghai before 2011 according to the sample study period, and excludes the enterprises with major restructuring and missing main variables in the sample period. In addition, in order to ensure that the selected sample is representative of the retail industry, this paper takes the revenue share of retailing in the actual operation of enterprises as the standard according to the regulations of China Tax Law and Commerce Bureau, and eliminates enterprises whose main business share of retailing is below 60%, and finally gets 29 retail enterprises as the final research sample of this paper. Internationalization has increasingly become an important issue and an inevitable development trend in modern retail industry. This study focuses on Chinese enterprises and analyzes the impact of channel strategy on the efficiency of these 29 listed retail enterprises through Data Envelopment Analysis (DEA) and Tobit regression methods, with a view to improving the understanding of channel strategy for the retail industry at home and abroad. The aim is to raise awareness of channel strategy for the retail industry at home and abroad, and to provide useful and feasible suggestions for the channel selection of modern retail enterprises.
Section plan: The study of this paper is divided into seven parts. The first part is the introduction. It introduces the research background of this paper in detail, presents the research topic, and explains the significance of this paper. The second part and the third part mainly comb through the literature related to retail channel strategy and retail firm efficiency, analyze the path and the inner mechanism of constructing retail channel strategy on firm efficiency, and provide methodological and theoretical references for the following empirical analysis and research. Part 4 and Part 5 will conduct the empirical analysis according to the common process of DEA efficiency research. In the fourth part, the efficiency of 29 Chinese listed retail companies is measured by applying the DEA method, and then classified and analyzed according to the different channel strategies implemented. In the fifth section, the impact of channel strategy on the overall efficiency of retailing is empirically demonstrated by using the global covariate efficiency value measured by DEA-Malmquist as the explanatory variable, introducing a dummy variable of whether channel strategy is implemented or not, and controlling for other relevant internal and external possible influencing factors. The sixth section presents the practical and theoretical implications and recommendations of this paper. Finally, the conclusion section presents conclusions to promote the overall efficiency improvement of listed retail firms in their future sustainable development.
Point 3: Methods, Results seem okay to me. However, why till 2019 only? it is 2022. Due to COVID19 or any specific reason?
Response 3: We found this comment is important and useful. We apologize for the lack of consideration in our writing.As recommended, we extended the observation period from 2012-2019 to 2012-2021, to better examine the channel selection options for retailers in the era of the COVID-19.However, because of the completeness of the study time and the availability of data, we were unable to update the data to 2022.On the bright side, the data updated to 2021 still contributes much to the reliability of this paper.We have rewritten the discussion section and some of the conclusions based on the updated data, and in the discussion section we added suggestions for retail businesses to grow in today's epidemic. Here is the updated data and discussion section. The discussion section is in lines 29-53 of page 21,lines 1-54 of page 22,lines 1-54 of page 23 and lines 1-18 of page 24 in the Word's " Track Changes " model.
Specific modifications:
Table 3. Descriptive statistics for input-output indicators
|
|
2012 |
2013 |
2014 |
2015 |
2016 |
2017 |
2018 |
2019 |
2020 |
2021 |
|||||||
|
Input Indicators
|
Total number of employees (in persons) |
|
|
||||||||||||||
|
average value |
9859 |
10127 |
10661 |
11141 |
11125 |
11175 |
11586 |
12090 |
12358 |
12679 |
|||||||
|
minimum value |
254 |
256 |
239 |
229 |
219 |
205 |
207 |
200 |
204 |
206 |
|||||||
|
maximum value |
51365 |
57561 |
73085 |
75179 |
70440 |
84931 |
92047 |
110778 |
109657 |
111362 |
|||||||
|
standard deviation |
12106 |
12895 |
14878 |
15386 |
14643 |
16506 |
18103 |
21274 |
21974 |
22031 |
|||||||
|
Cost of main operations (in millions of dollars) |
|
|
|||||||||||||||
|
average value |
9093.58 |
9933.15 |
10234.20 |
11047.76 |
11593.11 |
13259.07 |
15217.32 |
16303.56 |
16923.47 |
17058.19 |
|||||||
|
minimum value |
655.14 |
725.27 |
671.60 |
691.32 |
641.53 |
671.12 |
709.91 |
652.84 |
678.39 |
694.86 |
|||||||
|
maximum value |
79955.58 |
88141.17 |
91011.73 |
112463.70 |
125467.80 |
156637.90 |
201719.10 |
222081.20 |
231677.65 |
239482.76 |
|||||||
|
standard deviation |
14761.22 |
16331.96 |
16962.96 |
20833.27 |
23268.41 |
29026.72 |
37341.80 |
41440.05 |
42893.09 |
43017.41 |
|||||||
|
Administrative expenses (in millions of dollars) |
|
|
|||||||||||||||
|
average value |
389.24 |
446.37 |
472.80 |
535.17 |
539.02 |
613.94 |
763.80 |
764.76 |
755.94 |
739.62 |
|||||||
|
minimum value |
41.35 |
49.72 |
45.86 |
47.37 |
43.68 |
43.33 |
48.46 |
50.25 |
46.22 |
43.71 |
|||||||
|
maximum value |
2350.11 |
2805.67 |
3356.57 |
4291.48 |
3946.27 |
4864.05 |
7462.56 |
8212.64 |
8165.39 |
8124.08 |
|||||||
|
standard deviation |
447.07 |
526.38 |
621.60 |
782.14 |
736.23 |
913.67 |
1416.26 |
1505.01 |
1489.66 |
1463.80 |
|||||||
|
Cost of sales (in millions of dollars) |
|
|
|||||||||||||||
|
average value |
1206.88 |
1367.65 |
1523.18 |
1720.05 |
1811.81 |
1991.26 |
2327.89 |
2690.79 |
2725.41 |
2913.05 |
|||||||
|
minimum value |
38.90 |
46.34 |
42.31 |
37.34 |
34.92 |
37.03 |
35.40 |
33.83 |
35.22 |
36.37 |
|||||||
|
maximum value |
11810.94 |
12739.71 |
14105.02 |
16644.68 |
17451.42 |
20635.78 |
26066.68 |
33532.02 |
34177.91 |
34806.47 |
|||||||
|
standard deviation |
2226.00 |
2433.21 |
2727.11 |
3218.02 |
3387.21 |
3988.80 |
5086.90 |
6488.69 |
6921.39 |
7134.95 |
|||||||
|
Total number of stores (in units) |
|
|
|||||||||||||||
|
average value |
168 |
169 |
184 |
190 |
186 |
295 |
478 |
308 |
312 |
343 |
|||||||
|
minimum value |
1 |
1 |
1 |
1 |
1 |
1 |
1 |
1 |
1 |
1 |
|||||||
|
maximum value |
1705 |
1585 |
1696 |
1638 |
1510 |
3867 |
8881 |
3630 |
4519 |
4822 |
|||||||
|
standard deviation |
351 |
338 |
357 |
353 |
327 |
733 |
1638 |
705 |
882 |
831 |
|||||||
|
Output Indicators |
Revenue from main business (in millions of yuan) |
|
|
||||||||||||||
|
average |
9093.58 |
9933.15 |
10234.20 |
11047.76 |
11593.11 |
13259.07 |
15217.32 |
16303.56 |
16953.29 |
16837.71 |
|||||||
|
minimum |
655.14 |
725.27 |
671.60 |
691.32 |
641.53 |
671.12 |
709.91 |
652.84 |
679.48 |
692.36 |
|||||||
|
maximum |
79955.58 |
88141.17 |
91011.73 |
112463.70 |
125467.80 |
156637.90 |
201719.10 |
222081.20 |
229056.38 |
234617.59 |
|||||||
|
standard deviation |
14761.22 |
16331.96 |
16962.96 |
20833.27 |
23268.41 |
29026.72 |
37341.80 |
41440.05 |
42371.87 |
44298.53 |
|||||||
|
Gross margin |
|
|
|||||||||||||||
|
average |
0.16 |
0.16 |
0.16 |
0.16 |
0.17 |
0.16 |
0.17 |
0.16 |
0.17 |
0.18 |
|||||||
|
minimum |
0.09 |
0.09 |
0.10 |
0.10 |
0.10 |
0.11 |
0.11 |
0.10 |
0.11 |
0.11 |
|||||||
|
maximum |
0.23 |
0.32 |
0.30 |
0.25 |
0.25 |
0.25 |
0.27 |
0.27 |
0.27 |
0.28 |
|||||||
|
standard deviation |
0.03 |
0.04 |
0.04 |
0.04 |
0.04 |
0.04 |
0.04 |
0.04 |
0.4 |
0.4 |
|||||||
Table 4. Malmquist Productivity Index and its decomposition for listed firms in the retail sector (2012-2021)
|
|
Company Name |
PEC |
SEC |
PTC |
STC |
MI |
|
Implementing a dual-channel strategy companies |
||||||
|
Of which: implementation of integration strategies |
Suning.com, PRC e-commerce company |
1.0000 |
1.0000 |
1.0000 |
0.9895 |
0.9895 |
|
Three Rivers Shopping |
1.0000 |
1.0000 |
0.9989 |
0.9977 |
0.9964 |
|
|
Xinhua all over the world |
0.9949 |
0.9991 |
0.9955 |
0.9962 |
0.9862 |
|
|
Xujiahui neighborhood of Shanghai |
1.0000 |
1.0000 |
0.9970 |
0.9986 |
0.9956 |
|
|
Tianhong Stock |
1.0000 |
1.0000 |
1.0002 |
1.0000 |
1.0002 |
|
|
Dongbai Group |
1.0003 |
0.9986 |
0.9926 |
1.0005 |
0.9920 |
|
|
Chongqing Department Store |
0.9976 |
0.9916 |
0.9975 |
1.0016 |
0.9879 |
|
|
Shoushang shares |
0.9952 |
1.0048 |
0.9981 |
0.9995 |
0.9955 |
|
|
Hualian Supermarket |
0.9976 |
0.9999 |
1.0001 |
1.0001 |
0.9977 |
|
|
Wenfeng shares |
1.0000 |
0.9977 |
0.9934 |
0.9996 |
0.9907 |
|
|
Wing Fai Supermarket |
1.0044 |
0.9924 |
1.0006 |
0.9982 |
0.9953 |
|
|
Eurasian Group |
1.0000 |
0.9916 |
1.0000 |
0.9810 |
0.9724 |
|
|
Zhongxing Commercial |
1.0000 |
1.0000 |
1.0000 |
1.0000 |
1.0000 |
|
|
Tongcheng Holdings |
1.0100 |
1.0013 |
0.9967 |
1.0040 |
1.0094 |
|
|
Guangbai shares |
1.0000 |
1.0000 |
1.0011 |
1.0000 |
1.0011 |
|
|
AUA share |
0.9958 |
0.9982 |
0.9972 |
0.9950 |
0.9861 |
|
|
Hefei Department Store |
1.0000 |
0.9982 |
0.9984 |
0.9872 |
0.9836 |
|
|
China Hundred Group |
0.9996 |
1.0024 |
0.9997 |
1.0023 |
1.0037 |
|
|
Ginza shares |
0.9932 |
1.0006 |
0.9981 |
1.0006 |
0.9924 |
|
|
BBK |
1.0046 |
0.9920 |
0.9917 |
1.0066 |
0.9935 |
|
|
Beijing urban and rural areas |
0.9959 |
0.9970 |
0.9871 |
1.0012 |
0.9809 |
|
|
new world |
1.0000 |
1.0000 |
1.0000 |
1.0000 |
1.0000 |
|
|
Average of enterprises implementing integration strategies. |
0.9995 |
0.9984 |
0.9975 |
0.9981 |
0.9932 |
|
|
No integration strategy implemented |
Nanning Department Store |
0.9974 |
1.0020 |
0.9909 |
1.0011 |
0.9890 |
|
Xinhua Department Store |
0.9949 |
0.9991 |
0.9955 |
0.9962 |
0.9862 |
|
|
Ningbo Ciba |
1.0000 |
1.0000 |
1.0000 |
1.0000 |
1.0000 |
|
|
Wangfujing neighborhood of central Beijing, famous for its many shops and restaurants |
1.0000 |
0.9922 |
1.0000 |
1.0003 |
0.9921 |
|
|
Average of enterprises not implementing integration strategies. |
0.9981 |
0.9983 |
0.9966 |
0.9994 |
0.9918 |
|
|
Average of dual-channel strategy firms |
0.9983 |
0.9994 |
1.0073 |
1.0983 |
1.1721 |
|
|
Single-channel strategy |
||||||
|
|
Hangzhou Xiebai |
1.0038 |
1.0004 |
1.0060 |
0.9998 |
1.0106 |
|
Maoye Commercial |
1.0000 |
1.0000 |
1.0008 |
1.0000 |
1.0013 |
|
|
Hanshang Group |
1.0000 |
0.9849 |
0.9786 |
1.0353 |
0.9981 |
|
|
Average of single-channel strategy firms |
1.0013 |
0.9951 |
0.9951 |
1.0117 |
1.0325 |
|
Table 6. Descriptive statistics of variables
|
Impact variables |
2013 |
2014 |
2015 |
2016 |
2017 |
2018 |
2019 |
2020 |
2021 |
|
Level of receivables (receivables) |
|
|
|||||||
|
average |
0.0065 |
0.0061 |
0.0059 |
0.0075 |
0.0102 |
0.0122 |
0.0146 |
0.0138 |
0.0142 |
|
minimum |
0.0000 |
0.0000 |
0.0001 |
0.0003 |
0.0006 |
0.0004 |
0.0004 |
0.0004 |
0.0004 |
|
maximum |
0.0329 |
0.0344 |
0.0381 |
0.0316 |
0.0501 |
0.0732 |
0.1584 |
0.1682 |
0.1893 |
|
standard deviation |
0.0068 |
0.0069 |
0.0072 |
0.0079 |
0.0115 |
0.0152 |
0.0288 |
0.0317 |
0.0396 |
|
Selling expense ratio (marketing) |
|
|
|||||||
|
average |
0.0885 |
0.0966 |
0.0992 |
0.1014 |
0.0993 |
0.1039 |
0.1081 |
0.1143 |
0.1375 |
|
minimum |
0.0205 |
0.0220 |
0.0213 |
0.0194 |
0.0212 |
0.0210 |
0.0193 |
0.0204 |
0.0254 |
|
maximum |
0.1697 |
0.1793 |
0.1874 |
0.1992 |
0.1843 |
0.1895 |
0.2074 |
0.2676 |
0.2865 |
|
standard deviation |
0.0476 |
0.0503 |
0.0525 |
0.0525 |
0.0513 |
0.0516 |
0.0546 |
0.0569 |
0.0583 |
|
Fixed asset input intensity (fixed) |
|
|
|||||||
|
average |
0.2708 |
0.2758 |
0.2734 |
0.2818 |
0.2797 |
0.2752 |
0.2671 |
0.2890 |
0.2946 |
|
minimum |
0.0677 |
0.0536 |
0.0538 |
0.0603 |
0.0749 |
0.0662 |
0.0752 |
0.0815 |
0.0947 |
|
maximum |
0.7292 |
0.7269 |
0.7000 |
0.6633 |
0.6608 |
0.6490 |
0.6400 |
0.6695 |
0.6946 |
|
standard deviation |
0.1593 |
0.1604 |
0.1656 |
0.1635 |
0.1528 |
0.1562 |
0.1610 |
0.1677 |
0.1745 |
|
Level of technological input (technology) |
|
|
|||||||
|
average |
0.00126 |
0.00216 |
0.00221 |
0.00183 |
0.00170 |
0.00140 |
0.00148 |
0.00168 |
0.00186 |
|
minimum |
0.00000 |
0.00000 |
0.00000 |
0.00000 |
0.00000 |
0.00001 |
0.00002 |
0.00002 |
0.00002 |
|
maximum |
0.00515 |
0.01507 |
0.01489 |
0.01105 |
0.01014 |
0.00637 |
0.00680 |
0.00695 |
0.00734 |
|
standard deviation |
0.00155 |
0.00318 |
0.00333 |
0.00231 |
0.00218 |
0.00158 |
0.00178 |
0.00186 |
0.00189 |
|
Inventory turnover rate (Invent) |
|
|
|||||||
|
average |
18.64 |
16.38 |
16.64 |
15.95 |
17.18 |
18.91 |
19.79 |
19.92 |
20.17 |
|
minimum |
1.82 |
0.80 |
0.46 |
0.72 |
0.89 |
0.84 |
1.19 |
1.25 |
1.38 |
|
maximum |
106.86 |
68.32 |
64.10 |
64.38 |
78.08 |
95.06 |
116.85 |
134.98 |
150.73 |
|
standard deviation |
21.83 |
17.54 |
16.72 |
16.94 |
19.98 |
23.93 |
28.70 |
32.86 |
37.83 |
|
Gearing ratio (asset) |
|
|
|||||||
|
average |
0.5652 |
0.5514 |
0.5414 |
0.5279 |
0.5299 |
0.5210 |
0.5197 |
0.5158 |
0.5093 |
|
minimum |
0.1998 |
0.1871 |
0.1752 |
0.1865 |
0.1809 |
0.1666 |
0.1534 |
0.1521 |
0.1437 |
|
maximum |
0.9408 |
0.9401 |
0.9356 |
0.9315 |
0.9316 |
0.9298 |
0.9205 |
0.9201 |
0.9188 |
|
standard deviation |
0.1489 |
0.1545 |
0.1664 |
0.1689 |
0.1677 |
0.1781 |
0.1760 |
0.1794 |
0.1821 |
|
Enterprise size (scale) |
|
|
|||||||
|
average |
8.617 |
8.653 |
8.750 |
8.860 |
8.891 |
8.966 |
9.017 |
9.344 |
9.575 |
|
minimum |
6.607 |
6.631 |
6.656 |
6.712 |
6.714 |
6.715 |
7.229 |
7.486 |
7.492 |
|
Continued Table 6 |
|||||||||
|
Impact variables |
2013 |
2014 |
2015 |
2016 |
2017 |
2018 |
2019 |
2020 |
2021 |
|
maximum |
11.327 |
11.317 |
11.386 |
11.829 |
11.966 |
12.203 |
12.375 |
12.464 |
12.581 |
|
(statistics) standard deviation |
0.886 |
0.903 |
0.923 |
1.018 |
1.051 |
1.092 |
1.103 |
1.179 |
1.221 |
|
Total retail sales of consumer goods (retail, in billions of yuan) |
|
|
|||||||
|
average |
8157.45 |
9106.07 |
10069.10 |
11084.04 |
12161.93 |
13134.52 |
14043.80 |
14468.65 |
14679.18 |
|
minimum |
668.51 |
737.18 |
789.60 |
850.10 |
930.40 |
1330.10 |
1399.40 |
1473.45 |
1498.22 |
|
maximum |
25453.93 |
28471.15 |
31517.60 |
34739.10 |
38200.10 |
39767.10 |
42951.80 |
42977.35 |
43176.64 |
|
standard deviation |
4271.67 |
4767.73 |
5312.20 |
5861.13 |
6441.16 |
6739.47 |
7298.13 |
7375.57 |
7396.13 |
Discussion part:
Retail is the basis of commodity circulation, and the operation condition of retail industry is directly related to the speed and efficiency of commodity circulation; retail is the key carrier to guide production and expand consumption, and the development condition of retail industry is related to market prosperity and employment security. Based on the current development situation of retail enterprises in China, this paper summarizes the influence path of retail channel strategy on enterprise efficiency from four perspectives: market demand, channel resources, channel technology and channel system, and empirically analyzes the influence of retail channel strategy on enterprise comprehensive efficiency, which helps guide retail enterprises to adjust channel development strategy, promote the transformation and upgrading of retail industry, release development vitality and enhance development momentum.
Compared with other industries, the implementation of retail channel strategy is closely related to the successful transformation and upgrading of enterprises. With the development of technology, online channel strategies of different nature such as Internet platforms and mobile terminals have provided retail enterprises with new development solutions over time, but as the demographic dividend disappears and the drawbacks of online channels begin to emerge, the retail industry has started to experiment with various new channel practices. This has also led to academic discussions and studies on the relationship between the implementation of retail channel strategies and the efficiency of retail enterprises. However, there is a lack of literature on the relationship between the two, and most of the existing literature is only a description and summary of the symptoms, and further research is needed. Therefore, the theoretical significance of this paper is to further reveal the inner mechanism of the influence of retail channel strategy on enterprise efficiency according to the four elements of channel formation and evolution. The study of retail enterprise efficiency from the perspective of channel strategy selection not only enriches the study of the relationship between channel development and retail enterprise efficiency, but also provides retail enterprises with channel strategy guidance ideas, which has certain theoretical significance.
This paper finds that the implementation of a dual-channel strategy may have a positive impact on the overall efficiency of the firm in the early stage, but does not have a significant impact in the long term. Implementing a channel convergence strategy on this basis will have a positive impact on the overall efficiency of the firm. Given that most of the retailers have already opened dual-channel strategies, online channels do allow retailers with a large number of local stores to break through technological bottlenecks and improve overall efficiency. Therefore, this paper encourages domestic and international retailers implementing dual-channel strategies to implement corresponding channel integration measures after adopting online channel operations.
Based on the above findings, this paper makes the following suggestions:
(1)Focus on demand to promote efficiency. At this stage, retail enterprises need to recognize a problem, new channels or the implementation of various channel strategies should be around the customer, insight into the changes in customer demand, to achieve accurate service. This requires enterprises in all aspects of channel operations should be customer-centric online and offline resource allocation, with the channel to capture customers and seize customers. From the perspective of actual development, channel integration strategy refers to the integration of online stores, physical stores and modern logistics, so that the traditional retailing of people, goods and fields break through the limitations of time and space, bringing higher value returns for enterprises. However, in fact, many retail companies that have implemented convergence strategies have not seen significant growth in performance, and even a decline. The reason for this situation is that this transformation has not produced the expected effect on the improvement of the viscosity of existing users and the attraction of new ones. So to break the diversion dilemma, companies need to do is independent innovation, in the general implementation of the "retail + Internet" logic to add content section into the "retail + Internet + content". In the future, the mass marketing strategy with low marketing efficiency will gradually disappear, and mass marketing will be gradually transformed into niche marketing. Enterprises should first clarify the target customer groups and their needs, and then carry out precise marketing to them, so that customers can get a more ultimate shopping experience. In the channel integration mode, all channels of the enterprise can be used to help customers solve their problems. Each channel has advantages and disadvantages, and retail enterprises must give full play to the advantages of each channel, use and integrate existing resources to achieve mutual diversion of online and offline traffic, so that online traffic and offline traffic can be shared. In this way, enterprises can fully understand customer information and carry out accurate marketing to lay a good foundation.
(2)Good placement of resources to increase efficiency. Facing the impact of e-commerce and consumers' diversified channel demand, many retail enterprises are trying to operate multi-channel and actively implement relevant channel integration strategies, but the rapid development of online sales does not drive all enterprises to improve efficiency. The reason for this is that many traditional retailers are developing online channels as a passive choice and blindly following the trend forced by the market, and do not really understand how to operate online channels. At the same time, retailers often have two misconceptions about online sales: one is that the service cost of online channels is lower than that of physical stores; the other is that digital business growth to a certain scale will bring profitability. However, "more" is not the same as profitability, and there is no positive correlation between multi-channel operations and return on revenue. When companies lay out multi-channel operations, they should fully consider financial costs such as return on investment and develop appropriate channel strategies so that sales growth can really bring about an increase in profitability. With the maturity of the overall business ecology of the Internet, the expansion of online shopping scale has become an irreversible trend, but in the online retail boom, enterprises need to think cold. Although the addition of online channels allows companies to gain objective sales, the growth of digital sales does not necessarily lead to increased profitability, and may even threaten the overall effectiveness of the company. For example, for companies implementing dual-channel and channel convergence strategies, their supply chain and IT costs are much higher than the costs of previous offline channels. Therefore, retailers should realize that the key to building a multi-channel retail model is to look at the whole picture, find suitable partners from the perspective of a higher-level retail ecosystem, focus on improving their own core competitiveness, and attract more consumers through quality experiences.
(3)Enhance the strong effect of technology. Leveraging the "Internet+" to improve information technology capabilities and achieve integrated development in the COVID-19 era. In the study of the path of retail enterprises to influence the efficiency of enterprises through the introduction of technology, it is found that most retail enterprises' existing technology investment is limited to the investment in information software. With the in-depth development of "Internet+", retail enterprises should take advantage of "Internet+", play the role of Internet, coordinate the mutually exclusive influence of different channels, amplify the complementary advantages of channels, and realize multi-channel integration. To achieve integrated development by leveraging "Internet+", the key lies in the following two points. First, actively introduce online channels and implement dual-channel strategy, which should pay attention to the scale efficiency of enterprises and improve the operation process from the global system level to bring into play the synergy effect of multiple channels. Second, the introduction of the Internet and other related advanced technologies should be based on the strategic guidance of the enterprise, and should be introduced in a target-oriented and targeted manner under the condition that the enterprise fully understands its own strengths and weaknesses, rather than blindly following the trend of development and imitating other enterprises, resulting in a decline in enterprise efficiency. With the help of big data, improve data analysis capabilities and realize retail transformation. There are only a few retail enterprises using big data to operate in China, and the application of big data is still in the exploration and exploration stage. However, the advantages of big data are obvious to all, and enterprises can use big data to analyze consumer behavior information and improve category management and marketing strategies. In the era of big data, it is necessary to analyze the data. And to leverage big data to achieve transformation, retail enterprises should first broaden the depth and breadth of the application of big data, access to data and analysis of all parties, on the basis of data analysis to speed up the operation of enterprises in logistics, supply chain and other links, in the enterprise to make decisions to provide accurate more for information. Secondly, enterprises should focus on the development of big data products. In the process of upgrading to a platform-based enterprise can also be used to build a platform ecosystem with the application of big data, to provide help for the partners in the platform and achieve common development.
(4)Strategic management of superior efficiency. The ultimate goal of enterprises is to obtain profits, so when retail enterprises carry out the strategic layout of channels, they should start from the perspective of improving the return on investment, look at the whole picture, and let multi-channel retailing serve the overall development of enterprises. From the current point of view, most domestic retail enterprises in the implementation of multi-channel operation strategy does not establish this global concept, more just a blind behavior. To truly tap the value of multi-channel retailing, Chinese retailers must find the best time and the most suitable investment area for multi-channel layout from the strategic perspective of long-term development, so that multi-channel operation can become a great help for enterprises to break through the offline sales dilemma and achieve benign and rapid development. The fundamental support for multi-channel retailing and even omni-channel retailing is still the offline capability of retail enterprises. Therefore, the key to the sustainable development and long-term profitability of multi-channel retailing model is for retail enterprises to return to their original mindset and continuously enhance their offline capabilities. For example, in the empirical evidence, we found that the inventory turnover rate of retail enterprises has a positive impact on the overall efficiency of enterprises, so enterprises can consider optimizing their management techniques from aspects such as merchandise planning capability, purchasing and transportation capability, ordering and replenishment capability, etc. to improve the turnaround speed to further improve the overall efficiency of the enterprise. With the integration of different channels such as offline, PC and mobile, the decentralized nature of channels will also play a key role in retail inventory optimization, and the multi-channel shared inventory strategy will also effectively improve the speed of product delivery and reduce logistics costs.
Point 4: P-18, L-44: From the specific analysis of retail enterprises' overall efficiency in Chapter 4----what do you mean by chapter 4?????
Response 4: We found this suggestion is very useful. Thank you very much for reading carefully and finding this error. Again, we apologize for our oversight of the language. What we want to express is that from the specific analysis of the overall efficiency of retail enterprises in the section above. We have now revised this expression, they are in the lines 12 of page 20 in Word's " Track Changes " mode.
Specific modifications:
From the specific analysis of retail enterprises' overall efficiency in the section above.
Point 5: Discussion must be followed by implications. Implications (theoretical and practical) can be added as separate sections. (separate from the conclusion and expand)
Response 5: We found this suggestion is crucial. For an article,practical and theoretical implications are very important. Therefore, we added a discussion section which including practical and theoretical implications. They are in lines 30-53 of page 21 and lines 1-4 of page 22 in Word's " Track Changes " mode.
Specific modifications:
Practical implications: Retail is the basis of commodity circulation, and the operation condition of retail industry is directly related to the speed and efficiency of commodity circulation; retail is the key carrier to guide production and expand consumption, and the development condition of retail industry is related to market prosperity and employment security. Based on the current development situation of retail enterprises in China, this paper summarizes the influence path of retail channel strategy on enterprise efficiency from four perspectives: market demand, channel resources, channel technology and channel system, and empirically analyzes the influence of retail channel strategy on enterprise comprehensive efficiency, which helps guide retail enterprises to adjust channel development strategy, promote the transformation and upgrading of retail industry, release development vitality and enhance development momentum.
Theoretical implications: Compared with other industries, the implementation of retail channel strategy is closely related to the successful transformation and upgrading of enterprises. With the development of technology, online channel strategies of different nature such as Internet platforms and mobile terminals have provided retail enterprises with new development solutions over time, but as the demographic dividend disappears and the drawbacks of online channels begin to emerge, the retail industry has started to experiment with various new channel practices. This has also led to academic discussions and studies on the relationship between the implementation of retail channel strategies and the efficiency of retail enterprises. However, there is a lack of literature on the relationship between the two, and most of the existing literature is only a description and summary of the symptoms, and further research is needed. Therefore, the theoretical significance of this paper is to further reveal the inner mechanism of the influence of retail channel strategy on enterprise efficiency according to the four elements of channel formation and evolution. The study of retail enterprise efficiency from the perspective of channel strategy selection not only enriches the study of the relationship between channel development and retail enterprise efficiency, but also provides retail enterprises with channel strategy guidance ideas, which has certain theoretical significance.
Point 6: Further, limitations and subsequent future directions too can be added.
Response 6: Thank you very much for this useful suggestion. As suggested, we have added limitations and subsequent future directions of this research in the discussion section.They are in lines 19-34 of page 24.
Specific modifications:
Discussion part(limitations and future research directions):
This paper researches the impact of channel strategy selection on the overall efficiency of listed retail enterprises and achieves some research results, but there are still some shortcomings to be improved in the specific research process. Due to the availability of data, all the listed retail enterprises in this paper were selected within the sample period, and considering the requirement of homogeneity of decision units in the DEA model, the retail enterprises listed outside Shenzhen and Shanghai and those whose main business of retailing is less than 60% were removed from this paper, making the final sample size smaller. Besides, future studies may optimize or use other efficiency measurement models to expand the sample scope for research. Further research on refined channel strategies can be conducted in the future. This study focuses more on the impact of dual-channel strategy and convergence strategy on the efficiency of enterprises, but no in-depth empirical analysis has been conducted on the specific ways of implementing dual-channel strategy and convergence strategy (self-built or cooperative). Therefore, future research can explore the specific ways of implementing channel strategies for enterprises and propose clearer development directions for management decisions.
Point 7: Finally, where is sustainability???? How do you relate to the scope of the journal?
Response 7: Thank you very much for this valuable suggestion.We should clarify that this paper is to study how retail enterprises should choose channel strategies to maintain their business operations and sustainable future directions in the COVID-19 era when the real economy is under attack, and to provide suggestions for sustainable business operations. As suggested, to be more relevant to the main theme of the journal,we have emphasized the link between this paper and sustainability in the title, abstract, keywords, introduction, discussion and conclusions. They are on lines 2-3,10-14, 25 of page 1; lines 21-23 of page 2; lines 7-8 of page 3; lines 33-40 of page 21 and lines 45-53 of page 24 in Word's " Track Changes " mode.
Specific modifications:
Title: Research on the sustainability of channel strategy selection on the overall efficiency of listed retail enterprises:Evidence from China
Abstract: Appropriate channel strategies can help improve the operational efficiency of retail firms and stimulate the domestic demand market. To study the sustainability of retail enterprises in the COVID-19 era, this paper uses the data of 29 Chinese listed retail enterprises from 2012-2021 as a sample, and uses Data Envelopment Analysis (DEA) to evaluate and analyze the efficiency of listed retail enterprises that implement different channel strategies.
Keywords: retail efficiency; sustainability; retail enterprise; channel strategy; DEA model
Introduction:In the era of COVID-19, how should retailers choose their channel strategies to improve their overall efficiency, maintain their business and sustainable development in the future?/Finally, the conclusion section presents conclusions to promote the overall efficiency improvement of listed retail firms in their future sustainable development.
Discussion: Based on the current development situation of retail enterprises in China, this paper summarizes the influence path of retail channel strategy on enterprise efficiency from four perspectives: market demand, channel resources, channel technology and channel system, and empirically analyzes the influence of retail channel strategy on enterprise comprehensive efficiency, which helps guide retail enterprises to adjust channel development strategy, promote the transformation and upgrading of retail industry, release development vitality and enhance development momentum.
Conclusions: Therefore, in this paper, in order to empirically prove the impact of channel strategy on retail enterprise efficiency, two models are constructed to introduce two dummy variables of whether to implement dual-channel strategy and whether to implement channel convergence strategy respectively, and the impact of the above two channel strategies on the comprehensive retail efficiency is empirically proven by combining the quantifiable related indicators in the impact path proposed in this paper, so as to provide suggestions for how domestic and foreign retail enterprises can maintain their own development and future sustainability in the era of epidemic.
Thanks again to you for your careful review of our research paper during your busy schedules!

Reviewer 3 Report
Building a stable competitive advantage is becoming increasingly difficult and increasingly depends on the value for customers creation by the enterprise. Undoubtedly, an important element of this process are activities in the area of ​​distribution. In this context, I believe that the problem raised in the article is valid. Below are some critical remarks:
1. In the article the authors refer to post-COVID-19 time (lines 10-18, p. 1) - research and analyzes, however, concern the period before the pandemic (line 18, p. 1).
2. Is the action contained in the statement: "In addition, in order to ensure that the selected sample is representative of the retail industry, our study also eliminates enterprises whose main business of retailing accounts for less than 60%, and finally obtains 29 retail enterprises as final research sample. " (lines 39-42, p. 2) really guarantees sample representativeness?
3. The article does not explain the symbols in formulas 1 - 3 (p. 6). MI was not explained until p. 6 (line 34), although it was used before. In line 19, p. 6 the symbol Sg was misspelled.
4. In tables 1 and 4, "broken" text appears (the table is therefore unreadable).
5. Misspelled ait and uit symbols in line 12, p. 15. The symbols in table 7 differ from those shown in models 10 - 11.
Author Response
Response to Reviewer 3 Comments
Dear reviewer:
Hello, thank you very much for your careful review of our paper in your busy schedule and for your constructive suggestions. We have carefully studied your valuable suggestions and comments, and have revised and improved the paper according to your suggestions and comments (the modified part of the paper is marked out with the "Track Changes" function in Word). The specific responses are as follows:
Point 1: Building a stable competitive advantage is becoming increasingly difficult and increasingly depends on the value for customers creation by the enterprise. Undoubtedly, an important element of this process are activities in the area of distribution. In this context, I believe that the problem raised in the article is valid. Below are some critical remarks:
In the article the authors refer to post-COVID-19 time (lines 10-18, p. 1) - research and analyzes, however, concern the period before the pandemic (line 18, p. 1).
Response 1: We found this comment is important and useful. We apologize for the lack of consideration in our writing.As recommended, we extended the observation period from 2012-2019 to 2012-2021, to better examine the channel selection options for retailers in the era of the COVID-19.However, because of the completeness of the study time and the availability of data, we were unable to update the data to 2022.On the bright side, the data updated to 2021 still contributes much to the reliability of this paper.We have rewritten the discussion section and some of the conclusions based on the updated data, and in the discussion section we added suggestions for retail businesses to grow in today's epidemic. Here is the updated data and discussion section. The discussion section is in lines 29-53 of page 21,lines 1-54 of page 22,lines 1-54 of page 23 and lines 1-18 of page 24 in the Word's " Track Changes " model.
Specific modifications:
Table 3. Descriptive statistics for input-output indicators
|
|
2012 |
2013 |
2014 |
2015 |
2016 |
2017 |
2018 |
2019 |
2020 |
2021 |
|||||||
|
Input Indicators
|
Total number of employees (in persons) |
|
|
||||||||||||||
|
average value |
9859 |
10127 |
10661 |
11141 |
11125 |
11175 |
11586 |
12090 |
12358 |
12679 |
|||||||
|
minimum value |
254 |
256 |
239 |
229 |
219 |
205 |
207 |
200 |
204 |
206 |
|||||||
|
maximum value |
51365 |
57561 |
73085 |
75179 |
70440 |
84931 |
92047 |
110778 |
109657 |
111362 |
|||||||
|
standard deviation |
12106 |
12895 |
14878 |
15386 |
14643 |
16506 |
18103 |
21274 |
21974 |
22031 |
|||||||
|
Cost of main operations (in millions of dollars) |
|
|
|||||||||||||||
|
average value |
9093.58 |
9933.15 |
10234.20 |
11047.76 |
11593.11 |
13259.07 |
15217.32 |
16303.56 |
16923.47 |
17058.19 |
|||||||
|
minimum value |
655.14 |
725.27 |
671.60 |
691.32 |
641.53 |
671.12 |
709.91 |
652.84 |
678.39 |
694.86 |
|||||||
|
maximum value |
79955.58 |
88141.17 |
91011.73 |
112463.70 |
125467.80 |
156637.90 |
201719.10 |
222081.20 |
231677.65 |
239482.76 |
|||||||
|
standard deviation |
14761.22 |
16331.96 |
16962.96 |
20833.27 |
23268.41 |
29026.72 |
37341.80 |
41440.05 |
42893.09 |
43017.41 |
|||||||
|
Administrative expenses (in millions of dollars) |
|
|
|||||||||||||||
|
average value |
389.24 |
446.37 |
472.80 |
535.17 |
539.02 |
613.94 |
763.80 |
764.76 |
755.94 |
739.62 |
|||||||
|
minimum value |
41.35 |
49.72 |
45.86 |
47.37 |
43.68 |
43.33 |
48.46 |
50.25 |
46.22 |
43.71 |
|||||||
|
maximum value |
2350.11 |
2805.67 |
3356.57 |
4291.48 |
3946.27 |
4864.05 |
7462.56 |
8212.64 |
8165.39 |
8124.08 |
|||||||
|
standard deviation |
447.07 |
526.38 |
621.60 |
782.14 |
736.23 |
913.67 |
1416.26 |
1505.01 |
1489.66 |
1463.80 |
|||||||
|
Cost of sales (in millions of dollars) |
|
|
|||||||||||||||
|
average value |
1206.88 |
1367.65 |
1523.18 |
1720.05 |
1811.81 |
1991.26 |
2327.89 |
2690.79 |
2725.41 |
2913.05 |
|||||||
|
minimum value |
38.90 |
46.34 |
42.31 |
37.34 |
34.92 |
37.03 |
35.40 |
33.83 |
35.22 |
36.37 |
|||||||
|
maximum value |
11810.94 |
12739.71 |
14105.02 |
16644.68 |
17451.42 |
20635.78 |
26066.68 |
33532.02 |
34177.91 |
34806.47 |
|||||||
|
standard deviation |
2226.00 |
2433.21 |
2727.11 |
3218.02 |
3387.21 |
3988.80 |
5086.90 |
6488.69 |
6921.39 |
7134.95 |
|||||||
|
Total number of stores (in units) |
|
|
|||||||||||||||
|
average value |
168 |
169 |
184 |
190 |
186 |
295 |
478 |
308 |
312 |
343 |
|||||||
|
minimum value |
1 |
1 |
1 |
1 |
1 |
1 |
1 |
1 |
1 |
1 |
|||||||
|
maximum value |
1705 |
1585 |
1696 |
1638 |
1510 |
3867 |
8881 |
3630 |
4519 |
4822 |
|||||||
|
standard deviation |
351 |
338 |
357 |
353 |
327 |
733 |
1638 |
705 |
882 |
831 |
|||||||
|
Output Indicators |
Revenue from main business (in millions of yuan) |
|
|
||||||||||||||
|
average |
9093.58 |
9933.15 |
10234.20 |
11047.76 |
11593.11 |
13259.07 |
15217.32 |
16303.56 |
16953.29 |
16837.71 |
|||||||
|
minimum |
655.14 |
725.27 |
671.60 |
691.32 |
641.53 |
671.12 |
709.91 |
652.84 |
679.48 |
692.36 |
|||||||
|
maximum |
79955.58 |
88141.17 |
91011.73 |
112463.70 |
125467.80 |
156637.90 |
201719.10 |
222081.20 |
229056.38 |
234617.59 |
|||||||
|
standard deviation |
14761.22 |
16331.96 |
16962.96 |
20833.27 |
23268.41 |
29026.72 |
37341.80 |
41440.05 |
42371.87 |
44298.53 |
|||||||
|
Gross margin |
|
|
|||||||||||||||
|
average |
0.16 |
0.16 |
0.16 |
0.16 |
0.17 |
0.16 |
0.17 |
0.16 |
0.17 |
0.18 |
|||||||
|
minimum |
0.09 |
0.09 |
0.10 |
0.10 |
0.10 |
0.11 |
0.11 |
0.10 |
0.11 |
0.11 |
|||||||
|
maximum |
0.23 |
0.32 |
0.30 |
0.25 |
0.25 |
0.25 |
0.27 |
0.27 |
0.27 |
0.28 |
|||||||
|
standard deviation |
0.03 |
0.04 |
0.04 |
0.04 |
0.04 |
0.04 |
0.04 |
0.04 |
0.4 |
0.4 |
|||||||
Table 4. Malmquist Productivity Index and its decomposition for listed firms in the retail sector (2012-2021)
|
|
Company Name |
PEC |
SEC |
PTC |
STC |
MI |
|
Implementing a dual-channel strategy companies |
||||||
|
Of which: implementation of integration strategies |
Suning.com, PRC e-commerce company |
1.0000 |
1.0000 |
1.0000 |
0.9895 |
0.9895 |
|
Three Rivers Shopping |
1.0000 |
1.0000 |
0.9989 |
0.9977 |
0.9964 |
|
|
Xinhua all over the world |
0.9949 |
0.9991 |
0.9955 |
0.9962 |
0.9862 |
|
|
Xujiahui neighborhood of Shanghai |
1.0000 |
1.0000 |
0.9970 |
0.9986 |
0.9956 |
|
|
Tianhong Stock |
1.0000 |
1.0000 |
1.0002 |
1.0000 |
1.0002 |
|
|
Dongbai Group |
1.0003 |
0.9986 |
0.9926 |
1.0005 |
0.9920 |
|
|
Chongqing Department Store |
0.9976 |
0.9916 |
0.9975 |
1.0016 |
0.9879 |
|
|
Shoushang shares |
0.9952 |
1.0048 |
0.9981 |
0.9995 |
0.9955 |
|
|
Hualian Supermarket |
0.9976 |
0.9999 |
1.0001 |
1.0001 |
0.9977 |
|
|
Wenfeng shares |
1.0000 |
0.9977 |
0.9934 |
0.9996 |
0.9907 |
|
|
Wing Fai Supermarket |
1.0044 |
0.9924 |
1.0006 |
0.9982 |
0.9953 |
|
|
Eurasian Group |
1.0000 |
0.9916 |
1.0000 |
0.9810 |
0.9724 |
|
|
Zhongxing Commercial |
1.0000 |
1.0000 |
1.0000 |
1.0000 |
1.0000 |
|
|
Tongcheng Holdings |
1.0100 |
1.0013 |
0.9967 |
1.0040 |
1.0094 |
|
|
Guangbai shares |
1.0000 |
1.0000 |
1.0011 |
1.0000 |
1.0011 |
|
|
AUA share |
0.9958 |
0.9982 |
0.9972 |
0.9950 |
0.9861 |
|
|
Hefei Department Store |
1.0000 |
0.9982 |
0.9984 |
0.9872 |
0.9836 |
|
|
China Hundred Group |
0.9996 |
1.0024 |
0.9997 |
1.0023 |
1.0037 |
|
|
Ginza shares |
0.9932 |
1.0006 |
0.9981 |
1.0006 |
0.9924 |
|
|
BBK |
1.0046 |
0.9920 |
0.9917 |
1.0066 |
0.9935 |
|
|
Beijing urban and rural areas |
0.9959 |
0.9970 |
0.9871 |
1.0012 |
0.9809 |
|
|
new world |
1.0000 |
1.0000 |
1.0000 |
1.0000 |
1.0000 |
|
|
Average of enterprises implementing integration strategies. |
0.9995 |
0.9984 |
0.9975 |
0.9981 |
0.9932 |
|
|
No integration strategy implemented |
Nanning Department Store |
0.9974 |
1.0020 |
0.9909 |
1.0011 |
0.9890 |
|
Xinhua Department Store |
0.9949 |
0.9991 |
0.9955 |
0.9962 |
0.9862 |
|
|
Ningbo Ciba |
1.0000 |
1.0000 |
1.0000 |
1.0000 |
1.0000 |
|
|
Wangfujing neighborhood of central Beijing, famous for its many shops and restaurants |
1.0000 |
0.9922 |
1.0000 |
1.0003 |
0.9921 |
|
|
Average of enterprises not implementing integration strategies. |
0.9981 |
0.9983 |
0.9966 |
0.9994 |
0.9918 |
|
|
Average of dual-channel strategy firms |
0.9983 |
0.9994 |
1.0073 |
1.0983 |
1.1721 |
|
|
Single-channel strategy |
||||||
|
|
Hangzhou Xiebai |
1.0038 |
1.0004 |
1.0060 |
0.9998 |
1.0106 |
|
Maoye Commercial |
1.0000 |
1.0000 |
1.0008 |
1.0000 |
1.0013 |
|
|
Hanshang Group |
1.0000 |
0.9849 |
0.9786 |
1.0353 |
0.9981 |
|
|
Average of single-channel strategy firms |
1.0013 |
0.9951 |
0.9951 |
1.0117 |
1.0325 |
|
Table 6. Descriptive statistics of variables
|
Impact variables |
2013 |
2014 |
2015 |
2016 |
2017 |
2018 |
2019 |
2020 |
2021 |
|
Level of receivables (receivables) |
|
|
|||||||
|
average |
0.0065 |
0.0061 |
0.0059 |
0.0075 |
0.0102 |
0.0122 |
0.0146 |
0.0138 |
0.0142 |
|
minimum |
0.0000 |
0.0000 |
0.0001 |
0.0003 |
0.0006 |
0.0004 |
0.0004 |
0.0004 |
0.0004 |
|
maximum |
0.0329 |
0.0344 |
0.0381 |
0.0316 |
0.0501 |
0.0732 |
0.1584 |
0.1682 |
0.1893 |
|
standard deviation |
0.0068 |
0.0069 |
0.0072 |
0.0079 |
0.0115 |
0.0152 |
0.0288 |
0.0317 |
0.0396 |
|
Selling expense ratio (marketing) |
|
|
|||||||
|
average |
0.0885 |
0.0966 |
0.0992 |
0.1014 |
0.0993 |
0.1039 |
0.1081 |
0.1143 |
0.1375 |
|
minimum |
0.0205 |
0.0220 |
0.0213 |
0.0194 |
0.0212 |
0.0210 |
0.0193 |
0.0204 |
0.0254 |
|
maximum |
0.1697 |
0.1793 |
0.1874 |
0.1992 |
0.1843 |
0.1895 |
0.2074 |
0.2676 |
0.2865 |
|
standard deviation |
0.0476 |
0.0503 |
0.0525 |
0.0525 |
0.0513 |
0.0516 |
0.0546 |
0.0569 |
0.0583 |
|
Fixed asset input intensity (fixed) |
|
|
|||||||
|
average |
0.2708 |
0.2758 |
0.2734 |
0.2818 |
0.2797 |
0.2752 |
0.2671 |
0.2890 |
0.2946 |
|
minimum |
0.0677 |
0.0536 |
0.0538 |
0.0603 |
0.0749 |
0.0662 |
0.0752 |
0.0815 |
0.0947 |
|
maximum |
0.7292 |
0.7269 |
0.7000 |
0.6633 |
0.6608 |
0.6490 |
0.6400 |
0.6695 |
0.6946 |
|
standard deviation |
0.1593 |
0.1604 |
0.1656 |
0.1635 |
0.1528 |
0.1562 |
0.1610 |
0.1677 |
0.1745 |
|
Level of technological input (technology) |
|
|
|||||||
|
average |
0.00126 |
0.00216 |
0.00221 |
0.00183 |
0.00170 |
0.00140 |
0.00148 |
0.00168 |
0.00186 |
|
minimum |
0.00000 |
0.00000 |
0.00000 |
0.00000 |
0.00000 |
0.00001 |
0.00002 |
0.00002 |
0.00002 |
|
maximum |
0.00515 |
0.01507 |
0.01489 |
0.01105 |
0.01014 |
0.00637 |
0.00680 |
0.00695 |
0.00734 |
|
standard deviation |
0.00155 |
0.00318 |
0.00333 |
0.00231 |
0.00218 |
0.00158 |
0.00178 |
0.00186 |
0.00189 |
|
Inventory turnover rate (Invent) |
|
|
|||||||
|
average |
18.64 |
16.38 |
16.64 |
15.95 |
17.18 |
18.91 |
19.79 |
19.92 |
20.17 |
|
minimum |
1.82 |
0.80 |
0.46 |
0.72 |
0.89 |
0.84 |
1.19 |
1.25 |
1.38 |
|
maximum |
106.86 |
68.32 |
64.10 |
64.38 |
78.08 |
95.06 |
116.85 |
134.98 |
150.73 |
|
standard deviation |
21.83 |
17.54 |
16.72 |
16.94 |
19.98 |
23.93 |
28.70 |
32.86 |
37.83 |
|
Gearing ratio (asset) |
|
|
|||||||
|
average |
0.5652 |
0.5514 |
0.5414 |
0.5279 |
0.5299 |
0.5210 |
0.5197 |
0.5158 |
0.5093 |
|
minimum |
0.1998 |
0.1871 |
0.1752 |
0.1865 |
0.1809 |
0.1666 |
0.1534 |
0.1521 |
0.1437 |
|
maximum |
0.9408 |
0.9401 |
0.9356 |
0.9315 |
0.9316 |
0.9298 |
0.9205 |
0.9201 |
0.9188 |
|
standard deviation |
0.1489 |
0.1545 |
0.1664 |
0.1689 |
0.1677 |
0.1781 |
0.1760 |
0.1794 |
0.1821 |
|
Enterprise size (scale) |
|
|
|||||||
|
average |
8.617 |
8.653 |
8.750 |
8.860 |
8.891 |
8.966 |
9.017 |
9.344 |
9.575 |
|
minimum |
6.607 |
6.631 |
6.656 |
6.712 |
6.714 |
6.715 |
7.229 |
7.486 |
7.492 |
|
Continued Table 6 |
|||||||||
|
Impact variables |
2013 |
2014 |
2015 |
2016 |
2017 |
2018 |
2019 |
2020 |
2021 |
|
maximum |
11.327 |
11.317 |
11.386 |
11.829 |
11.966 |
12.203 |
12.375 |
12.464 |
12.581 |
|
(statistics) standard deviation |
0.886 |
0.903 |
0.923 |
1.018 |
1.051 |
1.092 |
1.103 |
1.179 |
1.221 |
|
Total retail sales of consumer goods (retail, in billions of yuan) |
|
|
|||||||
|
average |
8157.45 |
9106.07 |
10069.10 |
11084.04 |
12161.93 |
13134.52 |
14043.80 |
14468.65 |
14679.18 |
|
minimum |
668.51 |
737.18 |
789.60 |
850.10 |
930.40 |
1330.10 |
1399.40 |
1473.45 |
1498.22 |
|
maximum |
25453.93 |
28471.15 |
31517.60 |
34739.10 |
38200.10 |
39767.10 |
42951.80 |
42977.35 |
43176.64 |
|
standard deviation |
4271.67 |
4767.73 |
5312.20 |
5861.13 |
6441.16 |
6739.47 |
7298.13 |
7375.57 |
7396.13 |
Discussion part:
Retail is the basis of commodity circulation, and the operation condition of retail industry is directly related to the speed and efficiency of commodity circulation; retail is the key carrier to guide production and expand consumption, and the development condition of retail industry is related to market prosperity and employment security. Based on the current development situation of retail enterprises in China, this paper summarizes the influence path of retail channel strategy on enterprise efficiency from four perspectives: market demand, channel resources, channel technology and channel system, and empirically analyzes the influence of retail channel strategy on enterprise comprehensive efficiency, which helps guide retail enterprises to adjust channel development strategy, promote the transformation and upgrading of retail industry, release development vitality and enhance development momentum.
Compared with other industries, the implementation of retail channel strategy is closely related to the successful transformation and upgrading of enterprises. With the development of technology, online channel strategies of different nature such as Internet platforms and mobile terminals have provided retail enterprises with new development solutions over time, but as the demographic dividend disappears and the drawbacks of online channels begin to emerge, the retail industry has started to experiment with various new channel practices. This has also led to academic discussions and studies on the relationship between the implementation of retail channel strategies and the efficiency of retail enterprises. However, there is a lack of literature on the relationship between the two, and most of the existing literature is only a description and summary of the symptoms, and further research is needed. Therefore, the theoretical significance of this paper is to further reveal the inner mechanism of the influence of retail channel strategy on enterprise efficiency according to the four elements of channel formation and evolution. The study of retail enterprise efficiency from the perspective of channel strategy selection not only enriches the study of the relationship between channel development and retail enterprise efficiency, but also provides retail enterprises with channel strategy guidance ideas, which has certain theoretical significance.
This paper finds that the implementation of a dual-channel strategy may have a positive impact on the overall efficiency of the firm in the early stage, but does not have a significant impact in the long term. Implementing a channel convergence strategy on this basis will have a positive impact on the overall efficiency of the firm. Given that most of the retailers have already opened dual-channel strategies, online channels do allow retailers with a large number of local stores to break through technological bottlenecks and improve overall efficiency. Therefore, this paper encourages domestic and international retailers implementing dual-channel strategies to implement corresponding channel integration measures after adopting online channel operations.
Based on the above findings, this paper makes the following suggestions:
(1)Focus on demand to promote efficiency. At this stage, retail enterprises need to recognize a problem, new channels or the implementation of various channel strategies should be around the customer, insight into the changes in customer demand, to achieve accurate service. This requires enterprises in all aspects of channel operations should be customer-centric online and offline resource allocation, with the channel to capture customers and seize customers. From the perspective of actual development, channel integration strategy refers to the integration of online stores, physical stores and modern logistics, so that the traditional retailing of people, goods and fields break through the limitations of time and space, bringing higher value returns for enterprises. However, in fact, many retail companies that have implemented convergence strategies have not seen significant growth in performance, and even a decline. The reason for this situation is that this transformation has not produced the expected effect on the improvement of the viscosity of existing users and the attraction of new ones. So to break the diversion dilemma, companies need to do is independent innovation, in the general implementation of the "retail + Internet" logic to add content section into the "retail + Internet + content". In the future, the mass marketing strategy with low marketing efficiency will gradually disappear, and mass marketing will be gradually transformed into niche marketing. Enterprises should first clarify the target customer groups and their needs, and then carry out precise marketing to them, so that customers can get a more ultimate shopping experience. In the channel integration mode, all channels of the enterprise can be used to help customers solve their problems. Each channel has advantages and disadvantages, and retail enterprises must give full play to the advantages of each channel, use and integrate existing resources to achieve mutual diversion of online and offline traffic, so that online traffic and offline traffic can be shared. In this way, enterprises can fully understand customer information and carry out accurate marketing to lay a good foundation.
(2)Good placement of resources to increase efficiency. Facing the impact of e-commerce and consumers' diversified channel demand, many retail enterprises are trying to operate multi-channel and actively implement relevant channel integration strategies, but the rapid development of online sales does not drive all enterprises to improve efficiency. The reason for this is that many traditional retailers are developing online channels as a passive choice and blindly following the trend forced by the market, and do not really understand how to operate online channels. At the same time, retailers often have two misconceptions about online sales: one is that the service cost of online channels is lower than that of physical stores; the other is that digital business growth to a certain scale will bring profitability. However, "more" is not the same as profitability, and there is no positive correlation between multi-channel operations and return on revenue. When companies lay out multi-channel operations, they should fully consider financial costs such as return on investment and develop appropriate channel strategies so that sales growth can really bring about an increase in profitability. With the maturity of the overall business ecology of the Internet, the expansion of online shopping scale has become an irreversible trend, but in the online retail boom, enterprises need to think cold. Although the addition of online channels allows companies to gain objective sales, the growth of digital sales does not necessarily lead to increased profitability, and may even threaten the overall effectiveness of the company. For example, for companies implementing dual-channel and channel convergence strategies, their supply chain and IT costs are much higher than the costs of previous offline channels. Therefore, retailers should realize that the key to building a multi-channel retail model is to look at the whole picture, find suitable partners from the perspective of a higher-level retail ecosystem, focus on improving their own core competitiveness, and attract more consumers through quality experiences.
(3)Enhance the strong effect of technology. Leveraging the "Internet+" to improve information technology capabilities and achieve integrated development in the COVID-19 era. In the study of the path of retail enterprises to influence the efficiency of enterprises through the introduction of technology, it is found that most retail enterprises' existing technology investment is limited to the investment in information software. With the in-depth development of "Internet+", retail enterprises should take advantage of "Internet+", play the role of Internet, coordinate the mutually exclusive influence of different channels, amplify the complementary advantages of channels, and realize multi-channel integration. To achieve integrated development by leveraging "Internet+", the key lies in the following two points. First, actively introduce online channels and implement dual-channel strategy, which should pay attention to the scale efficiency of enterprises and improve the operation process from the global system level to bring into play the synergy effect of multiple channels. Second, the introduction of the Internet and other related advanced technologies should be based on the strategic guidance of the enterprise, and should be introduced in a target-oriented and targeted manner under the condition that the enterprise fully understands its own strengths and weaknesses, rather than blindly following the trend of development and imitating other enterprises, resulting in a decline in enterprise efficiency. With the help of big data, improve data analysis capabilities and realize retail transformation. There are only a few retail enterprises using big data to operate in China, and the application of big data is still in the exploration and exploration stage. However, the advantages of big data are obvious to all, and enterprises can use big data to analyze consumer behavior information and improve category management and marketing strategies. In the era of big data, it is necessary to analyze the data. And to leverage big data to achieve transformation, retail enterprises should first broaden the depth and breadth of the application of big data, access to data and analysis of all parties, on the basis of data analysis to speed up the operation of enterprises in logistics, supply chain and other links, in the enterprise to make decisions to provide accurate more for information. Secondly, enterprises should focus on the development of big data products. In the process of upgrading to a platform-based enterprise can also be used to build a platform ecosystem with the application of big data, to provide help for the partners in the platform and achieve common development.
(4)Strategic management of superior efficiency. The ultimate goal of enterprises is to obtain profits, so when retail enterprises carry out the strategic layout of channels, they should start from the perspective of improving the return on investment, look at the whole picture, and let multi-channel retailing serve the overall development of enterprises. From the current point of view, most domestic retail enterprises in the implementation of multi-channel operation strategy does not establish this global concept, more just a blind behavior. To truly tap the value of multi-channel retailing, Chinese retailers must find the best time and the most suitable investment area for multi-channel layout from the strategic perspective of long-term development, so that multi-channel operation can become a great help for enterprises to break through the offline sales dilemma and achieve benign and rapid development. The fundamental support for multi-channel retailing and even omni-channel retailing is still the offline capability of retail enterprises. Therefore, the key to the sustainable development and long-term profitability of multi-channel retailing model is for retail enterprises to return to their original mindset and continuously enhance their offline capabilities. For example, in the empirical evidence, we found that the inventory turnover rate of retail enterprises has a positive impact on the overall efficiency of enterprises, so enterprises can consider optimizing their management techniques from aspects such as merchandise planning capability, purchasing and transportation capability, ordering and replenishment capability, etc. to improve the turnaround speed to further improve the overall efficiency of the enterprise. With the integration of different channels such as offline, PC and mobile, the decentralized nature of channels will also play a key role in retail inventory optimization, and the multi-channel shared inventory strategy will also effectively improve the speed of product delivery and reduce logistics costs.
Point 2: Is the action contained in the statement: "In addition, in order to ensure that the selected sample is representative of the retail industry, our study also eliminates enterprises whose main business of retailing accounts for less than 60%, and finally obtains 29 retail enterprises as final research sample. " (lines 39-42, p.2) really guarantees sample representativeness?
Response 2: This comment is crucial. For an article, it is important to be able to clearly articulate how the sample was selected. Your valuable comments have made us aware of this problem.
How should the industry be defined for wholesale and retail businesses? China's Commerce Bureau gives the following explanation: "The industry definition of wholesale and retail is distinguished by the purchase object and the actual use of the goods, and those sold directly to the final consumer belong to retail. Enterprises define which industry they belong to by the percentage of wholesale and retail revenue in their actual operations, not by the selling price as wholesale or retail, or the amount of sales."
And in China's Tax Law, many regulations clearly state that "enterprises whose main business income accounts for more than 60% of the total income of the enterprise" are subject to different tax policies. Thus, it is easy to see that the definition of whether an enterprise belongs to the industry is based on whether the main business income of the enterprise exceeds 60% of the total income of the enterprise.
Based on the comment, we further explained in the article why the sample was selected the way it was.They are on lines 32-37 of page 2 in Word's " Track Changes " mode.
Specific modifications:
In addition, in order to ensure that the selected sample is representative of the retail industry, this paper takes the revenue share of retailing in the actual operation of enterprises as the standard according to the regulations of China Tax Law and Commerce Bureau, and eliminates enterprises whose main business share of retailing is below 60%, and finally gets 29 retail enterprises as the final research sample of this paper.
Point 3: The article does not explain the symbols in formulas 1 - 3 (p. 6). MI was not explained until p. 6 (line 34), although it was used before. In line 19, p. 6 the symbol Sg was misspelled.
Response 3: Thank you for your careful review. As suggested, we advanced the explanation of MI to where it appears in the article. And the spelling of symbols has been corrected. They are on lines 12-24 and line 33 of page 6 in Word's " Track Changes " mode.
Specific modifications:
DEA-Malmquist index can be divided into two categories according to the method of its reference frontier: first, Malmquist Index (MI) is calculated by referring to the frontier of different periods; second, MI is calculated by referring to the same frontier for all periods. The MI can be decomposed into the combined technical efficiency change (EC) and the technical change (TC), where the technical progress change is the change in the technical progress of industry production in a given period. The combined technical efficiency change can be decomposed into pure technical efficiency change (PEC) and scale efficiency change (SEC). The relationship between these indicators can be expressed as follows: MI = EC × TC and EC = PEC × SEC. When the MI is greater than 1, it means that the efficiency of the enterprise has improved, and vice versa, it means that the efficiency has decreased. When one of the constituent MI is greater than 1, it indicates that the index is contributing to the efficiency of the enterprise and vice versa.
the reference set for each period is
Point 4: In tables 1 and 4, "broken" text appears (the table is therefore unreadable).
Response 4: We found this suggestion is important. Based on your suggestions, we have reformatted and edited Tables 1 and 4.They are in page 8 and 13 in Word's " Track Changes " mode.
Specific modifications:
Table 1. Basic Retail Listed Companies
|
|
Company Code |
Company Name |
Adding online Channel time |
Implementation of integration Strategy time |
Remarks on integration strategies |
|
1 |
002024 |
Suning.com |
Before 2011 |
2013 |
2013 |
|
2 |
600712 |
Nanning Department Store |
2011 |
2019 |
Completion of the opening of online operations on takeaway platforms for all stores in 2019. |
|
3 |
601116 |
Three Rivers Shopping Mall |
2013 |
2015 |
2015 Partnership with Jingdong to Home |
|
4 |
002264 |
Xinhuadu Shopping Mall |
20216 |
2017 |
End of 2017 |
|
5 |
600814 |
Hangzhou Xiebai |
not |
not |
not |
|
6 |
002561 |
Xujiahui |
2014 |
2016 |
2016 Omnichannel, including online ordering and store pickup |
|
7 |
600828 |
Maoye Commercial |
2019 |
not |
not |
|
8 |
002419 |
Tianhong Stock |
Before 2011 |
2014 |
2014 |
|
9 |
600774 |
Hanshang Group |
2020 |
not |
not |
|
10 |
600693 |
Dongbai Group |
2018 |
2018 |
not |
|
11 |
600729 |
Chongqing Department Store |
2014 |
2018 |
2018 Pilot Mission, Jingdong to Home |
|
12 |
600723 |
Shoushang shares |
2015 |
not |
not |
|
13 |
600361 |
Hualian Supermarket |
2017 |
2017 |
Partnerships with third-party takeaway platforms in 2017 |
|
14 |
601010 |
Wenfeng shares |
2014 |
2018 |
2018 Amoy Fresh, continues to strengthen its partnership with Meituan Hungry |
|
15 |
601933 |
Wing Fai Supermarket |
2014 |
2014 |
September 2014 online order and payment, store pickup; October same-city delivery |
|
16 |
600697 |
Eurasian Group |
2012 |
2016 |
Eurasia to Home goes live in 2016 |
|
17 |
000715 |
Zhongxing Commercial |
2013 |
2018 |
2018 ZTE Cloud Shopping |
|
18 |
000419 |
Tongcheng Holdings |
2012 |
2016 |
2016 Department Store Division to implement the "Shopping without Hassle, Home Delivery" campaign |
|
19 |
600785 |
Xinhua Department Store |
2015 |
2019 |
2019 CCpark Store |
|
20 |
002187 |
Guangbai shares |
2013 |
2016 |
2016 Jingdong to Home
|
|
21 |
002277 |
AUA shares |
2013 |
2017 |
In 2017, we started the 020 model of "offline experience + community community + online micro store APP"; in 2018, we developed Youa Shopping" app synchronized with physical stores |
|
22 |
600857 |
Ningbo Ciba |
2016 |
not |
not |
|
23 |
000417 |
Hefei Department Store |
2012 |
2016 |
2016 Same City Delivery, Mobile App |
|
24 |
000759 |
Zhongbai Group |
2013 |
2017 |
2017 |
|
25 |
600858 |
Ginza shares |
2014 |
2015 |
2015 Supermarket Flash Sale |
|
26 |
002251 |
BBK |
2014 |
2018 |
April 2018 |
|
27 |
600859 |
Wangfujing |
2013 |
2020 |
2020 |
|
28 |
600861 |
Beijing urban and rural areas |
2011 |
2017 |
2017 Community Supermarket Department 118 Supermarket Logistics Centralized Distribution |
|
29 |
600628 |
New world |
2012 |
2013 |
2013 VW, order online and pick up offline. |
Table 4. Malmquist Productivity Index and its decomposition for listed firms in the retail sector (2012-2021)
|
|
Company Name |
PEC |
SEC |
PTC |
STC |
MI |
|
Implementing a dual-channel strategy companies |
||||||
|
Of which: implementation of integration strategies |
Suning.com, PRC e-commerce company |
1.0000 |
1.0000 |
1.0000 |
0.9895 |
0.9895 |
|
Three Rivers Shopping |
1.0000 |
1.0000 |
0.9989 |
0.9977 |
0.9964 |
|
|
Xinhua all over the world |
0.9949 |
0.9991 |
0.9955 |
0.9962 |
0.9862 |
|
|
Xujiahui neighborhood of Shanghai |
1.0000 |
1.0000 |
0.9970 |
0.9986 |
0.9956 |
|
|
Tianhong Stock |
1.0000 |
1.0000 |
1.0002 |
1.0000 |
1.0002 |
|
|
Dongbai Group |
1.0003 |
0.9986 |
0.9926 |
1.0005 |
0.9920 |
|
|
Chongqing Department Store |
0.9976 |
0.9916 |
0.9975 |
1.0016 |
0.9879 |
|
|
Shoushang shares |
0.9952 |
1.0048 |
0.9981 |
0.9995 |
0.9955 |
|
|
Hualian Supermarket |
0.9976 |
0.9999 |
1.0001 |
1.0001 |
0.9977 |
|
|
Wenfeng shares |
1.0000 |
0.9977 |
0.9934 |
0.9996 |
0.9907 |
|
|
Wing Fai Supermarket |
1.0044 |
0.9924 |
1.0006 |
0.9982 |
0.9953 |
|
|
Eurasian Group |
1.0000 |
0.9916 |
1.0000 |
0.9810 |
0.9724 |
|
|
Zhongxing Commercial |
1.0000 |
1.0000 |
1.0000 |
1.0000 |
1.0000 |
|
|
Tongcheng Holdings |
1.0100 |
1.0013 |
0.9967 |
1.0040 |
1.0094 |
|
|
Guangbai shares |
1.0000 |
1.0000 |
1.0011 |
1.0000 |
1.0011 |
|
|
AUA share |
0.9958 |
0.9982 |
0.9972 |
0.9950 |
0.9861 |
|
|
Hefei Department Store |
1.0000 |
0.9982 |
0.9984 |
0.9872 |
0.9836 |
|
|
China Hundred Group |
0.9996 |
1.0024 |
0.9997 |
1.0023 |
1.0037 |
|
|
Ginza shares |
0.9932 |
1.0006 |
0.9981 |
1.0006 |
0.9924 |
|
|
BBK |
1.0046 |
0.9920 |
0.9917 |
1.0066 |
0.9935 |
|
|
Beijing urban and rural areas |
0.9959 |
0.9970 |
0.9871 |
1.0012 |
0.9809 |
|
|
new world |
1.0000 |
1.0000 |
1.0000 |
1.0000 |
1.0000 |
|
|
Average of enterprises implementing integration strategies. |
0.9995 |
0.9984 |
0.9975 |
0.9981 |
0.9932 |
|
|
No integration strategy implemented |
Nanning Department Store |
0.9974 |
1.0020 |
0.9909 |
1.0011 |
0.9890 |
|
Xinhua Department Store |
0.9949 |
0.9991 |
0.9955 |
0.9962 |
0.9862 |
|
|
Ningbo Ciba |
1.0000 |
1.0000 |
1.0000 |
1.0000 |
1.0000 |
|
|
Wangfujing neighborhood of central Beijing, famous for its many shops and restaurants |
1.0000 |
0.9922 |
1.0000 |
1.0003 |
0.9921 |
|
|
Average of enterprises not implementing integration strategies. |
0.9981 |
0.9983 |
0.9966 |
0.9994 |
0.9918 |
|
|
Average of dual-channel strategy firms |
0.9983 |
0.9994 |
1.0073 |
1.0983 |
1.1721 |
|
|
Single-channel strategy |
||||||
|
|
Hangzhou Xiebai |
1.0038 |
1.0004 |
1.0060 |
0.9998 |
1.0106 |
|
Maoye Commercial |
1.0000 |
1.0000 |
1.0008 |
1.0000 |
1.0013 |
|
|
Hanshang Group |
1.0000 |
0.9849 |
0.9786 |
1.0353 |
0.9981 |
|
|
Average of single-channel strategy firms |
1.0013 |
0.9951 |
0.9951 |
1.0117 |
1.0325 |
|
Point 5: Misspelled ait and uit symbols in line 12, p. 15. The symbols in table 7 differ from those shown in models 10 - 11.
Response 5: Thank you for your careful review. As suggested, we corrected the spelling of ait and uit symbols. The symbols in Table 7 have also been corrected to be consistent with those in Models 10-11. They are on line 12 of page 16 and line 33 of page 18 in Word's " Track Changes " mode.
Specific modifications:
ait , uit are the constant term and random error term of the model, respectively.
(10)
Int (11)
Table 7. Tobit random effects panel regression results for the effect of channel strategy on the combined retail efficiency
|
Models 10 |
Models 11 |
||
|
Independent variables |
Regression results |
Independent variables |
Regression results |
|
Ch |
0.0043 (0.67) |
In |
0.0120* (1.84) |
|
Rec |
-0.2447 (-1.04) |
Rec |
-0.5847** (-2.10) |
|
Mar |
-0.1084 (-0.93) |
Mar |
-0.1009 (-0.69) |
|
Fi |
-0.0674* (-1.90) |
Fi |
-0.2138*** (-4.30) |
|
Te |
-1.8435* (-1.71) |
Te |
-0.8612 (-0.64) |
|
Inv |
0.0012** (2.26) |
Inv |
0.0008* (1.92) |
|
Ret |
1.10e-06 (1.06) |
Ret |
2.74e-06* (1.91) |
|
As |
-0.0006 (-0.02) |
As |
-0.0603 (-1.44) |
|
Sc |
-0.0129* (-1.75) |
Sc |
-0.0363*** (-3.00) |
|
P0 |
1.0992*** (16.95) |
P0 |
1.3648*** (11.87) |
Thanks again to you for your careful review of our research paper during your busy schedules!

Reviewer 4 Report
This study employs elaborated models to figure out the relationship between channel strategies and their efficiencies. This paper is well-developed, and the empirical analyses give us many theoretical and practical implications. I would like to discuss how this paper can be more improved:
1. Introduction
I think the introduction could be more tightened. It would be better if this section presents more succinctly the motivation of this research, why this topic is so important, and how this study contributes to the literature.
2. Evidence from China
The contextual aspects could be incorporated into this study. Why this study looks at Chinese enterprises? Are there any opportunities to expand our knowledge on the channel strategies which can improve efficiency? How can the findings of this study be generalized to other countries or contexts? The authors may want to discuss this to make the readership more comprehensive.
3. Tables
Last, for the better readership, Table 4 could be revised or re-arranged. And I might miss something, but in Table 7, I wonder why the model number starts with 10 (like Model 10 and Model 11). Readers might be curious about the numbering.
Anyway, I think this is a great work. I appreciate for all the efforts for this work.
Author Response
Response to Reviewer 4 Comments
Dear reviewer:
Hello, thank you very much for your careful review of our paper in your busy schedule and for your constructive suggestions. We have carefully studied your valuable suggestions and comments, and have revised and improved the paper according to your suggestions and comments (the modified part of the paper is marked out with the "Track Changes" function in Word). The specific responses are as follows:
This study employs elaborated models to figure out the relationship between channel strategies and their efficiencies. This paper is well-developed, and the empirical analyses give us many theoretical and practical implications. I would like to discuss how this paper can be more improved:
Point 1: Introduction
I think the introduction could be more tightened. It would be better if this section presents more succinctly the motivation of this research, why this topic is so important, and how this study contributes to the literature.
Response 1: Thank you very much for this valuable suggestion. We apologize for the looseness of the introduction. As the beginning of the article, the introduction is a very important part. Based on the comment, we divide the original Introduction into two new sections: Introduction and Literature review. We rewrote the introduction into four paragraphs: background, rationale&research questions, summarize and section plan, presents more succinctly the motivation of this research, and why this topic is so important. This is in the lines 27-43 of page 1, lines 1-55 of page 2 and lines 1-8 of page 3 in the Word's " Track Changes " model.
Specific modifications:
- Introduction
Background: The negative impact of the COVID-19 on China's economy still exists, and people's living habits and consumption patterns have changed. With the major domestic cycle as the main body and the dual domestic and international cycles promoting each other is the new pattern to accelerate economic recovery. In the whole domestic cycle, consumption has become the first pulling force of economic growth for six consecutive years, is the main focus of accelerating the release of domestic demand potential, but also to promote the key engine of the dual cycle. As a key vehicle to guide production and expand consumption, the retail industry is directly related to the speed and efficiency of commodity circulation, and is a key support force for national economic growth and social development. Since the retail industry is characterized by high revenue and low profit, the management efficiency of the retail industry is particularly important. Whether the current retail channel strategy can improve the efficiency of retail enterprises and reverse the decline is an urgent and important issue in the development of the retail industry.
Rationale&RQs: Nowadays, the retail industry has become a key force for national economic growth. Not only that, the retail industry has a large number of points and a wide range of characteristics, but also a large volume of labor demand, coupled with a low employment threshold, accommodating many laid-off and transferred labor, the contribution to national employment is also particularly outstanding. The retail industry has gradually become a major industry to promote economic growth and national employment, helping retail innovation and transformation, improving circulation efficiency and promoting consumer upgrading will become a key initiative for high-quality economic development. However, as economic development enters a new normal, the demographic dividend is nearly exhausted, economic growth and residential consumption growth is slowing down, the expansion of China's retail market is restricted after the epidemic, the development of Internet technology has extended retail channels from offline to online, the real economy has been hit, and the new network mobile channels have further intensified the brutal competition pattern in the retail industry. Due to the rapidly changing consumption habits and consumption concepts of the consumer groups, rising retail operating costs and the rapid development of online channels, retail companies are not only facing a series of uncertainties in their investment in online channels, but also the layout and planning of their offline stores and other preliminary investments are difficult to obtain equivalent returns in the short term. This series of reasons lead to many retail enterprises' revenue not being equal to their operating costs. In the face of the above dilemma, can the current channel strategy of retail enterprises improve the overall efficiency of the enterprise? In the era of COVID-19, how should retailers choose their channel strategies to improve their overall efficiency, maintain their business and sustainable development in the future?
Summarize: In summary, changes in the macro environment and intensified competition in the industry have prompted continuous changes in retail channels, and retail transformation is urgent. It is of great theoretical and practical significance to study the relationship between channel strategies of retail enterprises and the overall efficiency of enterprises to transform the retail industry and stimulate market vitality. Considering that the sample needs to meet the requirements of comparability, stability and data availability, this paper first selects the retail enterprises listed in Shenzhen and Shanghai before 2011 according to the sample study period, and excludes the enterprises with major restructuring and missing main variables in the sample period. In addition, in order to ensure that the selected sample is representative of the retail industry, this paper takes the revenue share of retailing in the actual operation of enterprises as the standard according to the regulations of China Tax Law and Commerce Bureau, and eliminates enterprises whose main business share of retailing is below 60%, and finally gets 29 retail enterprises as the final research sample of this paper. Internationalization has increasingly become an important issue and an inevitable development trend in modern retail industry. This study focuses on Chinese enterprises and analyzes the impact of channel strategy on the efficiency of these 29 listed retail enterprises through Data Envelopment Analysis (DEA) and Tobit regression methods, with a view to improving the understanding of channel strategy for the retail industry at home and abroad. The aim is to raise awareness of channel strategy for the retail industry at home and abroad, and to provide useful and feasible suggestions for the channel selection of modern retail enterprises.
Section plan: The study of this paper is divided into seven parts. The first part is the introduction. It introduces the research background of this paper in detail, presents the research topic, and explains the significance of this paper. The second part and the third part mainly comb through the literature related to retail channel strategy and retail firm efficiency, analyze the path and the inner mechanism of constructing retail channel strategy on firm efficiency, and provide methodological and theoretical references for the following empirical analysis and research. Part 4 and Part 5 will conduct the empirical analysis according to the common process of DEA efficiency research. In the fourth part, the efficiency of 29 Chinese listed retail companies is measured by applying the DEA method, and then classified and analyzed according to the different channel strategies implemented. In the fifth section, the impact of channel strategy on the overall efficiency of retailing is empirically demonstrated by using the global covariate efficiency value measured by DEA-Malmquist as the explanatory variable, introducing a dummy variable of whether channel strategy is implemented or not, and controlling for other relevant internal and external possible influencing factors. The sixth section presents the practical and theoretical implications and recommendations of this paper. Finally, the conclusion section presents conclusions to promote the overall efficiency improvement of listed retail firms in their future sustainable development.
Point 2: Evidence from China
The contextual aspects could be incorporated into this study. Why this study looks at Chinese enterprises? Are there any opportunities to expand our knowledge on the channel strategies which can improve efficiency? How can the findings of this study be generalized to other countries or contexts? The authors may want to discuss this to make the readership more comprehensive.
Response 2: We found this suggestion is important. This study focuses on Chinese companies because China is one of the countries that has been affected heavily by the COVID-19, and the impact on China's real economy has had a huge impact on the future growth of domestic retail enterprises. In this context, how retailers will choose their channel strategies to face future growth is the subject of our study. We hope that this paper will expand your knowledge on the channel strategies which can improve efficiency. Therefore, we added a discussion section, offering suggestions for the development of domestic and foreign companies that choose channel strategies in order to suggest a clearer development direction for the management decisions of retail companies. We hope our findings of this study can be generalized to other countries or contexts. They are in the lines 29-53 of page 21, lines 1-54 of page 22, lines 1-54 of page 23 and lines 1-18 of page 24 in Word's " Track Changes " mode.
Specific modifications:
- Discussion
Retail is the basis of commodity circulation, and the operation condition of retail industry is directly related to the speed and efficiency of commodity circulation; retail is the key carrier to guide production and expand consumption, and the development condition of retail industry is related to market prosperity and employment security. Based on the current development situation of retail enterprises in China, this paper summarizes the influence path of retail channel strategy on enterprise efficiency from four perspectives: market demand, channel resources, channel technology and channel system, and empirically analyzes the influence of retail channel strategy on enterprise comprehensive efficiency, which helps guide retail enterprises to adjust channel development strategy, promote the transformation and upgrading of retail industry, release development vitality and enhance development momentum.
Compared with other industries, the implementation of retail channel strategy is closely related to the successful transformation and upgrading of enterprises. With the development of technology, online channel strategies of different nature such as Internet platforms and mobile terminals have provided retail enterprises with new development solutions over time, but as the demographic dividend disappears and the drawbacks of online channels begin to emerge, the retail industry has started to experiment with various new channel practices. This has also led to academic discussions and studies on the relationship between the implementation of retail channel strategies and the efficiency of retail enterprises. However, there is a lack of literature on the relationship between the two, and most of the existing literature is only a description and summary of the symptoms, and further research is needed. Therefore, the theoretical significance of this paper is to further reveal the inner mechanism of the influence of retail channel strategy on enterprise efficiency according to the four elements of channel formation and evolution. The study of retail enterprise efficiency from the perspective of channel strategy selection not only enriches the study of the relationship between channel development and retail enterprise efficiency, but also provides retail enterprises with channel strategy guidance ideas, which has certain theoretical significance.
This paper finds that the implementation of a dual-channel strategy may have a positive impact on the overall efficiency of the firm in the early stage, but does not have a significant impact in the long term. Implementing a channel convergence strategy on this basis will have a positive impact on the overall efficiency of the firm. Given that most of the retailers have already opened dual-channel strategies, online channels do allow retailers with a large number of local stores to break through technological bottlenecks and improve overall efficiency. Therefore, this paper encourages domestic and international retailers implementing dual-channel strategies to implement corresponding channel integration measures after adopting online channel operations.
Based on the above findings, this paper makes the following suggestions:
(1)Focus on demand to promote efficiency. At this stage, retail enterprises need to recognize a problem, new channels or the implementation of various channel strategies should be around the customer, insight into the changes in customer demand, to achieve accurate service. This requires enterprises in all aspects of channel operations should be customer-centric online and offline resource allocation, with the channel to capture customers and seize customers. From the perspective of actual development, channel integration strategy refers to the integration of online stores, physical stores and modern logistics, so that the traditional retailing of people, goods and fields break through the limitations of time and space, bringing higher value returns for enterprises. However, in fact, many retail companies that have implemented convergence strategies have not seen significant growth in performance, and even a decline. The reason for this situation is that this transformation has not produced the expected effect on the improvement of the viscosity of existing users and the attraction of new ones. So to break the diversion dilemma, companies need to do is independent innovation, in the general implementation of the "retail + Internet" logic to add content section into the "retail + Internet + content". In the future, the mass marketing strategy with low marketing efficiency will gradually disappear, and mass marketing will be gradually transformed into niche marketing. Enterprises should first clarify the target customer groups and their needs, and then carry out precise marketing to them, so that customers can get a more ultimate shopping experience. In the channel integration mode, all channels of the enterprise can be used to help customers solve their problems. Each channel has advantages and disadvantages, and retail enterprises must give full play to the advantages of each channel, use and integrate existing resources to achieve mutual diversion of online and offline traffic, so that online traffic and offline traffic can be shared. In this way, enterprises can fully understand customer information and carry out accurate marketing to lay a good foundation.
(2)Good placement of resources to increase efficiency. Facing the impact of e-commerce and consumers' diversified channel demand, many retail enterprises are trying to operate multi-channel and actively implement relevant channel integration strategies, but the rapid development of online sales does not drive all enterprises to improve efficiency. The reason for this is that many traditional retailers are developing online channels as a passive choice and blindly following the trend forced by the market, and do not really understand how to operate online channels. At the same time, retailers often have two misconceptions about online sales: one is that the service cost of online channels is lower than that of physical stores; the other is that digital business growth to a certain scale will bring profitability. However, "more" is not the same as profitability, and there is no positive correlation between multi-channel operations and return on revenue. When companies lay out multi-channel operations, they should fully consider financial costs such as return on investment and develop appropriate channel strategies so that sales growth can really bring about an increase in profitability. With the maturity of the overall business ecology of the Internet, the expansion of online shopping scale has become an irreversible trend, but in the online retail boom, enterprises need to think cold. Although the addition of online channels allows companies to gain objective sales, the growth of digital sales does not necessarily lead to increased profitability, and may even threaten the overall effectiveness of the company. For example, for companies implementing dual-channel and channel convergence strategies, their supply chain and IT costs are much higher than the costs of previous offline channels. Therefore, retailers should realize that the key to building a multi-channel retail model is to look at the whole picture, find suitable partners from the perspective of a higher-level retail ecosystem, focus on improving their own core competitiveness, and attract more consumers through quality experiences.
(3)Enhance the strong effect of technology. Leveraging the "Internet+" to improve information technology capabilities and achieve integrated development in the COVID-19 era. In the study of the path of retail enterprises to influence the efficiency of enterprises through the introduction of technology, it is found that most retail enterprises' existing technology investment is limited to the investment in information software. With the in-depth development of "Internet+", retail enterprises should take advantage of "Internet+", play the role of Internet, coordinate the mutually exclusive influence of different channels, amplify the complementary advantages of channels, and realize multi-channel integration. To achieve integrated development by leveraging "Internet+", the key lies in the following two points. First, actively introduce online channels and implement dual-channel strategy, which should pay attention to the scale efficiency of enterprises and improve the operation process from the global system level to bring into play the synergy effect of multiple channels. Second, the introduction of the Internet and other related advanced technologies should be based on the strategic guidance of the enterprise, and should be introduced in a target-oriented and targeted manner under the condition that the enterprise fully understands its own strengths and weaknesses, rather than blindly following the trend of development and imitating other enterprises, resulting in a decline in enterprise efficiency. With the help of big data, improve data analysis capabilities and realize retail transformation. There are only a few retail enterprises using big data to operate in China, and the application of big data is still in the exploration and exploration stage. However, the advantages of big data are obvious to all, and enterprises can use big data to analyze consumer behavior information and improve category management and marketing strategies. In the era of big data, it is necessary to analyze the data. And to leverage big data to achieve transformation, retail enterprises should first broaden the depth and breadth of the application of big data, access to data and analysis of all parties, on the basis of data analysis to speed up the operation of enterprises in logistics, supply chain and other links, in the enterprise to make decisions to provide accurate more for information. Secondly, enterprises should focus on the development of big data products. In the process of upgrading to a platform-based enterprise can also be used to build a platform ecosystem with the application of big data, to provide help for the partners in the platform and achieve common development.
(4)Strategic management of superior efficiency. The ultimate goal of enterprises is to obtain profits, so when retail enterprises carry out the strategic layout of channels, they should start from the perspective of improving the return on investment, look at the whole picture, and let multi-channel retailing serve the overall development of enterprises. From the current point of view, most domestic retail enterprises in the implementation of multi-channel operation strategy does not establish this global concept, more just a blind behavior. To truly tap the value of multi-channel retailing, Chinese retailers must find the best time and the most suitable investment area for multi-channel layout from the strategic perspective of long-term development, so that multi-channel operation can become a great help for enterprises to break through the offline sales dilemma and achieve benign and rapid development. The fundamental support for multi-channel retailing and even omni-channel retailing is still the offline capability of retail enterprises. Therefore, the key to the sustainable development and long-term profitability of multi-channel retailing model is for retail enterprises to return to their original mindset and continuously enhance their offline capabilities. For example, in the empirical evidence, we found that the inventory turnover rate of retail enterprises has a positive impact on the overall efficiency of enterprises, so enterprises can consider optimizing their management techniques from aspects such as merchandise planning capability, purchasing and transportation capability, ordering and replenishment capability, etc. to improve the turnaround speed to further improve the overall efficiency of the enterprise. With the integration of different channels such as offline, PC and mobile, the decentralized nature of channels will also play a key role in retail inventory optimization, and the multi-channel shared inventory strategy will also effectively improve the speed of product delivery and reduce logistics costs.
Point 3: Tables
Last, for the better readership, Table 4 could be revised or re-arranged. And I might miss something, but in Table 7, I wonder why the model number starts with 10 (like Model 10 and Model 11). Readers might be curious about the numbering.
Response 3: Thank you for your careful review. As suggested, we revised and reformatted Table 4.They are in page 13 in Word's " Track Changes " mode.And Table 7 presents the estimates of the coefficients of the explanatory variables obtained based on models 10 and 11, which we added explanation on lines31-32 of page 18 in Word's " Track Changes " mode.
Specific modifications:
Table 7 gives the estimated values of the coefficients of the explanatory variables obtained based on models (10) and (11).
Table 4. Malmquist Productivity Index and its decomposition for listed firms in the retail sector (2012-2021)
|
|
Company Name |
PEC |
SEC |
PTC |
STC |
MI |
|
Implementing a dual-channel strategy companies |
||||||
|
Of which: implementation of integration strategies |
Suning.com, PRC e-commerce company |
1.0000 |
1.0000 |
1.0000 |
0.9895 |
0.9895 |
|
Three Rivers Shopping |
1.0000 |
1.0000 |
0.9989 |
0.9977 |
0.9964 |
|
|
Xinhua all over the world |
0.9949 |
0.9991 |
0.9955 |
0.9962 |
0.9862 |
|
|
Xujiahui neighborhood of Shanghai |
1.0000 |
1.0000 |
0.9970 |
0.9986 |
0.9956 |
|
|
Tianhong Stock |
1.0000 |
1.0000 |
1.0002 |
1.0000 |
1.0002 |
|
|
Dongbai Group |
1.0003 |
0.9986 |
0.9926 |
1.0005 |
0.9920 |
|
|
Chongqing Department Store |
0.9976 |
0.9916 |
0.9975 |
1.0016 |
0.9879 |
|
|
Shoushang shares |
0.9952 |
1.0048 |
0.9981 |
0.9995 |
0.9955 |
|
|
Hualian Supermarket |
0.9976 |
0.9999 |
1.0001 |
1.0001 |
0.9977 |
|
|
Wenfeng shares |
1.0000 |
0.9977 |
0.9934 |
0.9996 |
0.9907 |
|
|
Wing Fai Supermarket |
1.0044 |
0.9924 |
1.0006 |
0.9982 |
0.9953 |
|
|
Eurasian Group |
1.0000 |
0.9916 |
1.0000 |
0.9810 |
0.9724 |
|
|
Zhongxing Commercial |
1.0000 |
1.0000 |
1.0000 |
1.0000 |
1.0000 |
|
|
Tongcheng Holdings |
1.0100 |
1.0013 |
0.9967 |
1.0040 |
1.0094 |
|
|
Guangbai shares |
1.0000 |
1.0000 |
1.0011 |
1.0000 |
1.0011 |
|
|
AUA share |
0.9958 |
0.9982 |
0.9972 |
0.9950 |
0.9861 |
|
|
Hefei Department Store |
1.0000 |
0.9982 |
0.9984 |
0.9872 |
0.9836 |
|
|
China Hundred Group |
0.9996 |
1.0024 |
0.9997 |
1.0023 |
1.0037 |
|
|
Ginza shares |
0.9932 |
1.0006 |
0.9981 |
1.0006 |
0.9924 |
|
|
BBK |
1.0046 |
0.9920 |
0.9917 |
1.0066 |
0.9935 |
|
|
Beijing urban and rural areas |
0.9959 |
0.9970 |
0.9871 |
1.0012 |
0.9809 |
|
|
new world |
1.0000 |
1.0000 |
1.0000 |
1.0000 |
1.0000 |
|
|
Average of enterprises implementing integration strategies. |
0.9995 |
0.9984 |
0.9975 |
0.9981 |
0.9932 |
|
|
No integration strategy implemented |
Nanning Department Store |
0.9974 |
1.0020 |
0.9909 |
1.0011 |
0.9890 |
|
Xinhua Department Store |
0.9949 |
0.9991 |
0.9955 |
0.9962 |
0.9862 |
|
|
Ningbo Ciba |
1.0000 |
1.0000 |
1.0000 |
1.0000 |
1.0000 |
|
|
Wangfujing neighborhood of central Beijing, famous for its many shops and restaurants |
1.0000 |
0.9922 |
1.0000 |
1.0003 |
0.9921 |
|
|
Average of enterprises not implementing integration strategies. |
0.9981 |
0.9983 |
0.9966 |
0.9994 |
0.9918 |
|
|
Average of dual-channel strategy firms |
0.9983 |
0.9994 |
1.0073 |
1.0983 |
1.1721 |
|
|
Single-channel strategy |
||||||
|
|
Hangzhou Xiebai |
1.0038 |
1.0004 |
1.0060 |
0.9998 |
1.0106 |
|
Maoye Commercial |
1.0000 |
1.0000 |
1.0008 |
1.0000 |
1.0013 |
|
|
Hanshang Group |
1.0000 |
0.9849 |
0.9786 |
1.0353 |
0.9981 |
|
|
Average of single-channel strategy firms |
1.0013 |
0.9951 |
0.9951 |
1.0117 |
1.0325 |
|
Point 4: English language and style are fine/minor spell check required.
Response 4: Thank you for this valuable suggestion. As suggested, we have combed through the entire paper, and revised and corrected some expressions.
Thanks again to you for your careful review of our research paper during your busy schedules!

Round 2
Reviewer 1 Report
The authors make a great effort. The paper should be accepted for publication.